J Physiol 603.19 (2025) pp 5477–5508

5477

# Endurance training increases a ubiquitylated form of histone H3 in the skeletal muscle, supporting *Notch1* upregulation in an MDM2-dependent manner

Brian Lam[1], Manpreet Gulri[1], Sokaina Akhtar[1] 🆔, Pierre Lemieux[1], Monica Tawadrous[1] 🆔, Mayoorey Murugathasan[2], Ali A. Abdul-Sater[2] and Emilie Roudier[1] 🆔

[1]*Angiogenesis Research Group, School of Kinesiology and Health Science and the Muscle Health Research Centre, Faculty of Health, York University, Toronto, Ontario, Canada*
[2]*School of Kinesiology and Health Science and the Muscle Health Research Centre, Faculty of Health, York University, Toronto, Ontario, Canada*

Handling Editors: Paul Greenhaff & Kevin Murach

The peer review history is available in the Supporting Information section of this article (https://doi.org/10.1113/JP288947#support-information-section).

The Journal of Physiology

**Abstract figure legend** Whether epigenetic silencing histone marks play a role once skeletal muscle adaptations have occurred after endurance training remains unclear. In the Polycomb repressive complex 2 (PRC2), EZH2 tri-methylates lysine 27 on histone H3 (H3K27$^{me3}$) favouring a repressive chromatin state. As a proof-of-concept, we assessed the impact of training on muscle H3K27$^{me3}$. C57Bl6 mice ran for 9 weeks. Training increased the abundance of a ubiquityl-form of H3 (H3$^{Ub}$) in mixed and glycolytic-dominant muscles. Training led to an H3K27$^{me3}$ enrichment on the promoter of genes important for muscle remodelling: *Kdr* and *Notch1*. Following the canonical model, *Kdr* expression decreased following training. Yet, *Notch1* mRNA was upregulated. Our results provide evidence that MDM2-dependent ubiquitylation of H3 is required to activate *Notch1* expression after repeated exposures to contractile activity. The discovery of H3$^{Ub}$ might explain the dichotomic effect of H3K27$^{me3}$ marking on *Kdr* and *Notch1* genes.

M. Gulri, S. Akhtar and P. Lemieux contributed equally to this work and as second co-authors.

**Abstract**   At the onset of training, each exercise session transiently shifts the distribution of histone post-transcriptional modifications (HPTMs) to activate genes that drive muscle adaptations. The resulting cyclic changes in gene expression promote the acquisition of high oxidative capacities and gains in capillaries. If training stops or remains at the same intensity, adaptation ceases. Whether silencing HPTMs helps to halt adaptation remains understudied. The E3 ubiquitin ligase murine double minute (MDM2) and enhancer of zester homolog 2 (EZH2) interact and tri-methylate histone H3 on lysine 27 ($H3K27^{me3}$), silencing genes. C57Bl6 mice ran for 9 weeks (5 days a week) maintaining a constant running speed for the last 5 weeks of training. Muscles were collected 72 h after the last run. Training increased MDM2 and EZH2 proteins and led to an $H3K27^{me3}$ enrichment in *Kdr* and *Notch1* regulatory sequences. *Kdr* mRNA levels decreased, following the canonical model that $H3K27^{me3}$ silences genes. *Notch1* mRNA increased. Trained muscles had greater levels of $H3K27^{me3}$ detected at 25 kDa and no change at the expected molecular weight of 17 kDa. The 25 kDa band was identified as a ubiquitylated form of H3 ($H3^{Ub}$). C2C12 myotubes exposed to four consecutive days of 90 min electrostimulation had higher levels of $H3^{Ub}$. EZH2 inhibition counteracted the electrostimulation-driven accumulation of $H3^{Ub}$ and increased *Notch1* mRNA. Serdemetan, an MDM2 ring domain inhibitor, reduced *Notch1* mRNA and $H3^{Ub}$ level in myotubes. MDM2-dependent ubiquitylation of H3 might upregulate *Notch1* when endurance training ceases. The role $H3^{Ub}$ plays in establishing a new muscle homeostasis remains unclear.

(Received 15 April 2025; accepted after revision 1 August 2025; first published online 25 August 2025)

**Corresponding author** E. Roudier: Angiogenesis Research Group, School of Kinesiology and Health Science and the Muscle Health Research Centre, Faculty of Health, York University, Rm. 427 Life Sciences Building, 4700 Keele St., Toronto, ON M3J 1P3, Canada.    Email: eroudier@yorku.ca

**Key points**

- Whether epigenetic silencing histone marks play a role once skeletal muscle adaptations have occurred following endurance training remains unclear.
- The E3 ubiquitin ligase MDM2 and the epigenetic writer EZH2 interact to establish $H3K27^{me3}$ marks that silence genes, and endurance training increased the expression of both proteins.
- After weeks of training new capillaries were established, and lower levels of *Kdr* mRNA and increased $H3K27^{me3}$ marking on *Kdr* regulatory sequences question whether silencing of this positive regulator of angiogenesis is required to halt microvascular remodelling.
- Training increases skeletal muscle abundance of a ubiquitylated form of H3 ($H3^{Ub}$); in myotubes EZH2 inhibition limits $H3^{Ub}$ accumulation after contractile activity repeated over 4 days and MDM2 inhibition reduces $H3^{Ub}$ levels and upregulates *Notch1* expression.
- MDM2-dependent ubiquitylation of H3 might explain why $H3K27^{me3}$ enrichment fails to silence *Notch1* after training; whether $H3^{Ub}$ is crucial to halt adaptation and establish a new muscle homeostasis requires further investigation.

Brian Lam is a final-year PhD student at York University, Canada. He holds bachelor's and master's degrees in Kinesiology and Health Science from York University. His doctoral research focuses on the epigenetic regulation of exercise-driven angio-adaptation. Currently, he is investigating the role of exercise-induced epigenetic factors in mediating paracrine signalling within muscle tissue. **Emilie Roudier** is an associate professor in the school of Kinesiology and Health Science at York University, Canada. Her research team studies how beneficial (e.g. physical activity) or detrimental environmental factors (e.g. air pollution) establish epigenetic marks on cells and tissues to better understand how small blood vessels remain healthy.

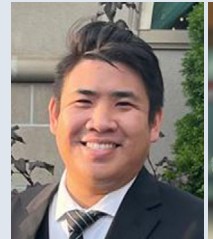
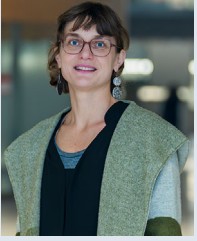

## Introduction

The skeletal muscle represents about 40% of the whole-body mass. This tissue supports movement as well as cardiovascular and metabolic health. To properly function, the skeletal muscle is equipped with an impressive network of vascular capillaries (Wagenmakers et al., 2016). The muscle and its capillary network are highly plastic and subjected to important remodelling to optimize blood supply and muscle function when chronic change in muscle workload occurs (Bloor, 2005; Booth et al., 2015; Egginton, 2009). Endurance training leads to metabolic adaptation and a coordinated expansion of the skeletal muscle capillary network through angiogenesis, the formation of blood vessels from pre-existing capillaries (Arany, 2008; Egginton, 2009; Hudlicka et al., 1992; Narkar et al., 2011).

Over the last three decades, multiple studies have helped elucidate the molecular processes that support the skeletal muscle adaptation to endurance training (reviewed in Booth et al., 2015, 2015). During one bout of endurance exercise, contractile activity transiently changes the microenvironmental stimuli present in the skeletal muscle. These changes modulate the expression of metabolic and angiogenesis-related genes in the muscle fibres, satellite muscle cells and microvascular endothelial cells. Oscillations in the activity of numerous transcription factors and co-activators, such as forkhead box O protein (FOXO), hypoxia inducible factor-1 (HIF1), tumour protein-53 (TP53), myocyte enhancer factor-2 (MEF2), activator protein-1 (AP1) or peroxisome proliferator activated gamma coactivator-$\alpha$ (PGC-1$\alpha$), lead to timely changes in gene expression (Arany et al., 2008; Baresic et al., 2014; Egan & Zierath, 2013; Puntschart et al., 1998; Tachtsis et al., 2016). The oscillations help in coordinating mitochondrial biogenesis, angiogenesis, fibre type transition and satellite cell renewal. This is achieved through changes in the expression of genes, such as *Myh7*, *Notch1*, *Vegfa* and *Kdr*, coding for Myosin heavy chain $\beta$, Notch receptor 1, and Vascular endothelial growth factor-A or its receptor 2, respectively (Chinsomboon et al., 2009, 2009; Geng et al., 2010; Lloyd et al., 2003; Narkar et al., 2011; Palstra et al., 2014; Pinto et al., 2025).

The role of epigenetic processes (i.e. methylation and histone modifications) in skeletal muscle adaptation to training has generated considerable interest (Jacques et al., 2019; Landen et al., 2019; McGee & Hargreaves, 2019). Yet, the role of histone post-translational modifications (HPTMs) remains not fully understood. Enrichment in histone H3 acetylation (H3$^{Ac}$) and in trimethylation of H3 on lysine 4 (H3K4$^{me3}$) relax the chromatin structure in genes. While these activation marks emerge as modulators of the skeletal muscle phenotype (Kawano et al., 2015; Pandorf et al., 2009), new evidence supports

the idea that H3$^{Ac}$ and H3K4$^{me3}$ marks could control skeletal muscle adaptation to endurance exercise. At the onset of training, the first bout of endurance exercise stimulates the expression of thousands of genes in the skeletal muscle. This massive activation of genes correlates with an increase in the global abundance of histone H3 acetylation on lysine 36 (H3K36$^{Ac}$) (McGee et al., 2009). Indeed, H3K4$^{me3}$ activation marks are more abundant following endurance exercise on the alternate promoter of *Pgc1α* (Lochmann et al., 2015). When endurance exercises are repeated regularly over weeks, histone activation marks (i.e. H3K4$^{me1}$ and H3K27$^{Ac}$) are enriched in the enhancer sequences of genes that are upregulated by this endurance training programme (Williams et al., 2021). This epigenetic process stimulates the expression of genes involved in tissue remodelling, lipid metabolism and immune response (Williams et al., 2021). After weeks of training, a bout of exercise has a reduced capacity to generate transcriptional changes in the skeletal muscle (Barrès et al., 2012; Popov et al., 2019). This suggests that many genes that were initially responsive to one bout of endurance exercise could then be silenced after training. Yet, our knowledge of the role of silencing histone marks during endurance training remains unclear.

Our previous work indicates that the E3 ubiquitin ligase MDM2 (murine double minute 2) supports skeletal muscle vascular homeostasis and endurance exercise angiogenesis (Roudier et al., 2012). MDM2 angiogenic properties rely, at least in part, on its capacity to bind and regulate transcription factors, such as TP53, HIF-1$\alpha$ and FOXO1 (Aiken et al., 2016; Fu et al., 2009; Muthumani et al., 2014; Nieminen et al., 2005; Pfaff et al., 2018; Wang et al., 2019). MDM2 can act as a chromatin modifier in stem cells and cancer cell lines through its capacity to bind to the chromatin and to histones (Minsky & Oren, 2004; Riscal, Le Cam et al., 2016; Riscal, Schrepfer et al., 2016). MDM2 also interacts with elements of the polycomb repressive complexes (PRCs; Klusmann et al., 2018; Wienken et al., 2016). Through binding to enhancer of zeste homolog 2 (EZH2), a core subunit of PRC2, MDM2 supports the trimethylation of histone H3 on lysine 27 (H3K27$^{me3}$) in stem cells and cancer cell lines. After H3K27$^{me3}$ marking of regulatory promoters followed by the subsequent ubiquitylation of histone H2A at lysine 199 (H2AK119$^{Ub}$), a sequence of events supports silencing of genes (Wienken et al., 2016). Based on our previous observation that endurance training increases the expression of MDM2 protein in skeletal muscle (Roudier et al., 2013), here we investigated whether training changes the abundance of MDM2-EZH2-related histone silencing marks in the skeletal muscle after adaptations have settled.

## Methods

### Ethical approval

All mice were purchased from Charles River Inc. (Saint Constant, QC, Canada), and animal-related procedures were approved by York University animal care committee in accordance with Canadian Council on Animal Care (CCAC) regulations (Protocol 2020-03). All investigators involved in handling animals understand the animal ethical principles under which the *Journal of Physiology* operates.

### Mouse training protocol

Fourteen female C57Bl6 mice arrived in the animal facilities at the age of 7 weeks. All mice were fed the standard chow *ad libitum*. The mice were house under a 12 h light/12 h dark cycle After 1 week of quarantine, all mice were acclimatized to treadmill exercise for 5 days at 20 m/min prior to the 8 week training programme, before being randomized in a trained *vs.* sedentary group ($n = 7$ per group) (trained and sedentary, at the age of 9 weeks) Sedentary mice were allowed to roam their cages. During the training programme, trained mice ran for 1 h per day, 5 days per week, as previously described (Murugathasan et al., 2023). The speed was gradually increased up to 30 m/min as the training programme progressed: week 1: 20 m/min, week 2: 22 m/min, week 3: 24.2 m/min, week 4: 26.6 m/min, weeks 5–9: 30 m/min. This represents speeds of 1.2–1.8 km/h. The mice ran at approximately 70–75% of their $\dot{V}_{O_2max}$, which can be characterized as moderate to vigorous intensity. Sedentary mice were exposed to the same environment and manipulations. Instead of running, sedentary mice were put in and out of the treadmill and stay in their cage exposed to the sound of the treadmill as the trained group was performing their running session between 09.00 h and noon. After 9 weeks (1 week of treadmill acclimatization + 8 weeks of training), sedentary and trained mice were anaesthetized (isoflurane/oxygen inhalation) 72 h after the last bout of exercise following a similar schedule of exposure to the treadmill room environment or running session. Inhaled isoflurane/oxygen was used to achieve deep anaesthesia during tissue collection. First the animal is placed in the induction chamber with no isoflurane flow but with oxygen flow (1–2 L/min). Then, isoflurane flow was increased gradually (0.5% up to 5%). When the animal reaches deep anaesthesia, the isoflurane flow is decreased to 3%. The animal is transferred to the surgery table and immediately placed into a non-rebreathing tubing to maintain isoflurane delivery. Mice were placed on a warming pad to maintain core body temperature while under anaesthesia. Our previous experience with sedentary and trained mice suggests that 3% isoflurane with 1.5 L/min oxygen flow rate is usually sufficient for mice of this age and weight to be maintained under deep anaesthesia. During anaesthesia, animals were monitored for absence of pain reflex (foot withdrawal from pinch) prior to commencing surgery. Animals were under close observation during the full anaesthesia procedure, to ensure that deep anaesthesia was maintained throughout the protocol. Several lower limb muscles (including the soleus, gastrocnemius, plantaris and tibialis anterior) were removed, weighed and snap-frozen in liquid nitrogen. The extensor digitorum longus (EDL) muscles were embedded in OCT on a spatula from tendon to tendon to maintain the structure of the muscle. EDL muscles were then frozen in isopentane and cooled by liquid nitrogen. The mice were killed by cervical dislocation. The gastrocnemius, soleus and plantaris were utilized for protein and mRNA studies. The EDL muscle was utilized for immunohistochemistry. The gastrocnemius and tibialis anterior were used for chromatin immunoprecipitation (ChIP) experiments.

### Determination of capillary to fibre ratio by immunohistochemistry

Three cryosections were performed per animal in the mid-belly of the EDL muscle. Cross-sections 10 μm thick were allowed to dry for a few minutes before being fixed with 3.7% formalin solution in PBS for 15 min. After three washes with PBS, cross-sections were incubated for 1 h (room temperature) with Griffonia Simplicifolia Lectin I (GSL I) isolectin B4 conjugated to DyLight® 594 (#DL-1207; Vector Laboratories, Burlington, ON, Canada, dilution 1:333 in PBS) and Wheat Germ Agglutinin (WGA) conjugated to Alexa Fluor™ 350 (#W11263; Invitrogen, Waltham, MA, USA, dilution 1:500 in PBS). After three washes with PBS, slides were mounted with coverslips using Vectashield mounting medium (#H-1400; Vector Laboratories). Imaging was performed using an Axiovert 200M microscope (Carl Zeiss Canada Ltd., Toronto, ON, Canada) and imaged with a Hamamatsu cooled digital CCD camera using the Metamorph software. For each stain, identical exposure settings were used across specimens. For each animal, three cross-sections were stained. For each section, capillaries and muscle fibres were counted to determine the capillary-to-fibre ratio. The average of the three sections represents the final value for each animal ($n = 6$ per group).

### Cell culture

C2C12 myoblasts were purchased from ATCC (ATCC, Rockville, MD, USA, cat. no. CRL-1772). C2C12 cells were cultured onto six-well plates with a loading density

of 800,000 cells per well and cultivated in growth media (DMEM, Wisent Bio Products, cat. no. 319010; St Bruno, QC, Canada) supplemented with 10% fetal bovine serum (FBS, cat. no. 080150; Wisent Bio Products) and an antibiotic solution containing 10,000 U/mL penicillin and 100 mg/mL streptomycin (Gibco, cat. no. 15140122; Thermofisher, Burlington, ON, Canada). Once C2C12 myoblasts reached confluency (90–95%), the C2C12 myotubes were differentiated into myotubes through serum starvation; the growth medium was replaced with a differentiation medium: DMEM containing 2% heat-inactivated horse serum (Wistent Bio Products, cat. no. 065250) and 1% penicillin-streptomycin.

C57BL/6 mouse primary skeletal muscle microvascular endothelial cells (mSMECs) were purchased from Cell Biologics (cat. no. C57-6220,Cedarlane, Burlington, ON, Canada). All primary microvascular endothelial cells were platted on gelatin-coated culture dishes (Bioshop, cat. no. GEL771.500, Burlington, ON, Canada). Cells were cultured in complete Endothelial Cell Media with supplements (Complete Mouse Endothelial Cell Medium, Cell Biologics, cat. no. M1168, Cedarlane).

### Electric pulse stimulation

Differentiated C2C12 myotubes were subjected to electrostimulation (electrical pulse stimulation, EPS) to mimic skeletal muscle contraction observed at submaximal aerobic exercise (Carter et al., 2001; Connor et al., 2001; Vepkhvadze et al., 2021). C2C12 myotubes were allowed to differentiate for 5 days prior to being exposed to EPS. The C2C12 cells were stimulated using a six-well cell culture lid which was modified with two parallel platinum wire electrodes extending into the wells containing medium. A total volume of 5 mL per well was required to ensure consistent and equal delivery of EPS throughout the media. Cells were stimulated using a Harvard Apparatus Stimulator CS System (Harvard Apparatus, Montreal, QC, Canada); 8 V was applied at a frequency of 5 Hz with alternating polarity of electrical current (5 ms) for 90 min once a day for four consecutive days. These conditions support synchronous contractions below maximal contractility and avoid tetanic contractions (Connor et al., 2001; Marotta et al., 2004; Tamura et al., 2020a). The cultured myotubes were harvested 3 h after the last bout of EPS.

### Inhibition of EZH2 and MDM2 activity

Stock solutions of Serdemetan and GSK343 were dissolved in DMSO (Sigma Millipore, Burlington, ON, Canada, cat. no. D2650) prepared at $1000\times$. After 4 days of differentiation, C2C12 myotubes were incubated with an EZH2 inhibitor (GSK343; 1 and 5 μM, Cell Signaling, Danvers, MA, USA, cat. no. 66244), an MDM2 inhibitor

(Serdemetan; 1 μM, SelleckChem, Cedarlane, Burlington, ON, Canada, cat. no. S1172) or vehicle control (DMSO, dilution 1:1000, Thermofisher, Burlington, ON, Canada) for 24 or 48 h. Inhibitor-treated and vehicle control myotubes were subsequently subjected to EPS for 90 min a day for 4 days (as described above) before cell harvesting.

### Determination of fusion index by immunohistochemistry

Myotubes were stained for 10 min with 4% formaldehyde. After permeabilization (0.1% triton X-100) for 10 min, C2C12 myotubes were blocked with 2% BSA for 45 min at room temperature. The cell culture was stained with Myosin 4 antibody (MAb MF20) conjugated to Alexa fluor 488 (1:100 in 0.1% BSA, cat. no. 53-6503-83, Thermofisher). Nuclei were stained using the ProLong™ Diamond Antifade Mountant with DAPI (cat. no. P36966, Thermofisher). Images were taken using an Axiovert 200M microscope (Carl Zeiss Canada Ltd.) and imaged with a Hamamatsu cooled digital CCD camera using the Metamorph software. The fusion index was calculated on five biological replicates per condition as the ratio of the number of nuclei present in differentiated myotubes positive for MYH4 *versus* the total number of nuclei.

### Proteasome inhibition

A stock solution of MG-132 was dissolved in DMSO (Sigma Millipore, Burlington, ON, Canada, cat. no. D2650) prepared at $1000\times$. Differentiated C2C12 myotubes were treated for 6 h with MG132 (10 μM, SelleckChem, Cedarlane, Burlington, ON, Canada, cat. no. S2619) and proteins were extracted for immunoblotting analysis.

### Immunoblotting

Immunoblotting was carried out on protein extracts from mouse muscles, C2C12 myotubes and primary endothelial cells as previously described (Aiken et al., 2016; Lemieux et al., 2022). Briefly, muscle and cell samples were homogenized in a protein lysis buffer (ratio weight:volume of 1:15) composed of 50 mM Tris-base, 100 mM NaCl, 5 mM EDTA, 1% sodium deoxycholate, 1% triton X-100, 1 mM phenylmethylsulfonyl fluoride (PMSF), 1 mM NaF, 1 mM $Na_3VO_4$, protease inhibitor cocktail (Roche Complete Mini, cat. no. 04906845001; Sigma-Aldrich, Oakville, ON, Canada), phosphatase inhibitor cocktail (Roche PhosSTOP cat. no. 11836153001; Sigma-Aldrich), pH 8. After cell lysis or tissue homogenization, samples were briefly sonicated and incubated for 20 min at 4°C under 360° rotation. Homogenates were centrifuged at 16,000 *g* for 15 min, and supernatants were collected.

After total protein concentration determination (BCA assay, cat. no. B9643; Sigma-Aldrich), 20 µg of protein samples were denatured, separated by SDS-PAGE and blotted onto nitrocellulose membranes (Whatman BA95; Sigma-Aldrich). Quality of the transfer was controlled by Ponceau S Red staining. After blocking with 5% fat-free milk (dissolved in Tris-buffered saline with 0.1% Tween 20), the blots were probed with appropriate primary antibodies.

The following primary antibodies were used: rabbit polyclonal (pAb) $\alpha$-PECAM (CD31) (Invitrogen; cat. no. PA5-16301, Burlington, Canada), rabbit pAb $\alpha$-COX IV (Cell Signaling Technology; cat. no. 4844, New England Biolabs Ltd., Whitby, ON, Canada), rabbit monoclonal (mAb) $\alpha$-MDM2 (SMP14) (Santa Cruz; cat. no. sc-965, Santa Cruz, CA, USA) and mouse mAb (2A10) (non-commercial, kindly provided by Dr Mary Ellen Perry, NIH), rabbit mAb $\alpha$-EZH2 (D2C9, Cell Signaling Technology, cat. no. 5246), rabbit pAb $\alpha$-NEDD4 (Cell Signaling Technology; cat. no. 2740, New England Biolabs Ltd.), mouse mAb $\alpha$-Ubiquitin (P4D1) (Cell Signaling Technology; cat. no. 3936, New England Biolabs Ltd.), mouse mAb $\alpha$-Histone H3 (C96C10) (Cell Signaling Technology; cat. no. 3638, New England Biolabs Ltd.), rabbit pAb $\alpha$-Histone H2A (Cell Signaling Technology; 2578, New England Biolabs Ltd.), $\alpha$-H3K27$^{me3}$: rabbit pAb-195 and pAb-069 (cat. no. C15410195 and C15410069, respectively, both Diagenode, Denville, NJ, USA) and rabbit mAb (C36B11) (Cell Signaling cat. no. 9733, New England Biolabs Ltd.), rabbit pAb $\alpha$-H3K4$^{me3}$ (cat. no. 9727, New England Biolabs Ltd.), and rabbit mAb (D27C4) $\alpha$-H2AK119$^{ub}$ (cat. no. 8240). $\alpha/\beta$-TUBULIN (rabbit pAb, Cell Signaling Technology; cat. no. 21 485. New England Biolabs Ltd.), and $\beta$-ACTIN (mouse mAb 8H10D10, Cell Signaling Technology, cat. no. 3700) were detected as loading controls. After incubation with the appropriate HRP-conjugated secondary antibody, either $\alpha$-rabbit or $\alpha$-mouse (Cell Signaling Technology, cat. no. 7074 and 7076, respectively, New England Biolabs Ltd.) proteins were visualized with enhanced chemiluminescence (Pierce ECL; cat. no. 32106; Thermofisher Scientific, Burlington, ON, Canada) on an imaging station (iBright$^{TM}$ CL1500 Imaging System, Thermo Fisher Scientific) and quantified using iBright$^{TM}$ Analysis Software.

### Measure of mRNA

Skeletal muscle RNA isolation and purification were performed using an RNeasy fibrous tissue mini kit (Qiagen; cat. no. 74704, Valencia, CA, USA). In total, 15 mg of gastrocnemius and plantaris muscles were lysed in RLT kit buffer and homogenized in a Retsch MM400 tissue homogenizer. Cell culture RNA isolation and purification were performed using the miRNeasy micro kit (Qiagen; cat. no. 217084). RNA was quantified using a Gen5 plate reader (Biotek) at 1:20 dilution and 260/280 nm absorbance. Reverse transcription was done using 500 ng of RNA extracted from muscles or cells using the high-capacity RNA to cDNA kit (cat. no. 4387406, Thermofisher Scientific). Real-time qPCR was run using Taqman probes (listed below) and Taqman fast advanced master mix (cat. no. 4444557, Thermofisher Scientific) in triplicate. PCR was run for 40 cycles of denaturation at 95°C for 3 s, annealing and extension at 60°C for 30 s. *Kdr*, *Notch1*, V*egfa*, *Thbs1* and *Hhip* were measured (a list of probes is given in Table 1). The $\Delta\Delta Ct$ method was used to express the relative changes (Livak & Schmittgen, 2001). mRNA transcripts were normalized to the hypoxanthine-guanine phosphoribosyl transferase (*Hprt*) gene. Data are expressed as $2^{-\Delta\Delta Ct}$.

### Immunoprecipitation

**C2C12 myotubes.** Differentiated C2C12 myotubes were subjected to EPS for 90 min a day, once a day, for four consecutive days before cell lysis. In total, 200 µg of cell lysate protein was precleared with Protein A (cat. no. 73 778; Cell Signaling Technology, New England Biolabs, Whitby, ON, Canada) or Protein G (cat. no. 9006; Cell Signaling Technology) magnetic beads for 1 h. After preclearing, the cell lysates were incubated for 4 h with $\alpha$-Histone H3, $\alpha$-Tri-Methyl-Histone H3(K27), normal rabbit IgG (Cell Signaling Technology, cat. no. cs-4620, cs-973 and cs-2729, respectively) or normal mouse IgG (cat. no. C15400001; Diagenode). Samples were then incubated overnight with Protein A or Protein G magnetic beads. After overnight incubation, supernatants were denatured and run on SDS-PAGE. Ubiquitin, Histone H3 and Tri-Methyl-Histone H3(K27) protein expression levels were detected by immunoblotting as described above using the following antibodies: mouse mAb $\alpha$-Ubiquitin (P4D1) (Cell Signaling Technology; cat. no. 3936, New England Biolabs. Ltd.), mouse mAb $\alpha$-Histone H3 (96C10) (Cell Signaling Technology; cat. no. 3638, New England Biolabs. Ltd.), and $\alpha$-H3K27$^{me3}$ pAb-195 and pAb-069 (Diagenode, cat. no. C15410195 and C15410069, respectively) and rabbit mAb C36B11 (Cell Signaling Technology, cat. no. 9733, New England Biolabs. Ltd.). Conformation-specific $\alpha$-rabbit, Fc gamma-specific $\alpha$-mouse and light chain-specific $\alpha$-mouse (Cell Signaling Technology, cat. no. 5127, 96714 and 55802, respectively, all New England Biolabs. Ltd.) conjugated to HRP were used for ECL detection.

**Gastrocnemius skeletal muscle.** In total, 30–50 mg of control gastrocnemius muscle proteins were extracted as outlined above. Protein magnetic A beads (20 µL; cat.

**Table 1. List of Taqman probes used to measure RNA transcripts.**

| Assay ID | Target | Sequence |
| --- | --- | --- |
| Mm01545399_m1 | *Hprt* | GGACTGATTATGGACAGGACTGAAA |
| Mm01222421_m1 | *Kdr* | CATCCTAGAGCGCATGGCACCCATG |
| Mm00627185_m1 | *Notch1* | GCCGCAAGAGGCTTGAGATGCTCCC |
| Mm00437306_m1 | *Vegfa* | AGCAACATCACCATGCAGATCATGC |
| Mm01335418_m1 | *Thbs1* | GAAATACGAGTGTCGAGATTCCTAA |
| Mm00469580_m1 | *Hhip* | GCAGAATTGCCAAGTGTGAGCCAGC |

no. 73778, Cell Signaling, New England Biolabs. Ltd.) were pre-washed in 500 μL of RIPA and used to preclear 200 μL of lysate containing 600 μg of protein (20 min, at room temperature). Lysate was incubated overnight on rotation at 4°C with 5 μg of either H3K27$^{me3}$ or rabbit IgG (cat. no. C15410195 and C15410206, respectively, Diagenode, Denville, NJ, USA) antibodies. In total, 20 μL of pre-washed protein magnetic A beads were added to lysate and incubated at room temperature under rotation for 20 min. Using a magnetic rack, the immuno-complex was washed five times in 500 μL RIPA (4°C) and then resuspended in RIPA, denatured and analysed by SDS-page. In total, 20 μg of INPUT was used as a positive control, and $\alpha$-ubiquitin mouse and $\alpha$-H3 mouse were detected by immunoblotting. Conformation-specific $\alpha$-mouse was used for detection.

**Deubiquitylation assay.** A deubiquitinating enzyme cocktail (DUB) buffer was added to 40 μg of gastrocnemius protein extracted from control mouse. Briefly, DUB buffer was at pH of 7.5 and consisted of 50 mM Tris-HCl, 150 mM NaCl, 1 mM DTT, 0.5 mM EDTA, 0.25 mM PMSF, 1× protease inhibitor tablet and 1× phosphatase inhibitor tablet, and 100 ng of DUB cocktail (DB-0599-0025, Life Sensors, Malvern, USA). Muscle protein extract was incubated in DUB buffer with or without DUB cocktail at 37°C for 1 h. Samples were denatured and loaded onto SDS-PAGE and then immunoblotted for H3K27$^{me3}$ (mouse mAb C36B11, Cell Signaling Technology, cat. no. 3638, New England Biolabs Ltd.) and Ubiquitin (mouse mAb P4D1) (Cell Signaling Technology, cat. no. 3936, New England Biolabs Ltd.).

### In-gel tryptic digestion and mass spectrometry

H3K27$^{me3}$ protein pulldown was performed on C2C12 cell lysates or gastrocnemius protein extract using an $\alpha$-H3K27$^{me3}$ antibody (pAb195, Diagenode, cat. no. 15410195). After western blot electrophoresis, the gel was stained using Coomassie Blue. Visible 1 × 1 mm bands that migrated at 17 and 25 kDa were cut, and the In-Gel Tryptic Digestion kit (ThermoFisher Scientific,

cat. no. 89871) was used to digest and recover pure peptide fragments found at these molecular weights. The 17 kDa band was collected to verify H3K27$^{me3}$ was pulled down. The 25 kDa band was collected to identify unknown proteins and peptides that were pulled down with H3K27$^{me3}$. After digestion, the peptide fragments were analysed by mass spectrometry (LC-MS/MS, peptide mapping/purified sample) to identify the peptide sequences that have been isolated. MS was conducted by the ORU-YSci Core Mass Spectrometry Facility at York University.

### Chromatin immunoprecipitation

Approximately 30–50 mg of gastrocnemius or tibialis anterior muscles were homogenized in a mixer mill at 30 Hz (MM 400, Retsch GmbH Haan, Germany). Following homogenization and fixation (7–15 min, room temperature, 1% formaldehyde, Sigma-Aldrich, Cat. no. F8775), the sample was quenched (5 min, 125 mM glycine, BioShop, Cat. no. GLN001). The pellet was washed by resuspension in cold PBS (Gibco, Cat. no. 10010023) and centrifuged at 12,000 *g* for 4 min before being resuspended in ChIP Lysis Buffer (50 mM Tris, 10 mM EDTA, 1% SDS, pH 8.0), vortexed for 15 s and incubated on ice for 10–20 min. Following incubation, chromatin was sheared to achieve fragment sizes of 450–900 bp (gastrocnemius) and 200–700 bp (tibialis anterior) (Bioruptor Plus, Diagenode). Following shearing, an aliquot of chromatin was taken for DNA input and fragment size assessment. For immunoprecipitation, sheared chromatin was diluted 1:10 in ChIP Buffer (50 mM Tris, 167 mM NaCl, 1.1% Triton X-100, pH 8.0) and 6 μg of antibody for the histone target of interest was added overnight at 4°C on rotation: H3K27$^{me3}$ (Cat. no. C15410195), H3K4$^{me3}$ (Cat. no. 15410003) and H3K9$^{Ac}$ (Cat. no. C15410004, all Diagenode). The following day, 20 μL of protein-A/G-coated ChIP-grade beads were added to samples and incubated at 4°C on rotation for 4 h. Samples were then washed four times for 4 min with low salt wash buffer (50 mM Tris, 1 mM EDTA, 150 mM NaCl, 0.1% SDS, 1% Triton X-100, pH 8.0) followed by two washes for 5 min with high salt wash buffer (50 mM Tris,

**Table 2. List of Taqman probe used to detection histone mark enrichment following ChIP.**

| Target TSS | | Sequence | TSS location |
|---|---|---|---|
| *Kdr* | Forward primer | CATGACAAAACCCACCCAGAT | NC_000071.7 – Chr 5 |
| | Reverse Primer | TGGACCTAAGGATAGGCATTCTGT | c76374433–76373341 |
| | Probe | CCATGTGGATAAATC | |
| *Kdr'* | Forward primer | GGCCATGGACGTGGCTTA | |
| | Reverse Primer | GGAGATTGAGCTCAGGACATCAG | |
| | Probe | AAGTAAAGTGCTTCCTGCTGA | |
| *Notch1* | Forward primer | TGTCTATGCCCTGCCAAATTC | NC_000068.8 – Chr 2 |
| | Reverse Primer | CCTGTGAAGCTGTAGTCCAGGAT | c26359501–26358204 |
| | Probe | ACGGGCTACTGTGCC | |
| *Notch1'* | Forward primer | CCGCACTTGTGGCAGCTTA | |
| | Reverse Primer | CTACGTGGGCCCGAGATG | |
| | Probe | CCTCAACGGTGGTACAT | |
| *Notch1''* | Forward primer | CGCAGTCTCTACAGTGCTGGAA | |
| | Reverse Primer | CGAGTTGCACTGGCTGTCA | |
| | Probe | TATTTTAGCGACGGCCACT | |

1 mM EDTA, 500 mM NaCl, 0.1% SDS, 1% Triton X-100, pH 8.0). After washing, sample beads were resuspended in 150 µL of ChIP elution buffer (10 mM Tris, 5 mM EDTA, 300 mM NaCl, 0.5% SDS, pH 8.0) and incubated at 65°C for 45 min. The eluted supernatant was then collected and treated with 20 µg RNAse A (ThermoFisher Scientific, Cat. no. EN0531) for 30 min at 37°C, followed by treatment with Proteinase K at a final concentration of 0.6 mg/mL for 2 h at 65°C (ThermoFisher Scientific, Cat. no. EO0491). ChIP DNA was purified using a PureLink PCR Purification Kit according to the manufacturer's instructions (ThermoFisher Scientific, Cat. no. K310001). ChIP DNA was eluted in 50 µL of nuclease-free water (Ambion, Cat. no. 9932) and stored at −20°C until further qPCR analysis. Taqman probes used to detect enrichment in histone mark enrichment are listed in Table 2. The percentage of INPUT for ChIP-qPCR was calculated as follows (Solomon et al., 2021):

$$
\% \text{ INPUT} = 2^{\left(\left(Ct(INPUT) - Log2(Dilution\ Factor)\right) - Ct\left(Immunoprecipitated\right)\right)} * 100
$$

## Statistical analysis

For *in vivo* experiments, protein and mRNA analyses were performed on all animals ($n = 7$ per group) for most proteins and genes of interest. Some protein analyses had a lower number ($n = 5$) when samples with lower protein concentration were unavailable. For *in vitro* analyses, one representative experiment is shown with $n = 6$ biological replicates per condition; except when qualitative analyses were performed as in Figs 6 and 9*D* and *E*. The number of biological replicates is indicated in the figure legends. Most western blot analyses were performed using one technical replicate. When blotting issues were raised, a second technical replicate was performed for all samples. When two technical replicates were available the average of both values was used to express relative protein quantity. For qPCR measures, three technical replicates were performed. When the standard deviation for the Ct technical replicate was ≥0.3, three additional technical replicates were performed. All technical replicates (three or six) were used to perform the ΔΔCt analyses.

Statistical analyses were performed with Student's *t* test and one- and two-way ANOVAs using Prism 9.2 (GraphPad, San Diego, CA, USA). Outliers were identified using Grubb's method with an *α* value equal to 0.05. An F-test tested whether variances were different between groups. For one- and two-way ANOVAs, Tukey's multiple comparison and Bonferroni *post hoc* tests were used, respectively. Stand-alone comparison used Fisher's LSD test when an interaction was observed in two-way ANOVAs. Welch's correction was performed when variances were significantly different. To test correlation between the distribution of HPTMs on H3, we performed a Pearson's coefficient analysis and a linear regression. Squared *r* values ($r^2$) with a range from 0.8 to 1 were considered as strong, 0.5–0.79 as moderate and 0–0.49 as weak. When assessing enrichment in H3K9$^{Ac}$ activation mark, a paired non-parametric Wilcoxon tested the difference between the geometric means of enrichment on *Kdr* and *Notch1* transcription start site (TSS) regions. For all statistical analyses, a value of $P \leq 0.05$ was considered to be statistically significant.

## Results

### Nine weeks of endurance training elicited skeletal muscle angiogenesis

Nine weeks of endurance training led to significant metabolic and vascular adaptations in the skeletal muscle of female C57Bl6 mice. Indeed, endurance training increased the level of COX-IV and PECAM by 140% and 57% in the gastrocnemius muscle (Fig. 1*A*) and by 20% and 230% in the plantaris muscle (Fig. 1*B*). Nine weeks of endurance training also increased the capillary-to-fibre ratio in the extensor digitorum longus muscle (Fig. 1*C*). These data confirm that our endurance training protocol led to angiogenesis and some metabolic adaptations in the skeletal muscles recruited during treadmill running.

### Endurance training increased the protein levels of MDM2, EZH2 and NEDD4, three enzymes catalysing post-translational modifications of histone H3

We evaluated in the gastrocnemius muscle of female C57B6 mice whether endurance training will impact the expression of histone modifiers previously reported to regulate silencing HPTMs, including MDM2 and EZH2. We also tested whether endurance exercise will change the protein level of NEDD4, previously reported to ease re-activation of previously silenced genes through H3 ubiquitylation (Zhang et al., 2017).

First, we assessed whether training alters the expression of MDM2 and EZH2, as both enzymes could cooperate to establish post-transcriptional modifications on H2 and H3 (Wienken et al., 2016). We previously reported that endurance training increases MDM2 protein levels in the human vastus lateralis and the rodent plantaris muscles (Roudier et al., 2012, 2013). Here, we used the SMP14 antibody that has an epitope around the nuclear localization sequence (NLS) to detect MDM2 (Fig. 2*A*). The 2A10 antibody was used to detect whether the acidic domain becomes hypo-phosphorylated after training (Fig. 2*B*). Hypo-phosphorylation of the acidic domain stabilizes MDM2 and supports greater detection of MDM2 by the 2A10 antibody (Argentini et al., 2001; Meek & Hupp, 2010; Wang et al., 2012). Endurance training increased the detection of MDM2 by SMP14 by 42% in the gastrocnemius muscle ($P \leq 0.05$, trained *vs.* sedentary, Fig. 2*A*). 2A10 immunoblotting detected a 66% increase in MDM2 ($P \leq 0.05$, trained *vs.* sedentary, 2A10 clone, Fig. 2*B*). Endurance training upregulated MDM2 protein, possibly limiting its degradation.

As shown in Fig. 2*C*, endurance training significantly increased the expression of EZH2 protein (+60%, $P < 0.05$, trained *vs.* sedentary). Gastrocnemius muscle from trained mice had 33% more NEDD4 protein than muscle from sedentary animals ($P < 0.05$) (Fig. 2*D*).

### Endurance training increased the detection of an H3K27me3 band at 25 kDa in the gastrocnemius muscle

The increase in MDM2, EZH2 and NEDD4 proteins reported in Fig. 2 questions whether training could alter the abundance of HPTMs, impacting the global epigenetic pattern. MDM2 and EZH2 support the establishment of $H3K27^{me3}$ and $H2AK119^{Ub}$ marks that silence genes. NEDD4 is an E3 ubiquitin ligase for lysine residue 23 on histone H3 ($H3K23^{Ub}$), easing transcriptional re-activation of genes (Zhang et al., 2017). Promoters marked with $H3K4^{me3}$ are transcriptionally active (Wang et al., 2023). Therefore, we examined the global abundance of these HPTMs that have distinct impacts on transcription from gene activation to silencing: $H3K4^{me3}$ (activation), $H3K23^{Ub}$ (re-activation), $H2AK119^{Ub}$ (silencing) and $H3K27^{me3}$ (silencing).

Endurance training did not alter the global abundances of $H3K4^{me3}$ and $H3K23^{Ub}$ in the gastrocnemius muscle ($P > 0.05$, Fig. 3*A* and *B*). Similarly, the global abundance of $H2AK119^{Ub}$, was unchanged. The $H2AK119^{Ub}$ to H2A ratio remained comparable between sedentary and trained mice ($P > 0.05$, Fig. 3*C*). $H3K27^{me3}$ detected at its expected molecular weight (17 kDa) was not modified by 9 weeks of endurance training (Fig. 3*D*). Yet, we noted an increase in the detection of $H3K27^{me3}$ at 25 kDa. The abundance of this 25 kDa band normalized to H3 was increased by 81% in the gastrocnemius muscle of trained mice when compared to sedentary mice ($P = 0.0128$, Fig. 3*D*).

### Muscles with different metabolic phenotype presented differences in their level of expression of EZH2, MDm2 and H3K27me3, EZH2 and MDM2 at baseline and after endurance training

As shown in Fig. 1, endurance training increased the oxidative and capillarization in glycolytic dominant muscles that are rich in type IIB fibres (i.e. gastrocnemius and plantaris) (Augusto et al., 2004; Ballak et al., 2014). To further examine the physiological relevance of the band detected at 25 kDa, we first assessed whether the global abundance of $H3K27^{me3}$ could be a hallmark of a highly oxidative and vascularized muscle at baseline. Therefore, we measured the relative abundance of EZH2, MDM2 and $H3K27^{me3}$ in the oxidative soleus muscle, which is rich in oxidative fibres (type I and IIA) (Augusto et al., 2004; Ballak et al., 2014).

At baseline, the soleus had a reduced level of EZH2 and greater level of MDM2 when compared to the plantaris and gastrocnemius muscles (Fig. 4*A*). In the plantaris, endurance training increased the expression of MDM2 by 2.4 -fold without impacting EZH2 (Fig. 4*B*). In the soleus muscle, endurance training decreased the expression of

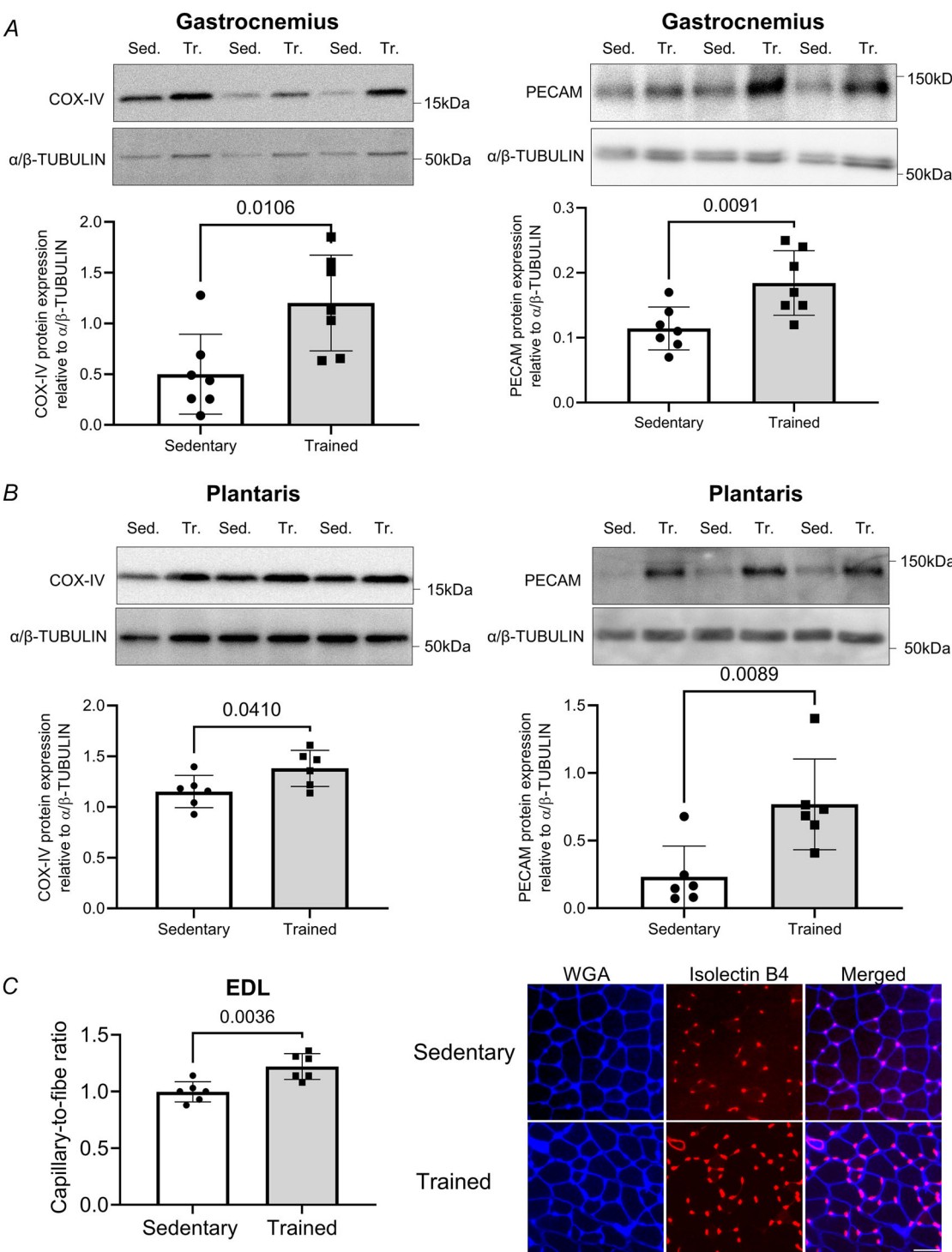

**Figure 1. Nine weeks of endurance training induces mouse skeletal muscle metabolic and vascular adaptations**
Female C57Bl6 mice were kept sedentary (Sed) or endurance trained (Tr) for 9 weeks (treadmill running, 5 days/week), and skeletal muscles were collected 72 h after the last bout of endurance exercise. *A* and *B*, representative immunoblots and densitometry analysis for COX-IV (left) and PECAM (right) protein expression levels in the (*A*) gastrocnemius and (*B*) plantaris muscles. $\alpha/\beta$-TUBULIN was used as a loading control. *C*, the extensor digitorum longus (EDL) muscle was stained for isolectin B4 and WGA. The capillary-to-fibre ratio was calculated. Representative images are shown for isolectin B4 and WGA as well as merged images. Scale bar = 50 μm. *A–C*, data show mean ± SD ($n = 6–7$). For *t* test reaching a $P \le 0.05$, exact *P* values are shown.

MDM2 by 2.8 -fold and had no impact of EZH2 (Fig. 4*C*). At baseline, high levels of H3K27$^{me3}$ at 17 and 25 kDa were observed in the oxidative soleus muscle when compared to the gastrocnemius and plantaris muscles (Fig. 5*A* and *B*). Similarly to what was observed in the gastrocnemius muscle (Fig. 3*D*), H3K27$^{me3}$ detected at 17 kDa remained unchanged in the plantaris and soleus muscles (Fig. 5*B* and *C*). The abundance of H3K27$^{me3}$ detected at 25 kDa increased by 87% in the plantaris muscle of trained mice when compared to muscles of sedentary mice (Fig. 5*B*). In the soleus muscle, training had no impact on the 25 kDa form of H3K27$^{me3}$ (Fig. 5*C*, $P = 0.2924$). Therefore, soleus had a greater abundance of H3K27$^{me3}$ at baseline; yet endurance training failed to further increase H3K27$^{me3}$ abundance in this oxidative muscle.

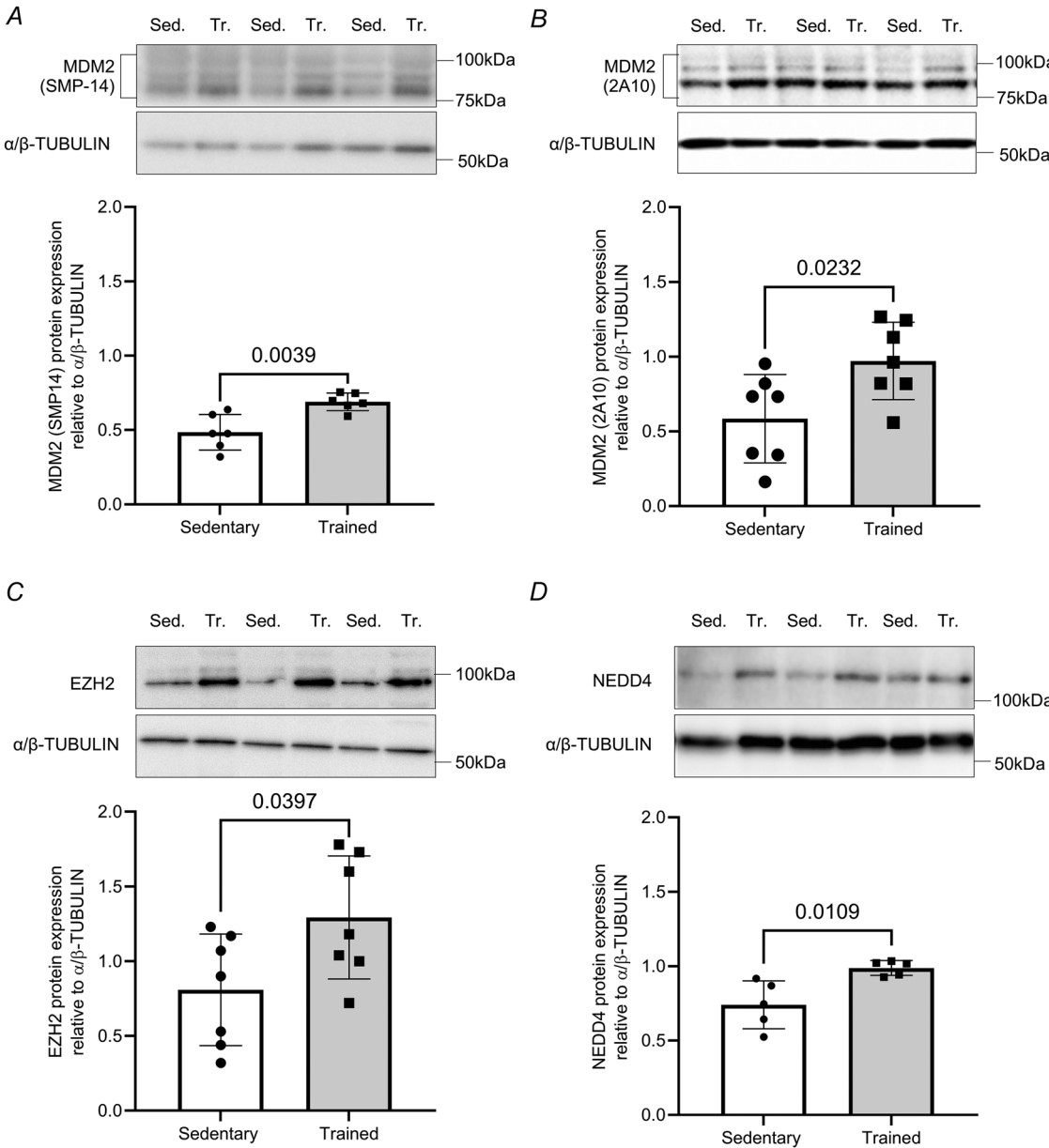

**Figure 2. Nine weeks of endurance training induces MDM2, EZH2 and NEDD4 protein expression in mouse gastrocnemius muscle**
Female C57Bl6 mice were kept sedentary (Sed) or endurance trained (Tr) for 9 weeks (treadmill running, 5 days/week), and skeletal muscles were collected 72 h after the last bout of endurance exercise. *A–D*, representative immunoblots and densitometry analysis for MDM2, measured using the (*A*) 2A10 and (*B*) SMP14 antibodies, (*C*) EZH2 and (*D*) NEDD4 protein expression levels in mouse gastrocnemius. $\alpha/\beta$-TUBULIN was used as a loading control. Data show mean ± SD (*n* = 7). For *t* test reaching $P \leq 0.05$, exact *P* values are shown.

We wanted to test whether the differential expression in EZH2 and MDM2 could promote differences in the levels of H3K27$^{me3}$ detected at 17 and 25 kDa. While the oxidative soleus muscle expressed high levels of MDM2 and H3K27$^{me3}$ (17 and 25 kDa) at baseline, mixed but glycolytic-dominant muscles (i.e. plantaris and gastrocnemius) gained greater levels of the E3 ubiquitin ligase MDM2 and greater detection of H3K27$^{me3}$ at 25 kDa after endurance training.

### The 25 kDa band detected by H3K27$^{me3}$ antibodies in the trained muscle contained both histone H3 and ubiquitin

To determine whether the shifted band was an artefact due to non-specific binding, we assessed whether different H3K27$^{me3}$ antibodies could detect the 25 kDa band in protein extracts from the skeletal muscle and in C2C12 myotubes (Fig. 6A). The 25 kDa band was detected in proteins extracted from the whole skeletal muscle (sedentary mice) and from resting C2C12 myotubes (non-electrostimulated) extracts using two rabbit polyclonal antibodies (pAb-195 and pAb-069) and a rabbit monoclonal antibody (C36B11). The band detected at 25 kDa was present in C2C12 myotubes and was difficult to detect in mSMECs, suggesting a degree of cell-type-specific expression of this form of histone modification (Fig. 6B).

To establish whether the 25 kDa band found more abundantly in the trained gastrocnemius muscle corresponds to a ubiquitinylated form of histone H3, we treated protein extracts from sedentary and trained

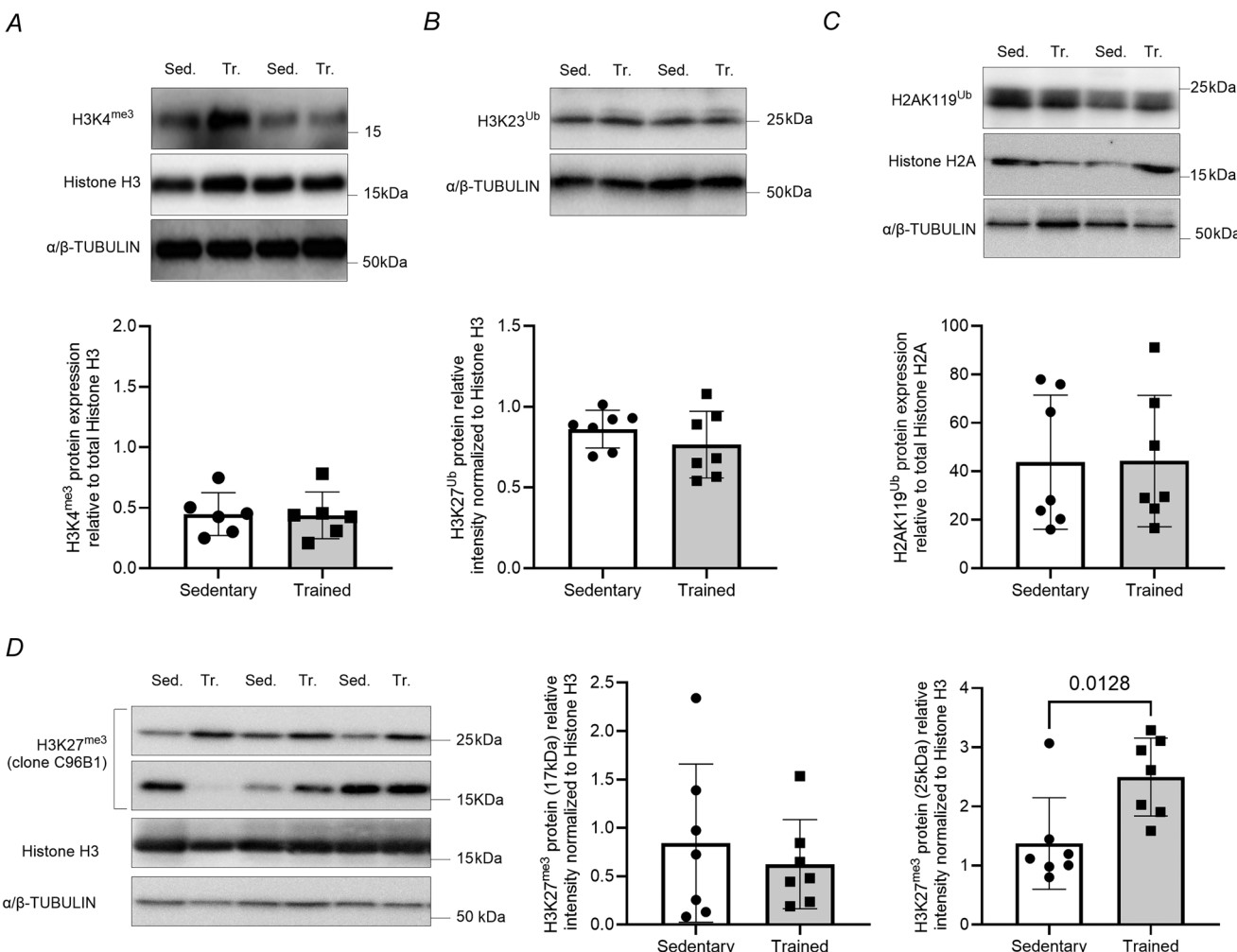

**Figure 3. Endurance training increases the abundance of an α-H3K27$^{me3}$ detected at 25 kDa in the gastrocnemius muscle**
*A–D*, representative immunoblots and densitometry analysis for (*A*) H3K4$^{me3}$, (*B*) H3K23$^{Ub}$, (*C*) H2AK119$^{Ub}$ and (*D*) H3K27$^{me3}$ protein expression detected at two different molecular weights, 17 kDa (left) and 25 kDa (right). Histone H2A, Histone H3 and α/β-TUBULIN were used as a loading control. Data show mean ± SD (*n* = 7). For *t* test reaching *P* ≤ 0.05, exact *P* values are shown.

gastrocnemius muscles with a broad range of DUBs (Fig. 6C). After treatment with DUBs, levels of H3K27$^{me3}$ detected at 25 kDa decreased in muscles from trained mice when compared to corresponding untreated trained

muscles (Fig. 6C). While untreated muscles from trained mice had greater level of the 25 kDa band detected by $\alpha$-H3K27$^{me3}$ when compared to muscles from sedentary mice, after DUB treatment similar levels were observed in

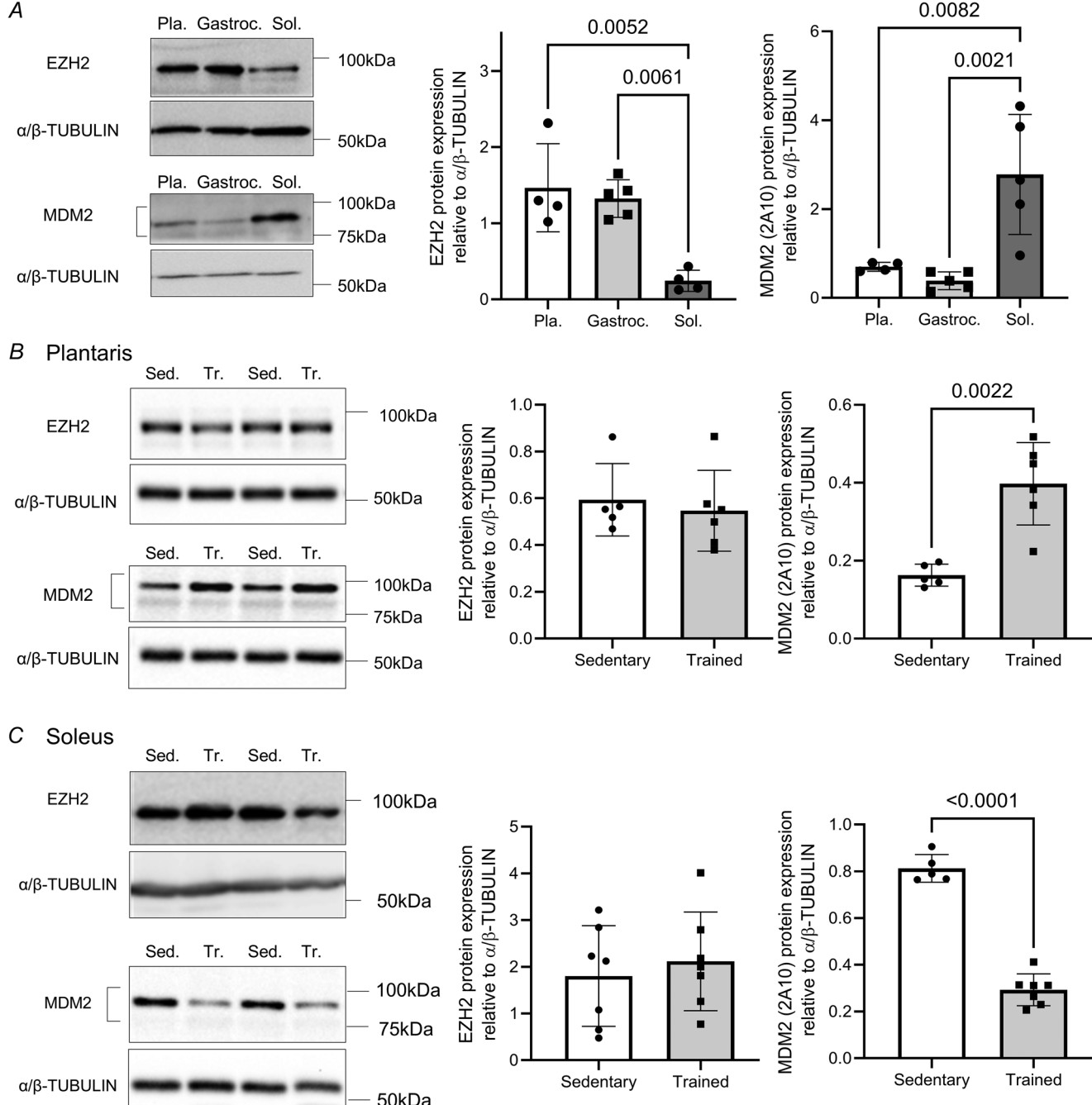

**Figure 4. The soleus muscle has low levels of EZH2 and high levels of MDM2 at baseline, and after endurance training MDM2 increases in the plantaris and decreases in the soleus muscle**
*A*, immunoblots and corresponding relative protein abundance show basal levels of EZH2 and MDM2 (2A10) in the plantaris (Pla.), gastrocnemius (Gastroc.) and soleus (Sol.) muscles (*n* = 5). Immunoblots shows the impact of endurance training on the expression of EZH2 and MDM2 (2A10) in the plantaris (*B*) and soleus (*C*) muscles. $\alpha/\beta$-TUBULIN was used as a loading control. Relative abundance of EZH2 or MDM2 is shown as mean ± SD (*n* = 5–7). For *t* test reaching *P* ≤ 0.05, exact *P* values are shown.

the sedentary and trained groups (Fig. 6*C*). The reduction in the abundance of the 25 kDa band detected by $\alpha$-H3k27$^{me3}$ supports the notion that H3 is ubiquitylated during endurance training.

To confirm that the 25 kDa band could be a ubiquitylated form of H3, we performed an immuno-precipitation assay using gastrocnemius muscle protein extract. Histone H3 was also detected at 25 kDa after immunoprecipitation with $\alpha$-H3K27$^{me3}$, though no band

was detected at 25 kDa using the control IgG (Fig. 6*D*). Ubiquitin was detected at 25 kDa, with intensity of the band being 43% greater using the $\alpha$-H3K27$^{me3}$ antibody than using the control IgG (Fig. 6*E*). After excision and trypsin digestion of the immunoprecipitation product, the shifted band at 25 kDa was analysed by MS. Peptides unique to histone H3 and ubiquitin were identified in the band collected at 25 kDa (Fig. 6*F*). Together, these data confirmed that the increased levels of H3K27$^{me3}$

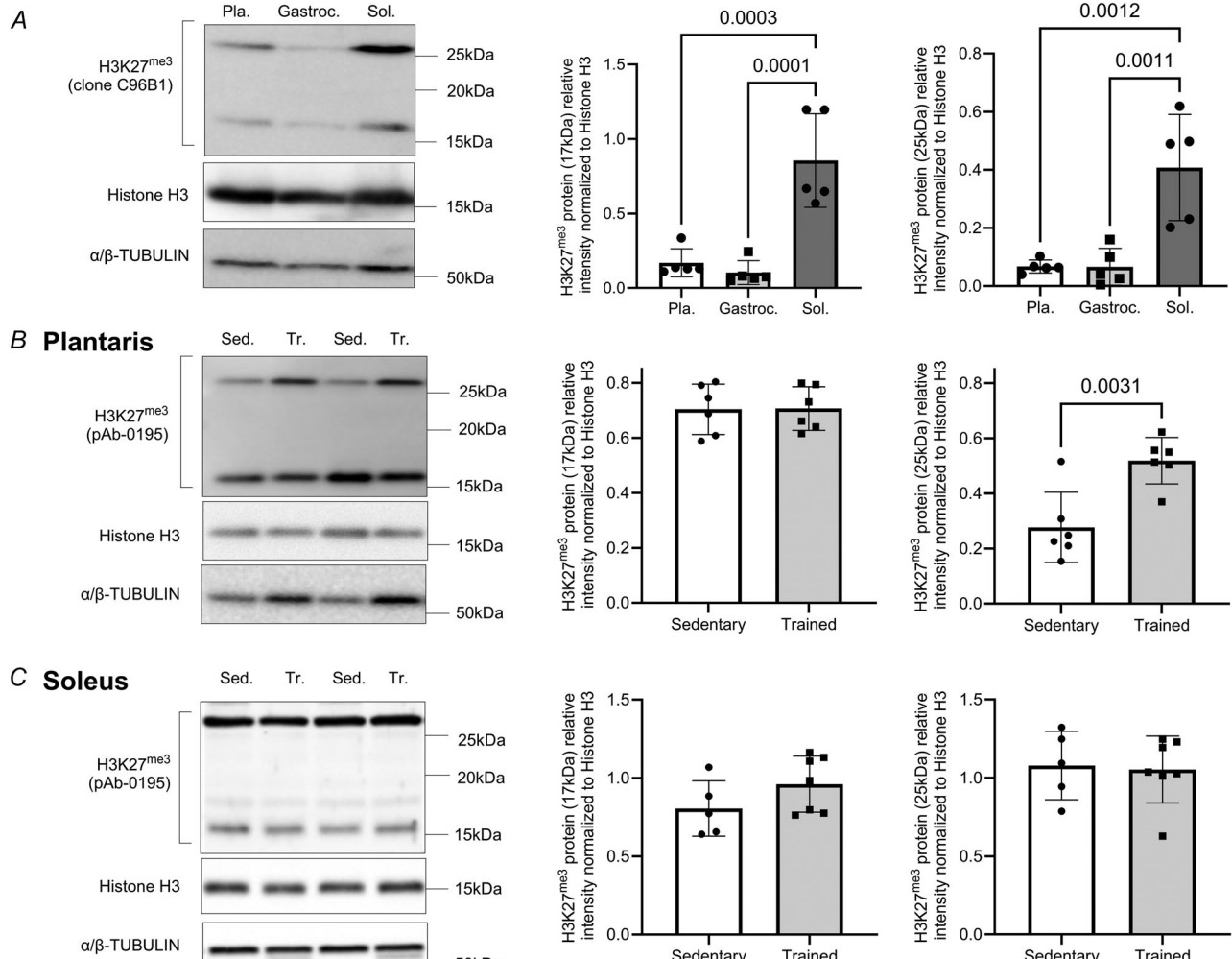

**Figure 5. The soleus muscle has higher levels of H3K27$^{me3}$ at baseline, but endurance training increases the abundance of the 25 kDa band in the plantaris muscle but not in the soleus muscle**
*A*, immunoblots show basal levels of H3K27$^{me3}$ in the plantaris (Pla.), gastrocnemius (Gastroc.) and soleus (Sol.) muscles. Histone H3 and $\alpha/\beta$-TUBULIN were used as a loading control. The relative abundance of H3K27$^{me3}$ detected at both 17 kDa (centre) and 25 kDa (right) is shown as mean $\pm$ SD ($n = 5$, one-way ANOVA overall effect of muscle type $P = 0.0001$ for 17 kDa and $P = 0.005$ for 25 kDa; for Dunnet *post hoc* test reaching $P \leq 0.05$, exact $P$ values are shown). *B*, immunoblots (left) show levels of H3K27$^{me3}$ measured using the polyclonal anti-body pAb-0195-05 (pAb-0195) in the plantaris muscle from sedentary (Sed) or trained (Tr) mice. Histone H3 and $\alpha/\beta$-TUBULIN were used as a loading control. The relative abundance of H3K27$^{me3}$ at 17 kDa (right) and at 25 kDa (centre) is shown as mean $\pm$ SD ($n = 5$, for *t* test reaching $P \leq 0.05$, exact $P$ values are shown). *C*, immunoblots (left) show levels of H3K27$^{me3}$ measured using the polyclonal antibody pAb-0195 in the soleus muscle from sedentary (Sed) or trained (Tr) mice. Histone H3 and $\alpha/\beta$-TUBULIN were used as a loading control. The relative abundance of H3K27$^{me3}$ at 17 kDa (right) and at 25 kDa (centre) is shown as mean $\pm$ SD ($n = 5$–6, *t* test retrieved no statistically significant differences).

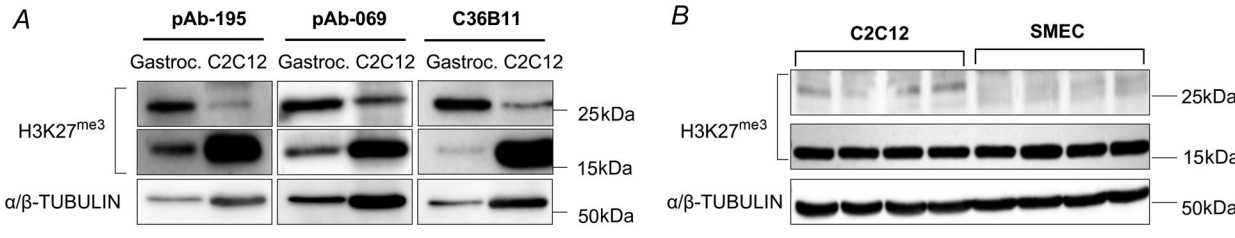

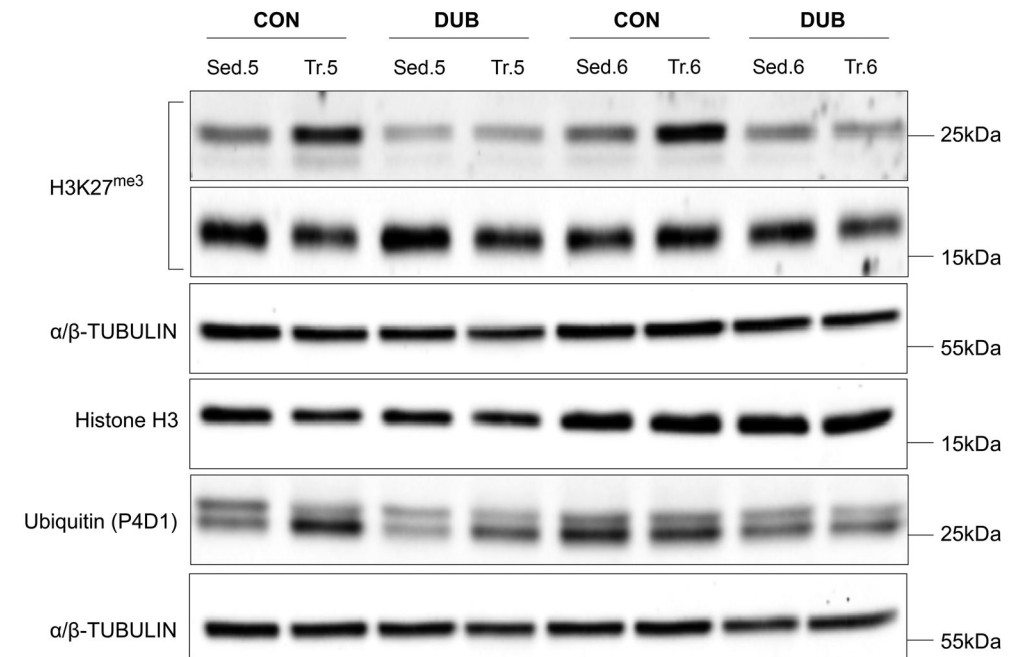

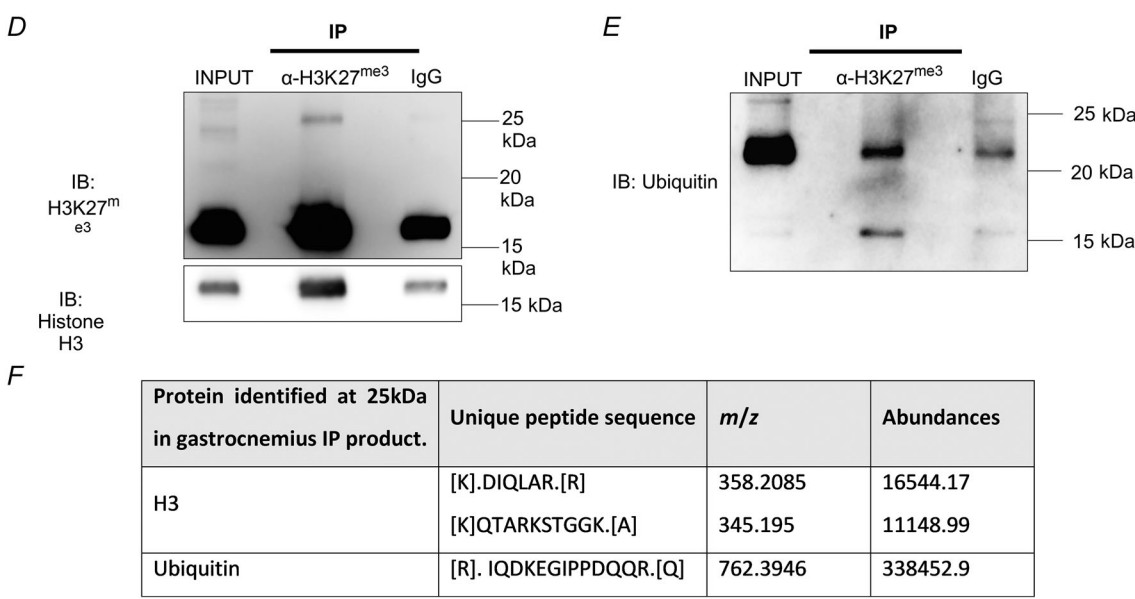

| Protein identified at 25kDa in gastrocnemius IP product. | Unique peptide sequence | *m/z* | Abundances |
|---|---|---|---|
| H3 | [K].DIQLAR.[R] | 358.2085 | 16544.17 |
| | [K]QTARKSTGGK.[A] | 345.195 | 11148.99 |
| Ubiquitin | [R]. IQDKEGIPPDQQR.[Q] | 762.3946 | 338452.9 |

**Figure 6. The increased abundance of H3K27$^{me3}$ detected at 25 kDa is partly explained by a ubiquitylation of H3 in the trained gastrocnemius muscle**

*A*, H3K27$^{me3}$ was detected at 25 kDa in whole gastrocnemius muscles and C2C12 myotubes by three $\alpha$-H3K27$^{me3}$ antibodies (polyclonal pAb-195, polyclonal pAb-069 and monoclonal C36B11). *B*, $\alpha$-H3K27$^{me3}$ (C36B11) detected

a band at 25 kDa in C2C12 myotubes that was not present in mouse skeletal muscle endothelial cells (mSMEC, *n* = 4 per cell type). *C*, protein extracts from sedentary (Sed.) and trained (Tr.) gastrocnemius muscles (*n* = 2 per group) were treated with a cocktail of broad-range deubiquitinating enzymes. Western blotting detected levels of ubiquitin (P4D1) and H3K27$^{me3}$ (C36B11). *D–F*, after immunoprecipitation of H3K27$^{me3}$ using pAb-195 in protein extract from gastrocnemius muscle, immunoblots detected histone H3 (*D*) and ubiquitin (*E*). *F*, MS analysis of the trypsin-digested 25 kDa band; table shows mass-to-charge ratio (*m/z*) and abundances (grouped) of unique peptides identified.

detected at 25 kDa in response to endurance training in gastrocnemius muscle could be due to an increased ubiquitylation of histone H3.

### Increased abundance of H3K27$^{me3}$ on TSS regions of *Notch1* and *Kdr* was associated with a decreased basal level of *Kdr* mRNA but increased level of *Notch1* after 9 weeks of training in the gastrocnemius muscle

We aimed to investigate whether changes in H3K27$^{me3}$ could explain changes in basal levels of gene expression after endurance training in glycolytic-dominant muscles rich in type IIB fibres (e.g. gastrocnemius, plantaris and tibialis anterior) (Augusto et al., 2004). First, we analysed the expression of angiogenesis-related genes in the gastrocnemius muscle: *Vegfa* and *Thbs1*, which serve as key indicators of the skeletal muscle angiogenic balance (Olfert & Birot, 2011); as well as *Kdr*, *Notch1* and *Hhip*, important for angiogenesis and the establishment of a slow twitch phenotype (Blanco & Gerhardt, 2013; Hasan et al., 2017; Hoier et al., 2020; Ochi et al., 2006; Olsen et al., 2020; Yamashita et al., 2025; Zhao et al., 2018). Interestingly, *Kdr*, *Notch1* and *Hhip* were previously reported to be differentially regulated by *Mdm2* knock-down potentially due to an altered distribution of H3K27$^{me3}$ in mouse embryonic fibroblasts (MEFs) (Wienken et al., 2016). After 9 weeks of training (Fig. 7*A*), basal levels of *Vegfa* increased by 57% while *Thbs1* expression remained similar between the sedentary and trained group (1.2 ± 0.7 *vs.* 0.9 ± 0.6, respectively). Training increased *Notch1* mRNAs by 52% and decreased the expression of *Kdr* by 44%. *Hhip* mRNA tended to be lower in the trained group but remained non-significant (*P* = 0.077).

Next, we analysed whether endurance training alters the abundance of H3K27$^{me3}$ marks at the TSS of *Kdr* and *Notch1*. ChIP using the α-H3K27$^{me3}$ antibody (pAb-0195) showed that endurance training led to an enrichment in H3K27$^{me3}$ marks on the TSS regions of *Kdr* and *Notch1* (Fig. 7*B*).

Ubiquitylation of H3 could be associated with gene transcriptional re-activation in tumour cells by promoting acetylation of H3 on residue lysine 9 (H3K9$^{Ac}$) (Zhang et al., 2017). H3K4$^{me3}$ and H3K9$^{Ac}$ are hallmarks of transcriptionally active genes in muscle cells and tissues (Cui et al., 2017; Kawano, 2021; Kawano et al., 2015; McGee et al., 2009; Wang et al., 2023). To explain why H3K27$^{me3}$

enrichment fails to restrain *Notch1* mRNA expression, we tested whether differential enrichments in these HPTMs were observed on *Kdr* and *Notch1* regulatory sequences.

When comparing enrichment in H3K27$^{me3}$ and H3K4$^{me3}$ across the TSS regions of *Kdr* and *Notch1*, we observed that *Notch1* TSS had a greater level of H3K27$^{me3}$ and H3K4$^{me3}$ marks than *Kdr* TSS (+51% and 33%, *P* = 0.005 and *P* = 0.028, Fig 7*C*). The *Notch1* promoter showed a positive and strong correlation between H3K27$^{me3}$ and H3K4$^{me3}$ (Fig. 7*D*). The correlation between H3K27$^{me3}$ and H3K4$^{me3}$ on the *Kdr* TSS region was weak (Fig 7*E*). Next, we tested whether *Notch1* and *Kdr* had differences in H3K9$^{Ac}$ enrichment. A gene effect was observed where *Notch1* had greater levels of H3K9$^{Ac}$ than *Kdr* TSS in another glycolytic and type IIB-dominant muscle (i.e. tibialis anterior, two-way ANOVA, *n* = 4). Training did not impact the enrichment in H3K9$^{Ac}$ of both *Kdr* and *Notch1* promoter (two-way ANOVA, Fig. 7*F*). Interestingly, post-training *Notch1* enrichment in H3K9$^{Ac}$ tended to be greater than that of *Kdr* (+78%, *t* test, *P* = 0.059, *n* = 4, Fig. 7*G*).

### In a glycolytic dominant type IIB muscle, the greater abundance in H3K27$^{me3}$ might support reactivation of *Notch1*, a gene where H3K27$^{me3}$ enrichment showed positive correlation H3 activation marks

We reported in Figs 4*C* and 5*C* that the oxidative soleus muscle responded differently to training when considering the expression of MDM2 (decreased) and H3K27$^{me3}$ detected at 25 kDa (unchanged). Interestingly, contrary to what was observed in the gastrocnemius muscle, *Kdr* mRNA did not change but *Notch1* expression decreased by 25% in the soleus muscle of trained mice when compared to sedentary mice (Fig. 8). This result led us to question whether the ubiquitylation of H3 detected at 25 kDa by α-H3K27$^{me3}$ and MDM2 are part of the mechanisms that upregulate *Notch1* expression in response to repeated exposure to contractile activity.

### Differentiation and repeated exposures to contractile activity increased the abundance of H3K27$^{me3}$ detected at 25 kDa in C2C12 myotubes

To further explore the underlying mechanisms, we tested *in-vitro* whether contractile activity alters the expression of H3K27$^{me3}$ in C2C12 cells. Based on the observation

that skeletal muscles have greater levels of H3K27$^{me3}$ than C2C12 cells, we questioned whether differentiation could impact the abundance of the shifted band of H3K27$^{me3}$. C2C12 myoblasts had significantly less of the 25 kDa band than C2C12 myotubes (Fig. 9*A*). As C2C12 myotubes were differentiated, the putative ubiquityl-H3 detected by $\alpha$-H3K27$^{me3}$ (C36B11) became more abundant, increasing 2.4- and 4.0-fold after 4 and 7 days of differentiation when compared to day 0 of differentiation (one-way ANOVA, $P < 0.0001$).

We tested whether the shift from 17 to 25 kDa could trigger proteasomal degradation of H3K27$^{me3}$ when differentiation is in progress (i.e. day 4 of differentiation). MG132 did not lead to an accumulation of the 25 kDa band (Fig. 9*B*), suggesting that the putative ubiquityl-H3 detected by $\alpha$-H3K27$^{me3}$ is not a proteolytic mark.

Next, we used EPS as an exercise-like stimulus in cultured C2C12 myotubes. EPS was repeated over 4 days to mimic repeating sessions (90 min daily) of sub-maximal exercise (Connor et al., 2001; Tamura et al., 2020*b*; Vepkhvadze et al., 2021). After 4 days of recurrent

exposure to EPS (Fig. 9*C*), we observed a significant increase in the expression of H3K27$^{me3}$ detected at both 17 kDa (CON *vs*. EPS, 1.00 $\pm$ 0.02 *vs*. 1.12 $\pm$ 0.04, $P = 0.013$, $n = 6$) and 25 kDa (CON *vs*. EPS, 1.00 $\pm$ 0.12 *vs*. 1.57 $\pm$ 0.21, $P = 0.037$, $n = 6$). These findings demonstrate that contractile activity increased the global expression of H3K27$^{me3}$ marks at 17 kDa and the form detected at 25 kDa in myotubes *in vitro*.

The observation that repeated exposure to EPS increased the abundance of the 25 kDa band in the C2C12 myotubes led us to question whether the effect of contractile activity was due to histone H3 ubiquitylation. After immunoprecipitating H3K27$^{me3}$, we observed greater levels of ubiquitin detected at 25 kDa and H3K27$^{me3}$ detected at 17 kDa in the C2C12 myotubes exposed to repeated EPS for 4 days compared to resting C2C12 myotubes (Fig. 9*D*). After immunoprecipitation of histone H3 (Fig. 9*E*), EPS-treated C2C12 myotubes showed greater levels of ubiquitin at 25 kDa and higher levels of H3K27$^{me3}$ at 17 and 25 kDa. These data indicate that repeated exposure to contractile activity increased the

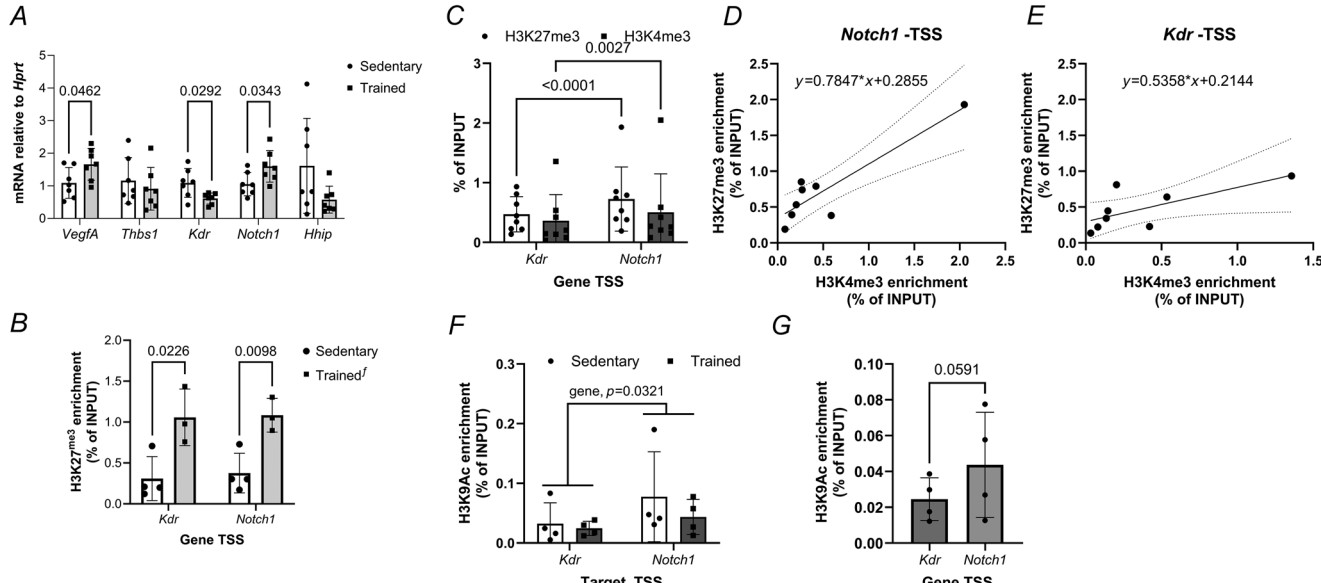

**Figure 7. Endurance training increases the abundance of H3K27$^{me3}$ marks on TSS of *Kdr* and *Notch1*, decreases *Kdr* mRNA but upregulates *Notch1* expression in glycolytic-dominant muscles**
*A*, qPCR data show basal level of mRNAs for *Vegfa*, *Thbs1*, *Kdr*, *Notch1* and *Hhip* in the gastrocnemius muscle of sedentary mice or trained mice (mean $\pm$ SEM, $n = 7$, *$P < 0.05$, t test). *B*, ChIP-qPCR assay was performed on gastrocnemius muscle using the $\alpha$-H3K27$^{me3}$ (pAb195). Data show enrichment in H3K27$^{me3}$ on *Kdr* and *Notch1* TSS regions (percentage of INPUT, $n = 3–4$, f overall impact of training *$P < 0.05$). *C*, comparison of H3K27$^{me3}$ and H3K4$^{me3}$ enrichment on *Kdr* and *Notch1* TSS regions measured by ChIP-qPCR on gastrocnemius muscle. Data show average enrichment on two or three regions of the TSS for *Kdr* and *Notch1*, respectively (Taqman probes used were KDR and KDR′ for *Kdr*, and NOTCH1, NOTCH1′ and NOTCH1″ for *Notch1*). Data are expressed as percentahe of INPUT (mean $\pm$ SD, $n = 5$, Fisher's test, *$P < 0.05$ and **$P < 0.01$). *D* and *E*, relationship between H3K27$^{me3}$ and H3K4$^{me3}$ enrichment on *Notch1* (*D*) and *Kdr* (*E*) TSS regions was tested using a linear regression ($n = 8$, Pearson, for *Notch1*: $r^2 = 0.8173$, slope different from 0 with $P = 0.0021$; for *Kdr*: $r^2 = 0.4490$, slope tended to be different from 0 with $P = 0.0502$). *F*, ChIP-qPCR shows enrichment in H3K9$^{Ac}$ on *Kdr* and *Notch1* TSS regions performed on tibialis anterior muscle. Data are mean $\pm$ SD; two-way ANOVA tested the effects of the genes TSS and training status ($P = 0.0321$ for the gene TSS effect). *G*, paired non-parametric Wilcoxon test for the difference in the abundance of H3K9$^{Ac}$ on *Kdr* and *Notch1* TSS region.

abundance of a ubiquitylated form of histone H3 in C2C12 myotubes.

## Under inhibition of EZH2 by GSK343 EPS could not upregulate ubiquitylated-H3 detected by $\alpha$-H3K27$^{me}$ antibody

Since the distribution of epigenetic marks is highly cell type-dependent, we further examined the underlying mechanisms responsible for the formation of the ubiquitylated form of H3 in C2C12 myotubes. With this in mind, we assessed whether EZH2 is required for the establishment of the ubiquitylation detected at 25 kDa. To do so, C2C12 myotubes were incubated with GSK343, a chemical inhibitor of EZH2 (1 and 5 μM), for 48 h and the resultant effect on H3K27$^{me3}$ expression was evaluated at both molecular weights.

GSK343 treatment (1 and 5 μM) significantly impacted the levels of H3K27$^{me3}$ protein level expressed at 17 kDa (Fig. 10*A*). GSK343 treatment reduced H3K27$^{me3}$ protein levels in C2C12 myotubes by 25% (1 μM) and 24% (5 μM) (untreated *vs.* 1 μM GSK343, $P \leq 0.05$, and untreated *vs.* 5 μM GSK343, $P \leq 0.05$, $n = 6$). This demonstrates that EZH2 is critical for maintaining the H3K27$^{me3}$ mark at 17 kDa. However, there was not a statistically significant difference in the expression of the ubiquitylated form detected by $\alpha$-H3K27$^{me3}$ at 25 kDa between untreated and GSK343-treated myotubes ($F = 0.872$, $P = 0.44$, $n = 6$, Fig. 10*A*). This suggests that EZH2 is not crucial for maintaining basal expression of the ubiquitylated mark.

We had observed that the ubiquitylated form of the H3 mark is responsive to exercise and EPS, an exercise-like stimulus (Figs 3*D* and 4*C*). Therefore, we next assessed whether EZH2 is required for mediating the exercise-induced increase in the ubiquitylated mark. Differentiated C2C12 myotubes were simultaneously treated with 4 days of EPS and GSK343 (1 μM) treatment.

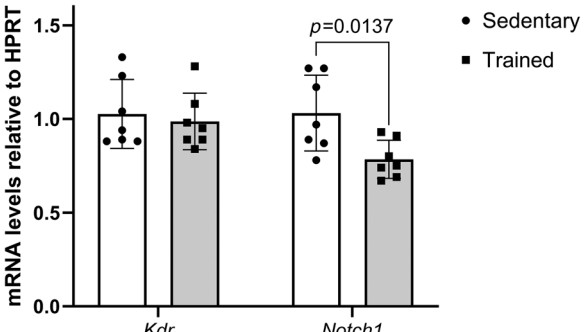

**Figure 8. In the oxidative soleus muscle, endurance training decreases *Notch1* mRNA but does not impact *Kdr* expression**
qPCR analyses of *Kdr* and *Notch 1* mRNAs relative to the housekeeping gene HPRT. Data show mean ± SD ($n = 7$; for *t* test reaching a $P \leq 0.05$ trained *vs.* sedentary, exact *P* values are shown).

A two-way ANOVA showed that both EPS ($P < 0.01$) and GSK343 treatment ($P \leq 0.05$) had a significant effect on the ubiquitylated form of H3 detected using $\alpha$-H3K27$^{me3}$. *Post hoc* analysis with Bonferonni correction for multiple comparisons found that, in the absence of EZH2 inhibition, EPS increased by 46% the expression of the ubiquitylated form detected at 25 kDa using $\alpha$-H3K27$^{me3}$ (DMSO: CON *vs.* EPS, $1.14 \pm 0.06$ *vs.* $1.66 \pm 0.11$). However, this stimulatory effect was attenuated when EZH2 was inhibited (with 1 μM GSK343: CON *vs.* EPS, $1.13 \pm 0.11$ *vs.* $1.28 \pm 0.07$, $P = 0.43$, $n = 6$) (Fig. 10*B*).

We next examined the effect of EZH2 inhibition (GSK343 treatment, 1 μM, 48 h) on the expression of *Notch1*, a target gene for H3K27$^{me3}$ marking. We observed a significant increase in *Notch1* mRNA expression in response to GSK343 treatment alone ($1.01 \pm 0.04$ *vs.* $1.90 \pm 0.11$, DMSO *vs.* GSK343 treatment, Fig. 10*C*), indicating that EZH2 activity repressed *Notch1* expression under resting conditions.

To assess whether E2H2 activity is required to promote changes in *Notch1* expression in response to repeated contractile activity *in vitro*, C2C12 myotubes were treated with 4 days of EPS (90 min daily) and GSK343 (1 μM) (Fig. 10*D*). A two-way ANOVA showed an important interaction ($P = 0.0048$) between the effect of the contractile activity (i.e. EPS) and EZH2 inhibition (i.e. GSK343) on *Notch1* mRNA levels. Bonferroni correction for multiple comparisons found that EPS increased *Notch1* expression in C2C12 myotubes (DMSO: CON *vs.* EPS, $1.02 \pm 0.20$ *vs.* $3.04 \pm 0.77$). Yet, in the presence of GSK343, EPS did not alter the level of expression of *Notch1* (GSK343: CON *vs.* EPS, $1.62 \pm 0.53$ *vs.* $1.86 \pm 0.39$, $P > 0.99$). Therefore, EZH2 activity was required to regulate *Notch1* expression in response to EPS.

## Inhibition of MDM2 activity decreases the relative abundance of ubiquitylated-H3 detected by $\alpha$-H3K27$^{me3}$ antibody

Expression of the ubiquitylated-H3 band detected at 25 kDa increased as myotube differentiation progressed (Fig. 9*A*). Interestingly, the relative protein expression of MDM2 also increased as myotube differentiation progressed (Fig. 11*A*). After 4 and 5 days of differentiation, MDM2 protein levels were increased 2.1- and 6.1-fold, respectively, compared to undifferentiated controls (UD CON *vs.* 4D Differentiation *vs.* 5D Differentiation, $0.432 \pm 0.061$ *vs.* $1.338 \pm 0.151$ *vs.* $3.05 \pm 0.214$, $P < 0.0001$, $n = 6$). Together, this leads us to postulate a potential role for MDM2 in regulating the ubiquitylation of the histone mark, H3K27$^{me3}$.

To assess whether MDM2 regulates this ubiquitylation, we treated differentiated C2C12 myotubes with

Serdemetan (1 μM), a chemical inhibitor of MDM2, for 24 or 48 h (Fig. 11*B–D*). Serdemetan binds to the RING domain of MDM2 to restrain its capacity to perform its canonical E3 ligase activities (Chargari et al., 2011).

Serdemetan treatment increased MDM2 expression in C2C12 myotubes (Fig. 11*B*, *f* effect of treatment, *P* = 0.0133). We observed that MDM2 inhibition significantly decreased the expression of the ubiquitylated

form detected using an $\alpha$-H3K27$^{me3}$ after both 24 and 48 h (24 h: CON *vs.* Serdemetan, 0.069 ± 0.008 *vs.* 0.040 ± 0.005, values are means ± SEM, *P* = 0.0152, Mann–Whitney *U* test, *n* = 6; 48 h: CON *vs.* Serdemetan, 0.070 ± 0.004 *vs.* 0.044 ± 0.005, values are means ± SEM, *P* < 0.01, Mann–Whitney *U* test, *n* = 6) (Fig. 11*C* and *D*). We next examined the effect of MDM2 inhibition (Serdemetan, 1 μM, 24–48 h) on the expression of *Notch1*.

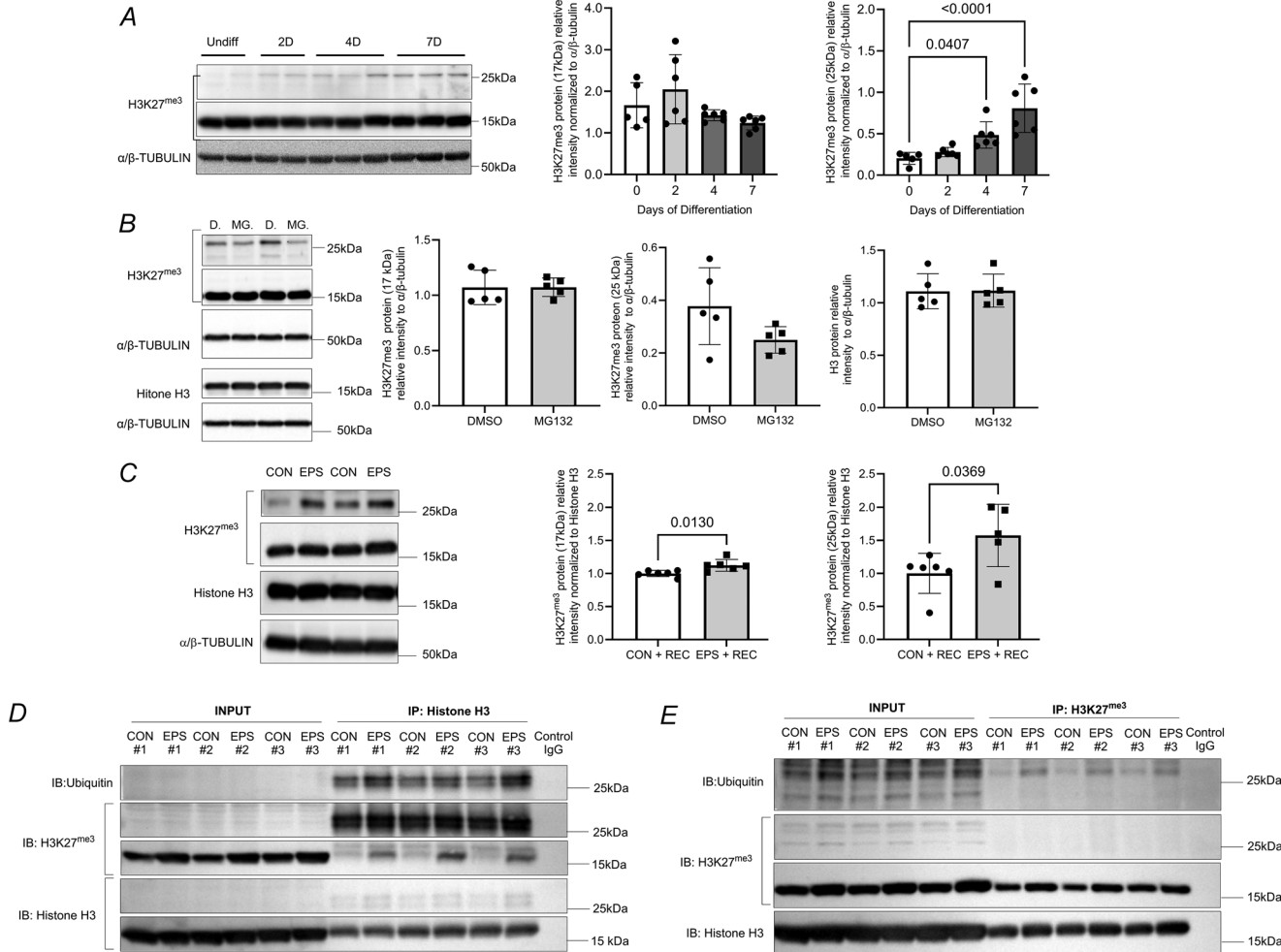

**Figure 9. Repeated exposure to contractile activity increases the expression of the putative ubiquitylated form of histone H3 detected by $\alpha$-H3K27$^{me3}$ in C2C12 myotubes**

*A*, C2C12 myoblasts were differentiated into myotubes over a period of 7 days. Immunoblots show level of H3K27$^{me3}$ (C36B11). $\alpha$/$\beta$-TUBULIN was used as a loading control. The relative abundance of H3K27$^{me3}$ at 17 kDa (right) and at 25 kDa (centre) is shown as mean ± SD (*n* = 5, one-way ANOVA, overall effect of differentiation *P* < 0.0001) For *post hoc* Dunnet test reaching *P* ≤ 0.05, exact *P* values are shown. *B*, after 4 days of differentiation, C2C12 myotubes were treated with MG132 (MG.) or DMSO (D.) as a control vehicle (7 h, 10 μM). Immunoblots show level of H3K27$^{me3}$ (pAb-195) and histone H3. $\alpha$/$\beta$-TUBULIN was used as a loading control. The relative abundance of H3K27$^{me3}$ at 17 kDa (right) and at 25 kDa (centre) is shown as mean ± SD (*n* = 5–6). *C*, representative immunoblots for H3K27$^{me3}$ in C2C12 myotubes following 4 days of repeated EPS (90 min/day). Repeated contractile activity in C2C12 myotubes increased the relative abundance of the H3K27$^{me3}$ mark detected at both 17 and 25 kDa. Histone H3 and $\alpha$/$\beta$-TUBULIN were used as a loading control. Data show mean ± SD (*n* = 6). For *t* test reaching *P* ≤ 0.05, exact *P* values are shown. *D*, H3K27$^{me3}$ was pulled down in control (CON) and EPS-stimulated C2C12 myotubes (EPS, 90 min/day). Immunoblots were performed for ubiquitin, histone H3 and H3K27$^{me3}$. *E*, H3 was pulled down in CON and EPS-stimulated C2C12 myotubes. Immunoblots were performed for ubiquitin, H3K27$^{me3}$ and histone H3.

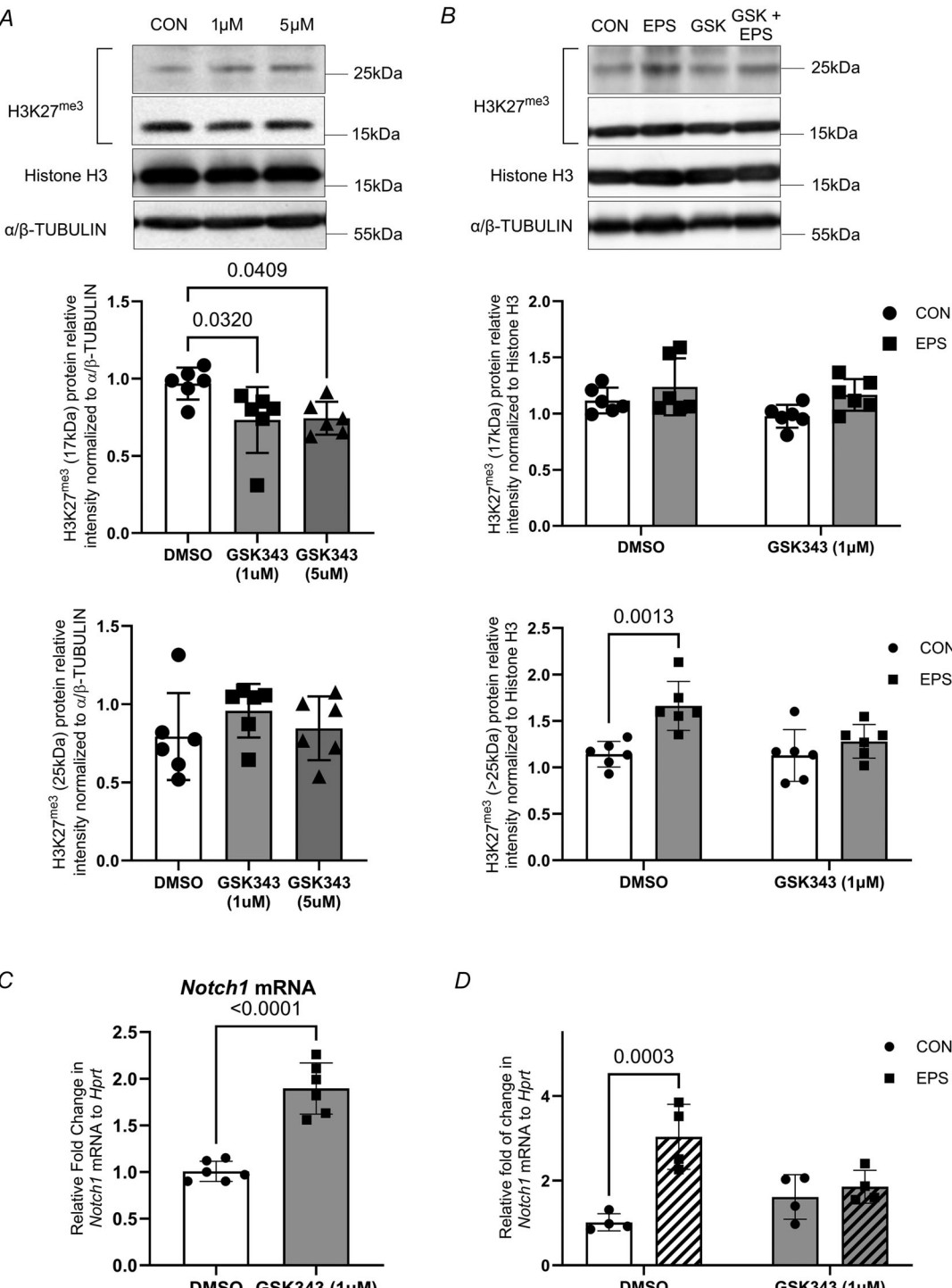

**Figure 10. Inhibition of EZH2 by GSK343 decreases the abundance of the H3K27^me3 mark detected at 17 kDa but not at 25 kDa**

*A*, C2C12 myotubes were incubated with 1 and 5 μM GSK343, an EZH2 inhibitor, for 48 h. Representative immunoblots and densitometry analysis for H3K27^me3 at 17 kDa (middle) and 25 kDa (bottom). *B*, myotubes were treated with 1 μM GSK343 and subjected to EPS for 4 days (90 min/day). Representative immunoblots and densitometry analyses for H3K27^me3 at 17 kDa (middle) and 25 kDa (bottom). *C*, EZH2 inhibition (1 μM, 48 h) increases the expression of *Notch1* mRNA expression. *A–C*, data show mean ± SD ($n = 6$, for *t* test and *post hoc* test reaching $P$ values ≤0.05, exact $P$ values are shown). *D*, C2C12 myotubes were treated with 1 μM GSK343 (48 h) and subjected to EPS for 4 days (90 min/day). In the presence of GSK343, EPS does not increase the levels of *Notch1* mRNA [$n = 4$, for *post hoc* test *vs.* corresponding control (CON or DMSO) reaching $P ≤ 0.05$, exact $P$ values are shown].

In the presence of Serdemetan, *Notch1* mRNA expression was reduced by 26% in C2C12 myotubes (CON *vs.* 48 h Sedermetan, $1.015 \pm 0.078$ *vs.* $0.7533 \pm 0.100$, $P = 0.048$, $n = 6$) (Fig. 11*E*).

To determine whether MDM2 activity was required to upregulate *Notch1* mRNA in response to contractile activity, differentiated C2C12 myotubes pre-treated with Serdemetan (1 µM) for 48 h were exposed to repeated EPS for 4 days (Fig. 11*F–I*). Two-way ANOVA indicated an overall negative effect of Serdemetan on *Notch1* expression (*f* with $P = 0.0001$, Fig. 11*F*) and an overall positive effect of EPS (* with $P = 0.0135$). EPS increased *Notch1* mRNA by 85%. Yet, in the presence of Serdemetan, Notch1 expression remained unchanged after EPS ($P = 0.5759$). Compellingly, while Serdemetan had no impact on the

protein level of H3K27$^{me3}$ detected at 17 kDa (Fig. 11*H*), it had an extremely significant effect on the level of H3K27$^{me3}$ detected at 25 kDa (Fig. 11*I*, $P = 0.0002$, −21%). Overall EPS had a positive effect on the band detected at 25 kDa ($P = 0.0235$, +13%). The putative ubiquityl form of H3 increased by 21.5% after EPS (EPS *vs.* CON, both DMSO pre-treated). In the presence of Serdemetan, EPS had no impact on the levels of H3K27$^{me3}$ detected at 25 kDa (Fig. 11*I*). Based on these observations, MDM2 activity is necessary to increase the abundance of ubiquityl-H3 and to prevent the upregulation of *Notch1* mRNA in response to the contractile activity.

Since differentiation led to an increase in the ubiquitylated form of H3, we tested whether EZH2 and MDM2 inhibitions had an impact on the differentiation

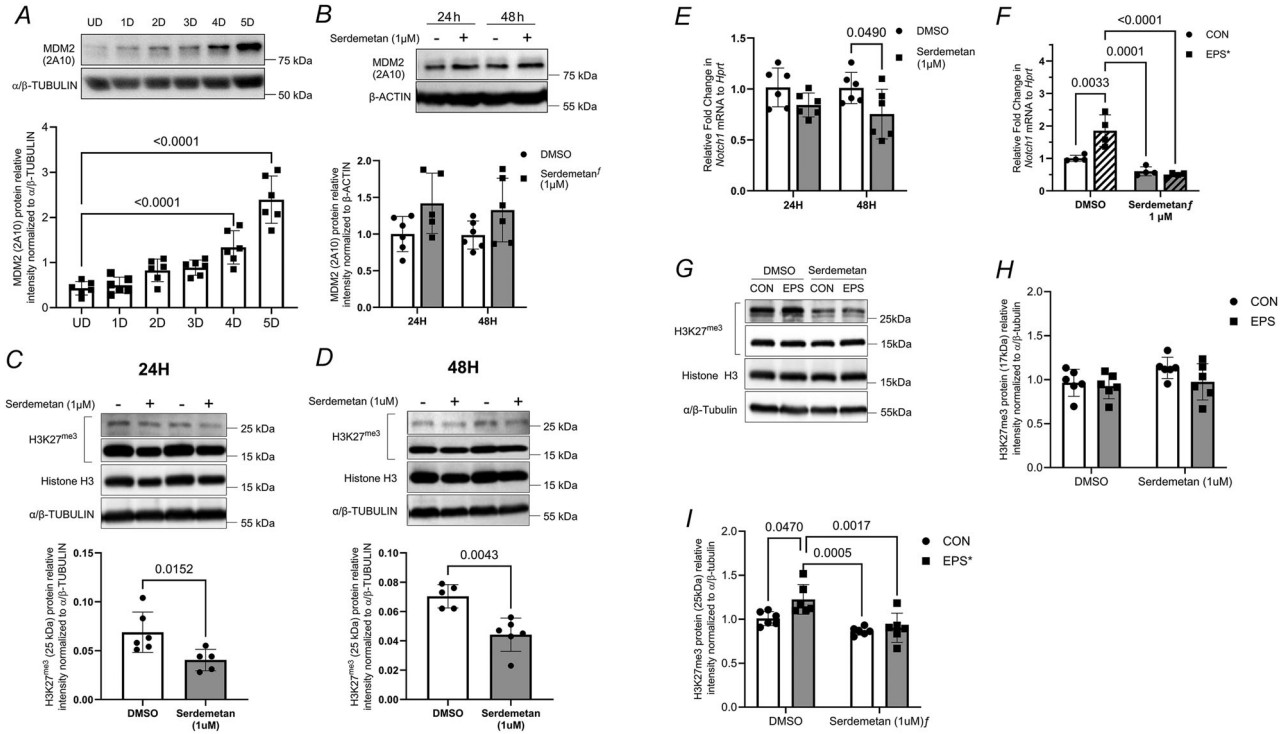

**Figure 11. Inhibition of MDM2 by Serdemetan reduces the abundance of H3K27$^{me3}$ at 25 kDa and the expression of *Notch1* mRNA in C2C12 myotubes**

*A*, C2C12 undifferentiated myoblasts (UD) were differentiated over a period of 5 days. Western blotting measured levels of MDM2 using the 2A10 antibody. $\alpha/\beta$-TUBULIN was used as a loading control. Data show mean $\pm$ SD ($n = 6$, one-way ANOVA). For *post hoc* test reaching $P \leq 0.05$, exact $P$ values are shown. *B–D*, C2C12 myotubes were treated with Serdemetan (1 µM for 24 and 48 h) or vehicle control (DMSO). *B*, immunoblotting tested the impact of Serdemetan on MDM2 protein level using 2A10 antibody. $\beta$-ACTIN was used as a loading control. Data are mean $\pm$ SD and show relative abundance of MDM2 ($n = 5$–6, *f* overall effect of Serdemetan). *C* and *D*, Western blotting measured the level of H3K27$^{me3}$. Histone H3 and $\alpha/\beta$-TUBULIN were used as a loading control. Data show relative mean $\pm$ SD ($n = 5$–6, for *t* test reaching $P \leq 0.05$, exact $P$ values are shown). *E*, MDM2 inhibition (Serdemetan, 1 µM, 48 h) decreased the expression of *Notch1* mRNA expression. Data show mean $\pm$ SD for 24 and 48 h of treatment ($n = 6$, for *post hoc* test reaching $P \leq 0.05$, exact $P$ values are shown). *F–I*, C2C12 myotubes were treated with 1 µM Serdemetan (48 h) or control vehicle (DMSO) prior and during EPS for 4 days (90 min/day). *F*, relative expression of *Notch1* mRNA to the housekeeping gene *Hprt* was measured. Data show mean $\pm$ SD [$n = 4$, for *post hoc* test *vs.* corresponding control (CON or DMSO) reaching $P \leq 0.05$, exact $P$ values are shown]. *G–I*, protein levels of H3K27$^{me3}$ were measured at 17 kDa (*H*) and 25 kDa (*I*). Histone H3 and $\alpha/\beta$-TUBULIN were used as a loading control. Data show mean $\pm$ SD ($n = 6$, two-way ANOVA). For *post hoc* test reaching $P \leq 0.05$, exact $P$ values are shown.

states of myotubes pre-treated with Serdemetan and GSK343. Pre-treatment with the inhibitors alone and after repeated exposure to EPS (4 days, 90 min daily) had no impact of the fusion index in C2C12 myotubes (Fig. 12). Under all conditions, the fusion index remained identical around 60% (Fig. 12). Hence, the differences in *Notch1* mRNA observed when C2C12 myotubes were pre-treated with inhibitors cannot by explained by a difference in differentiation states.

## Discussion

Here, we aimed to test whether endurance training increases gene silencing capacity through EZH2/MDM2-related HPTMs, when muscle remodelling comes to a standstill. Nine weeks of endurance training led to vascular and metabolic muscle adaptations. In the muscle tissue, training did not change the over-

all abundance of silencing marks (i.e. H3K27$^{me3}$ and H2AK119$^{Ub}$) and an activation mark H3K4$^{me3}$. Yet, training led to an H3K27$^{me3}$ enrichment on the promoter of genes important for skeletal muscle remodelling: *Kdr*, which codes for VEGFR2 (Hoier et al., 2020), and *Notch1*, which is a key regulator of angiogenesis and muscle regeneration (Benedito & Hellström, 2013; Vargas-Franco et al., 2022). H3K27$^{me3}$ enrichment credibly supports silencing of the *Kdr* gene, since training decreased *Kdr* mRNA expression. *Notch1* mRNA increased with training, suggesting that enrichment in H3K27$^{me3}$ failed to silence this gene. The presence of other modifications on H3 could explain this lack of silencing, potentially due to a higher abundance of activation marks on *Notch1* regulatory sequences (i.e. H3K4$^{me3}$ and H3K9$^{Ac}$) or an overall greater level of ubiquityl-H3. Indeed, training increased the detection of H3K27$^{me3}$ at a shifted molecular weight of 25 kDa in the skeletal muscle. This shift was

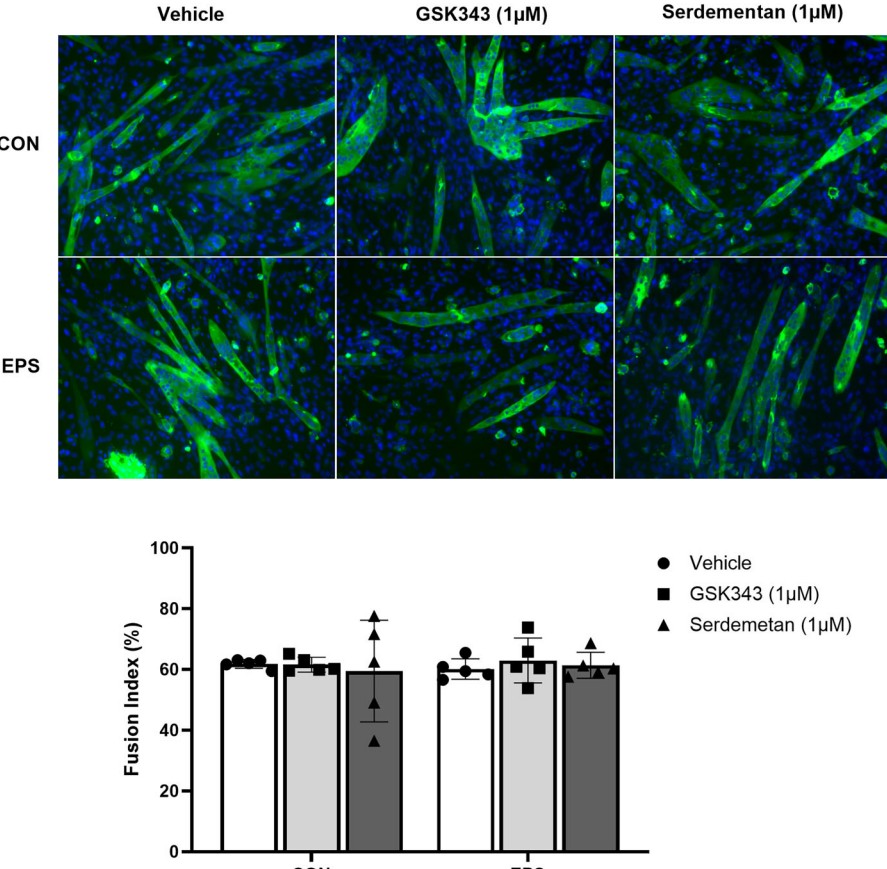

**Figure 12. Inhibitions of MDM2 or EZH2 by Serdemetan or GSK343 has no impact on fusion index in C2C12 myotubes**
C2C12 cells were differentiated for 4 days and then treated with Serdemetan (1 μM) or GSK343 (1 μM) for 48 h prior and during EPS for 4 days (90 min/day). *A*, representative merged images of MYH4 and DAPI staining of C2C12 myotubes show no differences. *B*, estimation of fusion index confirmed that after 4 days of differentiation inhibition of EZH2 or MDM2 had no impact on differentiation states even after repeated exposure to EPS. Data show mean ± SD ($n = 5$). Two-way ANOVA indicated no significant interaction ($P = 0.8594$). There was no significant impact of EPS or inhibition.

consistent with ubiquitylation of H3 (H3$^{Ub}$). While barely detectable in endothelial cells, this putative H3$^{Ub}$ is expressed in C2C12 myotubes. We mimicked daily exposure to submaximal exercise over 4 days using EPS (90 min daily) and observed higher levels of H3$^{Ub}$ in myotubes (detected at 25 kDA using an $\alpha$-H3K27$^{me3}$). EZH2 inhibition by GSK343 had no impact on basal expression of H3$^{Ub}$ but increased *Notch1* mRNA. While GSK343 treatment limited the EPS-driven accumulation of H3$^{Ub}$ and *Notch1* mRNA, the MDM2 inhibitor Serdemetan decreased levels of H3$^{Ub}$ and drastically reduced *Notch1* expression in myotubes at baseline and after repeated exposure to EPS. Our results provide evidence that MDM2-dependent ubiquitylation of H3 could increase *Notch1* expression in response to endurance training.

### Role of HPTMs in skeletal muscle

The epigenetic mechanisms that shift gene expression when the skeletal muscle adapts to endurance exercise remains to be fully understood. DNA methylation only partly explains the changes in gene expression observed with endurance training (Kanzleiter et al., 2015; Lindholm et al., 2014). After 3 months of training, only 15% of all differentially expressed genes in the human skeletal muscle follow the canonical model of a negative correlation between DNA methylation and gene expression (Lindholm et al., 2014). In C57Bl6 mice, a change in DNA methylation following the canonical model might explain about 7.5% of the differential gene expression occurring in the skeletal muscle after 4 weeks of training (Kanzleiter et al., 2015). In addition to DNA methylation, differences in the distribution of HPTMs also support transcriptomic changes in the skeletal muscle (Kawano et al., 2015; Li et al., 2024; McGee et al., 2009; Pandorf et al., 2009; Smith et al., 2023).

HPTMs play an important role in regulating muscle phenotype. Indeed, the distribution of HPTMs is considerably different between the plantaris (low C-to-F ratio and glycolytic) and the soleus (high C-to-F ratio and oxidative) muscles in rats (Kawano et al., 2015; Pandorf et al., 2009). In the highly glycolytic plantaris, an abundance of activation HPTMs (H3K4$^{me3}$ and H3$^{Ac}$) is found on the regulatory sequences of transcriptionally active genes that support a fast-twitch phenotype (e.g. *Myh1* and *Myh2*). Interestingly, there is a depletion of these activation HPTMs on genes associated with the slow-twitch phenotype in the plantaris muscle (e.g. *Myh7*) (Kawano et al., 2015; Pandorf et al., 2009). Though activation HPTMs have not been shown to support the acquisition of a slow-twitch phenotype in soleus muscle (Kawano et al., 2015), muscle unloading drives an enrichment of activation HPTMs onto the promoters of fast-twitch-related genes in the soleus (Kawano et al.,

2015; Pandorf et al., 2009). These findings support the notion that activation HPTMs are crucial for the establishment of a fast-twitch muscle phenotype.

Beyond basal muscle phenotype, HPTMs might be key to acquiring greater muscle oxidative metabolism and capillarization in response to training. Li et al. (2024) report that 8 weeks of moderate endurance training increases the abundance of H3K4$^{me3}$ on the promoter of *Myh7* and *Pgc1A* in rat gastrocnemius muscles. This aligns with our observed increases in *Notch1* and *VegfA* mRNAs in response to endurance exercise training. *Notch1* positively regulates the expression of MHC-I in the skeletal muscle, a protein coded for by the *Myh7* gene and is required to maintain and to develop oxidative muscle fibres (Yamashita et al., 2025). PGC1-$\alpha$ protein supports muscle angiogenesis and is a transcriptional co-activator of *VegfA* (Arany et al., 2008; Lin et al., 2002, 2005; Yamashita et al., 2025). Endurance training also increases the abundance of the silencing marker, H3K27$^{me3}$, on the promoter of *Myh4* (which encodes for MHC-IIb) – a gene that supports a fast-twitch phenotype (Li et al., 2024). This suggests that in response to endurance exercise training, activation and silencing HPTMs collaboratively shift transcription towards a more oxidative phenotype. Interestingly, at baseline, we report an overall greatest abundance of H3K27$^{me3}$ bands detected at 17 and 25 kDa in the oxidative soleus muscle compared to glycolytic plantaris and mixed-phenotype gastrocnemius muscles. In contrast to the observations made on plantaris and gastrocnemius muscles where the abundance of H3$^{Ub}$ increased, the overall abundance of H3K27$^{me3}$ detected at 25 kDa remained unchanged in the soleus muscle after endurance training. Together, these findings lead us to question whether these PRC2-related PTMs of H3 could be key for acquiring greater oxidative metabolism and a higher capillarization phenotype in muscles.

### Regulation of PRC2 target genes, HPTMs and muscle adaptations

If training stops or continues at the same intensity, muscle adaptation halts. Our study aimed to assess whether silencing HPTMs help halt muscle adaptations after weeks of endurance training to establish a new resting muscle homeostasis. We used the MDM2/EZH2 path as a framework to assess whether PRC2-related HPTMs help silence genes (i.e. *Kdr* and *Notch1*) once major muscle adaptations have occurred. MDM2 can interact with EZH2 to support PRC2 activity (Wienken et al., 2016). Here, endurance training increased basal levels of EZH2 and MDM2 proteins in the gastrocnemius muscle. This is consistent with previously reported enhanced abundance in muscle H3K27$^{me3}$ marks. In C57Bl6 mice, 4 weeks of endurance running exercise enhances the

abundance of myonuclear H3K27$^{me3}$ in tibialis anterior muscles collected 2 h following exercise. Interestingly, 2 weeks of training is not sufficient to alter the myonuclear level of H3K27$^{me3}$ (Ohsawa & Kawano, 2021). This observation suggests that as endurance training continues after a month, the activity of the PCR2 increases. Yet, training drives enrichment in H3K27$^{me3}$ in a gene-specific manner. Li et al. (2024) reported a decreased abundance of H3K27$^{me3}$ on the *Pgc1A* promoter and an increased abundance on the *Myh4* promoter. Here, we did not observe an impact of endurance training on the global abundance of H3K27$^{me3}$ detected at 17 kDa in the gastrocnemius of C57Bl6 mice. Interestingly, we reported enrichment in H3K27$^{me3}$ in the regulator sequences of *Kdr* and *Notch1* genes in the whole muscle tissue. MDM2 and EZH2 were reported to control the enrichment in H3K27$^{me3}$ marks on these *Kdr* and *Notch1* regulatory sites in MEFs (Wienken et al., 2016). Our results support the notion of increased PRC2 activity toward these genes following training, once muscle adaptations have already occurred.

VEGFR2 is crucial to sensing angiogenic factors. In the human and rodent skeletal muscles, *KDR/Kdr* expression peaks when the angiogenic process is highly engaged (Hoier et al., 2020; Lloyd et al., 2003). As reported here, endurance training decreased *Kdr* mRNA, meeting the canonical models where H3K27$^{me3}$ hinders gene transcription and MDM2 acts as a gene silencer (Guo et al., 2021; Wienken et al., 2017). Training reduces the capacity of a bout of acute exercise to upregulate *Kdr* expression in skeletal muscles (Olfert et al., 2001). After 9 weeks of training, muscles achieved microvascular adaptation, as shown by an increased C-to-F ratio and PECAM expression. The increased abundance of H3K27$^{me3}$ on the *Kdr* regulatory sequence and its reduced mRNA levels suggests that establishment of silencing marks by the PRC2 complex might help reduce the muscle sensitivity to pro-angiogenic signals (e.g. VEGF-A) once angiogenesis is completed.

Yet, the canonical axis does not apply to the regulation of *Notch1* in myotubes and in the muscle tissue. Despite an enrichment in H3K27$^{me3}$, *Notch1* mRNA increased. While *Notch1* mRNA stability and *Notch1* TSS transcriptional activity were not directly assessed, our results indicate that *Notch1* regulatory regions enriched in H3K27$^{me3}$ present concomitant enrichment of activation HPTMs. We observed a positive correlation between the H3K27$^{me3}$ and the H3K4$^{me3}$ marks on the *Notch1* promoter. Also, regulator sequences of *Notch1* potentially retained more H3K9$^{Ac}$ marks than *Kdr* after training (*P* = 0.059). Together, our findings question whether silencing (H3K27$^{me3}$) and activation (H3K4$^{me3}$ and H3K9$^{Ac}$) marks are both present on the *Notch1* locus. While muscle loci marked by H3K27$^{me3}$ and H3K4$^{me3}$ could be indicative

of exercise-responsive loci in the tibialis anterior muscle (Kawano, 2021; Shimizu & Kawano, 2022), co-occupancy remains uncertain. Similarly to Kawano and colleagues, our ChIP experiments for H3K4$^{me3}$ and H3K27$^{me3}$ were not performed sequentially (ChIP-reChIP), and both marks could be present on *Notch1* promoter yet on DNA fragments that may originate from different cell types. In the skeletal muscle multinucleated myofibres and multiple mononuclear cell populations coexist. This is particularly relevant as nuclear heterogeneity increases during exercise-driven remodelling (Koopmans et al., 2022; Murach et al., 2022). *Notch1* is expressed in multiple cell types present in the microvascular stem cell niche of the muscle, including the endothelial and satellite cells (Pitulescu et al., 2017). Endothelial expression of *Notch1* is essential for vascular expansion, supporting the establishment of a stalk endothelial phenotype during angiogenesis (Blanco & Gerhardt, 2013; Mack et al., 2017; Tammela et al., 2011). Enrichment of H3K27$^{me3}$ on the pulmonary endothelial *Notch1* reduces its expression, limiting angiogenesis (Tang et al., 2015). After 9 weeks of training (including the last 5 weeks at a constant intensity), vascular expansion is achieved. *Notch1* expression might not be required to support the stalk endothelial phenotype. Therefore, silencing of endothelial *Notch1* through H3K27$^{me3}$ cannot be excluded. Once angiogenesis has occurred, remodelling might still be ongoing, particularly in the microvascular satellite cell niche to reach a new homeostatic state. NOTCH1 signal supports the crosstalk between endothelial and muscle satellite cells to enhance niche homeostasis (Verma et al., 2018) and could support vascular maturation (Pedrosa et al., 2015). To halt angiogenesis, endothelial cells from the newly formed capillaries connect to NOTCH1-positive cells before being incorporated into the arterial vascular tree (Hasan et al., 2017; Mack et al., 2017; Pitulescu et al., 2017). Conversely, satellite cells produce VEGF-A to attract endothelial cells. In return, endothelial cells activate NOTCH1 signalling in satellite cells promoting self-renewal (Verma et al., 2018). In C57Bl6 mice, exercise-driven angiogenesis precedes the shift in fibre types (Waters et al., 2004). While angiogenesis has already halted, myofibre adaptation and satellite cell niche remodelling might still be continuing. *Notch1* expression supports the maturation and terminal differentiation of satellite cells (Yamashita et al., 2025). NOTCH1 activation in multinuclear myotubes triggers renewal of adjacent satellite cells (Bi et al., 2016), suggesting that *Notch1* expression in myofibres could enhance the replenishment of muscle stem cells. It would not be surprising that the *Notch1* gene is silenced in some cells (e.g. endothelial cells) while it remains transcriptionally active in other cells (e.g. myofibre and satellite cells). This explains why both activation H3K4$^{me3}$ or

H3K9$^{Ac}$ marks and silencing H3K27$^{me3}$ are found on the *Notch1* regulatory sequence when performing ChIP on whole muscle.

## Ubiquityl-H3, Notch1 expression and muscle C2C12 cells

Our results underline a potential role of H3$^{Ub}$ in muscle adaptation to endurance training, positively impacting *Notch1* gene expression. Very little is known regarding the physiological relevance of this ubiquityl-H3 in muscle. The abundance of this putative H3$^{Ub}$ form was significantly greater in myotubes when compared to myoblasts. The highly oxidative soleus muscle has greater levels of ubiquityl-H3 than other muscle at rest. Yet, weeks of training led to greater level of H3K27$^{me3}$ detected at 25 kDa in the plantaris and gastrocnemius muscles (trained *vs.* sedentary mice), not in the oxidative soleus muscle. Additionally, repeated exposures to electro-stimulation increased the levels of this putative form of H3$^{Ub}$ in C2C12 myotubes, mimicking the increase observed after endurance training in the tissue of glycolytic-dominant or mixed muscles. NEDD4 has previously been reported to ubiquitylate H3 on multiple lysine residues (K23, K36, K37) in tumour cells, easing the transcriptional re-activation of genes through the establishment of the H3K9$^{Ac}$ mark (Zhang et al., 2017). On the other hand, ubiquitylation of H3 on K18 (H3K18$^{Ub}$) reinforces the status of heterochromatin in cancer cells, silencing genes (Liu et al., 2025). Training increased basal levels of NEDD4 in the gastrocnemius muscle but had no impact on the global abundance of H3K23$^{Ub}$. Since *Notch1* regulatory sequences had an H3K27$^{me3}$ enrichment after training and a positive correlation could exist with activation HPTMs, it would be tempting to conclude that H3$^{Ub}$ supports timely upregulation of *Notch1* expression in myotubes, potentially in myofibres.

Previous studies reported downregulation of *Notch1* through H3K27$^{me3}$ enrichment in myoblasts, matching the canonical model of PRC2 acting as a gene silencer (Acharyya et al., 2010). Recruitment of EZH2 on DNA regions proximal to *Notch1* TSS leads to H3K27$^{me3}$ enrichment and reduction of *Notch1* mRNA in TNF-$\alpha$-treated satellite cells and C2C12 myoblasts (Acharyya et al., 2010). EZH2 expression in myoblasts stops the acquisition of a muscle phenotype by silencing muscle-specific genes (e.g. *Myh4*) through the establishment of H3K27$^{me3}$ on their promoters (Caretti et al., 2004). In this context, EZH2 activity is gene-dependent and regulated through its inter-action with Ying Yang 1 protein (YY1). As myoblasts differentiated into myotubes, EZH2 occupancy on muscle loci decreases leading to a removal of H3K27$^{me3}$ silencing mark on these genes to support the phenotypic change

(Caretti et al., 2004), suggesting that EZH2 is predominantly expressed and active in myoblasts to maintain stemness. Interestingly, the interaction of MDM2 with EZH2 supports the activity of PRC2 activity, maintaining stemness through H3K27$^{me3}$ marking in MEFs (Wienken et al., 2016). Here, EZH2 inhibition reduced the abundance of H3K27$^{me3}$ detected at 17 kDa and led to a higher basal level of *Notch1* mRNA in myotubes. This suggests that EZH2 activity might also be required to silence *Notch1* in myotubes under resting conditions. Yet, it cannot be excluded that some mononuclear myoblasts remained present in our culture, driving at least partially the positive effect of EZH2 inhibition on *Notch1* mRNA expression.

## Role of mononuclear cells in increasing H3$^{Ub}$ *in vivo*

Expression of EZH2 and H3K27me3 is crucial to muscle development and growth (Caretti et al., 2004; Juan et al., 2011; Woodhouse et al., 2013). EZH2 is highly expressed in activated muscle stem cells leading to an increased abundance of H3K27$^{me3}$ in young mice (8 weeks). In pre-mature mice, the EZH2 pathway might support the acquisition of a permissive state of chromatin where genes important for muscle growth and adaptation are marked with both H3K27$^{me3}$ and activating HPTMs (i.e. H3K4$^{me3}$) (Liu et al., 2013; Shimizu & Kawano, 2022). In adult mice, EZH2 supports muscle regeneration through the maintenance of the pool of stem cells (Juan et al., 2011). As mice grow, muscle EZH2 expression decreases gradually postnatally and H3K27$^{me3}$ abundance increases globally in quiescent muscle stem cells as mice age (i.e. 24 months) (Juan et al., 2011; Liu et al., 2013). In the present study, mice were trained from week 9 to week 18, reaching a mature age while being engaged in endurance training for a few weeks. Our observation that endurance training suggests that H3$^{Ub}$ might only prevail in young adult mice where muscle stem cells retain a permissive state of chromatin that might favour muscle adaptation and epigenetic memory (Liu et al., 2013; Sharples et al., 2016). Under a similar endurance training protocol, Murugathasan et al. (2023) reported that macrophages present important changes in their phenotype due to important epigenetic remodelling. Whether ubiquitylation of H3 plays a role in determining the phenotypes of muscle macrophages remains largely unknown in response to training. It cannot be excluded that the increase of H3$^{Ub}$ observed *in vivo* is driven by changes in the abundance and phenotype of muscle stem cells and macrophages (Juan et al., 2011; Liu et al., 2013; Oss-Ronen et al., 2022; Sawano et al., 2014; Tonkin et al., 2015; Walton et al., 2019; Wu et al., 2022). Future investigations conducted on young, mature, and older mice are warranted to delineate the exact role H3$^{Ub}$ in

different cell types recruited during muscle adaptation to training across the mouse lifespan.

## Role of MDM2-dependent ubiquitylation of H3 in the skeletal muscle and future directions

In C2C12 myotubes, repeated exposure to daily sessions of contractile activity increased the abundance of H3$^{Ub}$, detected at 25 kDa and confirmed by co-IP. MDM2 appears as a positive regulator of *Notch1* expression and H3$^{Ub}$ in myotubes. Previously, Wienken et al. (2016) reported that EZH2 recruits MDM2 in the promoters of PRC2 target genes. Since EZH2 inhibition prevented the increase in H3$^{Ub}$ and *Notch1* expression after repeated exposure to contractile activity, the positive impact of MDM2 on H3$^{Ub}$ and *Notch1* might require EZH2 activity. It remains uncertain whether MDM2 directly ubiquitylates H3. Serdemetan interacts with the MDM2 ring domain, impairing its E3 ubiquitin ligase activity (Chargari et al., 2011). Serdemetan-treated myotubes expressed significantly less *Notch1* mRNA and had a reduction in the global abundance of the putative H3$^{Ub}$. When exposed to repeated contractile activity, Serdemetan-treated myotubes could not increase *Notch1* mRNA and H3$^{Ub}$. A previous study reported that MDM2 interacts with histones, H2A, H2B and H3, promoting their mono-ubiquitination, an event that requires the MDM2 ring domain (Minsky & Oren, 2004). Indeed, multiple E3 ubiquitin ligases were reported to modify H3 through mono- and poly-ubiquitylation in cell lines with various impacts on gene transcription (Ishiyama et al., 2017; Vaughan et al., 2021; Wang et al., 2006; Zhang et al., 2017). Additionally, mono-ubiquitylation primes lysine for poly-ubiquitylation, a signal that can trigger proteasomal degradation of histone proteins (Vaughan et al., 2021). H3 poly-ubiquitylation is a key signal to weaken the association between histones and DNA (Wang et al., 2006), easing H3 dissociation from the chromatin potentially increasing its degradation and the turnover of histone (Xia et al., 2017). This mechanism might help clear pre-existing HPTMs on the chromatin and help loosen the nucleosome in the skeletal muscle. Indeed, Ohsawa & Kawano (2021) reported that endurance exercise increases the turnover of histone and the incorporation of new histone in transcriptional active loci on the skeletal muscle. This might support transcriptional activation of genes in loci where new histones are incorporated (Ohsawa & Kawano, 2021). Yet, our results support the idea that MDM2 activity promotes the formation of a form of H3$^{Ub}$ at a molecular weight coherent with mono-ubiquitylation. Whether MDM2-dependent mono-ubiquitylation of H3 could be prime H3 for later poly-ubiquitylation with the intervention of co-factors or E4 ubiquitin ligase remains uncertain (Brenkman et al.,

2008; Choi et al., 2019; Lai et al., 2001; Wu et al., 2011; Xia et al., 2017). Here, proteosome inhibition by MG132 did not lead to accumulation of H3$^{Ub}$ in C2C12 myotubes.

While H3$^{Ub}$ might transcriptionally activate *Notch1* through an EZH2–MDM2-dependent ubiquitylation of H3 in myotubes, it remains unknown whether this mechanism could be broadened to other genes important for muscle adaptations. Mono-ubiquitylation of H3 on multiple lysine residues is essential for H3 acetylation of TSS of glucose-responsive genes in tumour cells (Zhang et al., 2017). Interestingly, NEDD4 binds and stabilizes MDM2 in cell lines (Xu et al., 2015). Whether an interaction between NEDD4 and MDM2 can support gene re-activation through sequential changes in the post-transcriptional modifications present on H3 (i.e. ubiquitylation and acetylation) remains largely unexplored in muscle cells. Multiple mono-ubiquitination of H3 have been reported on a few lysine residues: K4, K18, K23, K121 and K125 (Dasgupta et al., 2024; Oss-Ronen et al., 2022; Wang et al., 2006; Zhang et al., 2017). Mono-ubiquityl forms of H3 are associated with the activity of other E3 ligase RNF8, CUL4A/B, UHRF1, NEDD4 or CUL4A/B that might be associated with either the maintenance of DNA methylation, DNA replication, DNA damage repair (DDR) and modulation of chromatin structure (Oss-Ronen et al., 2022). Future studies will need to identify which lysine residues on H3 are mono-ubiquitylated in response to contractile activity in muscle tissue and cells.

MDM2 has been reported to have a pro-angiogenic function *in vivo* in the skeletal muscle through its capacity to regulate transcription factors, such as FOXO1 (Aiken et al., 2016; Roudier et al., 2012). While MDM2 emerges as a key regulator of epigenetic process (Klusmann et al., 2018; Riscal, Le Cam et al., 2016; Riscal, Schrepfer et al., 2016; Wienken et al., 2017) it would be valuable to investigate whether MDM2 pro-angiogenic functions require H3$^{Ub}$. For example, hypomorphic mice that have a reduced MDM2 expression have a reduced capacity to perform angiogenesis in response to endurance exercise (Roudier et al., 2012). The results presented here question whether reduced expression of MDM2 is associated with a reduced capacity to ubiquitylate H3 on the promoters of pro-angiogenic genes, such as *Kdr*. To answer this question future studies will need to investigate how MDM2-dependent ubiquitylation of H3 is modulated during the time course of endurance training.

## Conclusions

This present work shows that MDM2 supports the establishment of a ubiquitylated form of H3 that could regulate the expression of genes in the myotubes and in the

skeletal muscle. The discovery of this mark might explain the reported dichotomic effect of H3K27$^{me3}$ marking by PRC2 on genes key to muscle adaptation (i.e. *Kdr* and *Notch1*). The enrichment of H3K27$^{me3}$ by PRC2 might silence *Kdr*. MDM2 dependent-ubiquitylation of H3 might explain why H3K27$^{me3}$ enrichment fails to silence *Notch1* in the days that follow cessation of training. Whether H3$^{Ub}$ is crucial to halt muscle adaptation or to establish a new muscle homeostasis, particularly within the microvascular satellite cell niche, remains unknown. Future investigations are required to delineate the impact of the ubiquitylation of H3 by MDM2 on *Notch1* signal and other key transcriptional pathways important for skeletal muscle adaptations during and following endurance training.

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

## Additional information

### Data availability statement

Data will be made available upon request to Dr Roudier. Western blot images, densitometry analyses and qPCR data are available in the file 19468_1_supp_data_406007_.syvpht.pdf

### Competing interests

The authors declare no conflicts of interest.

### Author contributions

E.R. and B.L. conceived and designed the research with the support of A.A.S. M.M. supported the endurance training protocol. B.L., M.M. and E.R. supported tissue collection. B.L., S.A., M.G. and B.L. supported the collection and analysis of skeletal muscle tissues. B.L., S.A., M.G., M.T., P.L. and E.R. interpreted the data. E.R., S.A. and P.L. performed and interpreted ChIP experiments. M.T. and B.L. performed C2C12 experiments, associated co-IP and interpretation of data. E.R. and B.L. prepared figures and drafted the first version of the manuscript. All authors edited and revised the manuscript supporting further analyses and interpretations of the data. All authors approved the final version of the manuscript.

### Funding

This research was supported by the Natural Sciences and Engineering Research Council of Canada (NSERC) Discovery Grants Program, grant # RGPIN-2020-07142DG awarded to Dr Emilie Roudier.

## Acknowledgements

The authors thank Eric Chung and Elizabeth Karvasarski for technical assistance in measuring capillary-to-fibre ratio and in performing western blotting using 2A10 antibody, respectively. The authors thank Dr Maxime Rossato for running MS analysis of IP products, Dr Michael Connor for sharing equipment required for EPS and Dr Olivier Birot for sharing equipment crucial for protein analyses.

## Keywords

angiogenesis, epigenetics, exercise physiology, gene expression, histone modifications, murine double minute 2, muscle adaptation

## Supporting information

Additional supporting information can be found online in the Supporting Information section at the end of the HTML view of the article. Supporting information files available:

**Peer Review History**
**Supp data**

