## [Peer Review History · The Journal of Physiology]

Endurance training increases a ubiquitylated form of Histone H3 in the skeletal muscle, supporting *Notch1* upregulation in an MDM2-dependent manner

Brian Lam, Sokaina Akhtar, Manpreet Gulri, Pierre Lemieux, Monica Tawadrous, Mayoorey Murugathasan, Ali A. Abdul-Sater, and Emilie Roudier

DOI: 10.1113/JP288947

Corresponding author(s): Emilie Roudier (eroudier@yorku.ca)

The following individual(s) involved in review of this submission have agreed to reveal their identity: Fran Morena-Clark (Referee #2)

Review Timeline:

Submission Date:	15-Apr-2025
Editorial Decision:	09-May-2025
Revision Received:	04-Jul-2025
Editorial Decision:	17-Jul-2025
Revision Received:	18-Jul-2025
Accepted:	01-Aug-2025

Senior Editor: Paul Greenhaff

Reviewing Editor: Kevin Murach

Transaction Report:

Dear Dr Roudier,

Re: JP-RP-2025-288947 "**Endurance training increases a ubiquitylated form of Histone H3 in the skeletal muscle, supporting *Notch1* upregulation in an MDM2-dependent manner**" by Brian Lam, Sokaina Akhtar, Manpreet Gulri, Monica Tawadrous, Pierre Lemieux, Mayoorey Murugathasan, Ali Abdul Sater, and Emilie Roudier

Thank you for submitting your manuscript to The Journal of Physiology. It has been assessed by a Reviewing Editor and by 2 expert referees and we are pleased to tell you that it is potentially acceptable for publication following satisfactory major revision.

REVISION CHECKLIST:

We look forward to receiving your revised submission.

Yours sincerely,

Paul Greenhaff
Senior Editor
The Journal of Physiology

REQUIRED ITEMS

- Author photo and profile. First or joint first authors are asked to provide a short biography (no more than 100 words for one author or 150 words in total for joint first authors) and a portrait photograph. These should be uploaded and clearly labelled together in a Word document with the revised version of the manuscript. See Information for Authors for further details.

- You must start the Methods section with a paragraph headed Ethical approval (https://jp.msubmit.net/cgi-bin/main.plex?form_type=display_requirements#methods).

Research must comply with The Journal's policies regarding animal experiments (<https://physoc.onlinelibrary.wiley.com/hub/animal-experiments>) and adherence to these policies must be stated in the manuscript.

Authors should confirm in their Methods section that their experiments were carried out according to the guidelines laid down by their institution's animal welfare committee, including an ethics approval reference number. The Methods section must contain a statement about access to food, water and housing, details of the anaesthetic regime: anaesthetic used, dose and route of administration, and method of killing the experimental animals.

- Please upload separate high-quality figure files via the submission form.

- You must upload original, uncropped western blot/gel images (including controls) if they are not included in the manuscript. This is to confirm that no inappropriate, unethical or misleading image manipulation has occurred. These should be uploaded as 'Supporting information for review process only'. Please label/highlight the original gels so that we can clearly see which sections/lanes have been used in the manuscript figures. For more information, see: <https://physoc.onlinelibrary.wiley.com/hub/journal-policies#imagmanip>.

- Please ensure that any tables are editable and in Word format, and wherever possible, embedded in the article file itself.

- Please ensure that the Article File you upload is a Word file.

- Papers must comply with the Statistics Policy: https://jp.msubmit.net/cgi-bin/main.plex?form_type=display_requirements#statistics.

In summary:

- If $n \leq 30$, all data points must be plotted in the figure in a way that reveals their range and distribution. A bar graph with data points overlaid, a box and whisker plot or a violin plot (preferably with data points included) are acceptable formats.
- If $n > 30$, then the entire raw dataset must be made available either as supporting information, or hosted on a not-for-profit repository, e.g. FigShare, with access details provided in the manuscript.
- 'n' clearly defined (e.g. x cells from y slices in z animals) in the Methods. Authors should be mindful of pseudoreplication.
- All relevant 'n' values must be clearly stated in the main text, figures and tables.
- The most appropriate summary statistic (e.g. mean or median and standard deviation) must be used. Standard Error of the Mean (SEM) alone is not permitted.
- Exact p values must be stated. Authors must not use 'greater than' or 'less than'. Exact p values must be stated to three significant figures even when 'no statistical significance' is claimed.

- Please include an Abstract Figure file, as well as the Figure Legend text within the main article file. The Abstract Figure is a piece of artwork designed to give readers an immediate understanding of the research and should summarise the main conclusions. If possible, the image should be easily 'readable' from left to right or top to bottom. It should show the physiological relevance of the manuscript so readers can assess the importance and content of its findings. Abstract Figures should not merely recapitulate other figures in the manuscript. Please try to keep the diagram as simple as possible and without superfluous information that may distract from the main conclusion(s). Abstract Figures must be provided by authors no later than the revised manuscript stage and should be uploaded as a separate file during online submission labelled as File Type 'Abstract Figure'. Please also ensure that you include the figure legend in the main article file. All Abstract Figures should be created using BioRender. Authors should use The Journal's premium BioRender account to export high-resolution images. Details on how to use and access the premium account are included as part of this email.

EDITOR COMMENTS

Reviewing Editor:

Your work has been evaluated by two experts in the field. Both find merit in the work and think the results are potentially impactful. Both reviewers raised concerns related to clarity in the Methods and interpretation, and some further experimentation may be required to address some concerns specifically raised by Reviewer 2, specifically as it relates to the inclusion of an earlier training time point and the cell culture work. Please respond to the reviewer comments in full.

Please also see 'Required Items' above.

Senior Editor:

Thank you for the manuscript submission to The Journal of Physiology, which has been considered by a reviewing editor and two expert reviewers. Both reviewers thought there was merit in the work and that the paper could be impactful. Nevertheless, both have raised a number of major concerns that need to be addressed if the manuscript is to be considered further. Importantly, these concerns are likely to require additional studies being conducted by the authors. On more specific points. The authors state that mice were anaesthetized 72 hours after the last bout of exercise and hind-limb muscles were harvested before the mice were euthanized by cervical dislocation. Please could the authors add a statement to the text to describe the anaesthesia protocol (dose, route, supplementary dosage if necessary) and how the depth of anaesthesia was determined and maintained. Furthermore, please ensure the manuscript shows mean \pm SD throughout (not SEM) and exact p values are presented (not $<$ or $>$). Thank you again for the manuscript submission and we look forward to receiving the revised manuscript in due course.

REFEREE COMMENTS

Referee #1:

The manuscript by Lam and colleagues sought to investigate and expand our understanding of exercise training adaptations in skeletal muscle with specific focus on histone post translational modifications (HPTMs). The authors further focused on role of the E3 ubiquitin ligase MDM2 and its interaction with EZH2 towards regulating Histone H3 via trimethylation. The authors identify an interesting HPTMs through a ubiquitylation modification and probe how this might influence training status through in vivo and in vitro models. Detailed below are my comments.

1) Figure 1. Representative microscope images used for the capillary-to-fibre ratio analysis in panel C. should be included in this figure.

2) Tubulin has been used as a loading control in numerous western blots, however often this control appears highly variable. I wondered if the authors had considered normalizing to total protein loading via measurement of the whole lane through either stain-free gel technology or coomassie/ponceau staining?

3) The authors state 'The band detected at 25kDa was present in C2C12 myotubes and was significantly less abundance and less detectable in mouse skeletal muscle endothelial cells (mSMECs)', is it possible that other cell types found within muscle could express this band?

4) The authors assessed differences in the protein abundance for the 25kda size between hindlimb muscles and reported highest amounts in the Soleus and with training. I'm wondering what the gene expression patterns for Notch1 and Kdr are given the evidence presented and how this pattern might compare to the gastrocnemius muscle itself?

5) In figures, the panels depicting data order for the 17kda and 25kd quantification (e.g. figure 6 A and B) show the 17kd data set first and then the 25kd data set, yet in other figures, this order is switched around. If color is not to be used to distinguish 17kda and 25kda related data, then I'd suggest keeping a consistent order.

6) Regarding MDM2, the following information might be added to the manuscript or additional data included; What type of E3 ubiquitin ligase is MDM2? Are there other known protein substrates that MDM2 might target for ubiquitylation? Is the ubiquitin modification on histone 3, mono or polyubiquitin? The ubiquitin antibody used suggests a polyubiquitin event.

Is there a way to tell if it is specific linkages such as K63 or K48, to assess whether the modification is proteolytic or non-proteolytic?

7) The authors state in the discussion that the plantaris and gastrocnemius muscle displayed the greatest level of H3K27me at the 25kda (page 35) but that is not represented in figure 6.

8) The inhibitor work for MDM2 and EZH2 in myotubes suggests changes in gene expression for Notch1, Im wondering if any change was observed for kdr expression? Can this proposed mechanism be tested in vivo, i.e. does MDM2 inhibition reduce angiogenesis and/or impair training adaptations?

Referee #2:

General Comments

This manuscript provides compelling insights into how endurance training influences epigenetic regulation in skeletal muscle, particularly through histone H3 ubiquitylation and its potential impact on Notch1 activation via MDM2. The study highlights an underexplored mechanism with important implications for exercise-induced muscle adaptation. However, several methodological and interpretative issues need to be addressed to fully support the authors' conclusions. These include the use of "immature"/still developing mice, bouncing back and forth between gastroc, plantaris, and TA, while making some important observations of differences in UbH3K27me3 (25kd) abundance in highly oxidative vs glycolytic tissues, limited sample sizes in key assays, and insufficient conditions (e.g., lack of EPS-treated groups, specifically in figure 9). Additionally, the current design does not test the hypothesis of "capped" adaptations, as it lacks comparisons across different training durations. Lastly, while angiogenic markers (Kdr, Notch1) are relevant targets, a broader methylomic+transcriptomic integration approach could better capture the global scope of epigenetic remodeling (although good points are made in the discussion about the disconnect between methylation and transcriptional changes). Addressing these issues would greatly improve the manuscript's coherence and impact.

Methods

(Page 4)

- The mice used (7-8 weeks old) are still undergoing postnatal development (as per Jackson Labs). How do the authors account for potential interactions between natural maturation and training-induced effects? Would they anticipate similar molecular outcomes in fully mature mice (12+ weeks)? Including a brief discussion of this limitation or justifying the use of young mice would be helpful.

(Page 6)

- Were sedentary animals also placed on stationary treadmills to control for handling and environmental stressors? Even minimal manipulation can trigger physiological responses. If not, this introduces a potential confounder that should be acknowledged or addressed in future designs.

(Page 7)

- Please clarify what confluency (%) was considered "confluent" before differentiation. This can substantially influence downstream differentiation and gene expression.

(Page 8)

- What solvents were used to dissolve GSK343 and Serdemetan? If DMSO was used, were vehicle controls included? Please ensure these are explicitly described in the methods.

- GSK343, while potent for EZH2, also exhibits activity toward EZH1 (60-fold selectivity). Given recent evidence tying EZH1 to exercise-induced H3K27me3 and H3K4me3 changes (e.g., PMID: 35666835), could off-target effects be influencing results? The authors could either address this in the discussion or consider testing an EZH1-selective or dual EZH1/2 inhibitor to exclude confounding effects.

- "24-48 hours" is a broad time window. It seems per figure 9 that it is 24 or 48 hours per condition. Please specify to improve clarity and reproducibility.

- Please clarify the number of biological and technical replicates per group for the in vitro experiments. This is critical for evaluating reproducibility and statistical robustness.

(Pages 6-7)

- Were the EPS stimulation settings (frequency, pulse duration, rest interval) derived from published protocols? What type of in vivo exercise does this stimulation attempt to mimic (e.g., endurance vs. resistance)? Including rationale for EPS parameters would improve interpretation.

(Page 12)

- Was a priori power analysis conducted to justify the use of n=7 for animal experiments? Were data tested for normality and equal variance? Please describe the statistical workflow in more detail.

Results

A major claim here is the role that methylation plays in gene silencing, and this idea of "capped muscle adaptations" occurs when stimuli are maintained at similar levels for a prolonged time. A little more rationale is needed to support the methodology set for the 9-week running protocol implemented here. The reviewer partially expected an earlier timepoint of 4-6 weeks where adaptations should be seen and then compared at the later 9-week protocol, and with that have a better handle on adaptations indeed "stopped" or "slowed down" from 6 to 9 weeks.

Figure 1 (Page 14)

- Can the authors include representative images for the capillary-to-fiber ratio quantification? Were there any observed differences in cross-sectional area (CSA), or was the increase driven solely by capillary density?
- In Figure 1, why do the n-values differ between COXIV and PECAM quantifications (7 vs. 4 samples from the same experimental groups)? Were some samples excluded due to technical failure or outlier removal? Please clarify exclusion criteria.
- Given that both gastrocnemius and plantaris are mixed fiber-type muscles, would the authors expect a different outcome in more oxidative muscles such as the soleus or diaphragm? Brief speculation or rationale would help contextualize tissue-specific findings.
- Was the SMP14-detected band for MDM2 quantified at 75 or 100 kDa? Please clarify which isoform is shown.
- Some control samples for EZH2 and the hypo-phosphorylation antibody show low protein levels while others are comparable to trained. Is this expected biological variability, or could it reflect technical inconsistency?
- In Figure 2A, the graph is marked as ** ($p < 0.01$) but the text states $p \leq 0.05$. Please revise to ensure consistency between text, figure legend, and graph.

Page 19

- The DUB experiments support your claim, but it would strengthen the conclusion to include additional protein samples (e.g., from both trained and sedentary animals) to verify consistency across biological replicates.

Page 22

- The structure of the results is sometimes hard to follow. For example, why is plantaris used in Figure 6 instead of gastrocnemius as in Figure 1? Improved figure transitions and rationale for tissue selection would enhance clarity and narrative flow.
- It is appropriate to hypothesize that increased H3K27me3-Ub is tied to myogenic differentiation considering its role in expression control of important myogenic genes such as MyoD1. However, skeletal muscle homogenates contain satellite cells, macrophages, and other non-myogenic populations (as recognized by the authors in the context of lack of silencing of notch1 in the discussion). Could the observed changes be partially driven by these cells? This possibility should be acknowledged.

Figure 7 (Page 24)

- In Figure 7C, statistical significance may be driven by a single high-value outlier. Similarly, Figure 7D's correlation is based on only 5 data points, heavily influenced by extremes. These findings are potentially underpowered. Consider increasing biological replicates to strengthen confidence in these associations.
- Additionally, could the Kdr IP correlation graph be included as it is referenced in the text?
- The switch to tibialis anterior (TA) in Figure 7E for H3K9ac is inconsistent with prior data from soleus, plantaris and gastrocnemius. Either unify muscle groups across epigenetic analyses or clearly justify these choices.
- The $n=4$ per group in these panels appears insufficient given the large variance, especially in control groups (especially as seen in E-F). Please comment on whether these datasets were adequately powered to detect interaction effects.
- Since the H3K27me3-Ub form is linked to differentiation, did the authors assess markers of differentiation or quantify fusion index in C2C12 cells? Including representative images of treated cultures (e.g., EPS or inhibitors) could help confirm that observed changes are not due to differences in differentiation states.

Figure 9

- Please include the relevant statistical significance markers in the figure panel, as they are mentioned in the figure

legend/text (Figure 9B).

- Including an EPS group in this experiment would strengthen the argument that MDM2 mediates epigenetic responses to contractile activity. This could tie together the MDM2 mechanism with exercise-mimetic stimuli response more convincingly.

Discussion

(Page 31)

- The sentence below seems to be incomplete. Do the authors mean "higher of H3Ub in stimulated myotubes"?

"We mimicked daily exercise over 4 days using EPS (90 minutes daily) and observed higher levels of H3Ub in myotubes (detected at 25kDA using an α -H3K27me3)."

(Page 33)

- Please insert citation following this sentence:

"In C57Bl6 mice, 4 weeks of endurance running exercise enhances the abundance of myonuclear H3K27me3 in tibialis anterior muscles collected 2 hours post-exercise."

(Page 34)

Although the statement is true to the data, vascular adaptations in rodents are noticeable as early as 2 weeks and more pronounced at 4 weeks post-training. Do the author infer that these vascular adaptations need to be stable so that gene silencing of *Krd*, for example, occurs? Again, having an earlier 4-week training group would be helpful here to directly support this statement.

"After 9 weeks of training, muscles achieved microvascular adaptation, as shown by an increased C-to-F ratio and PECAM expression. The increased abundance of H3K27me3 on *Kdr* regulatory sequence and its reduced mRNA levels suggests that establishment of silencing marks by the PRC2 complex might help reduce the muscle sensitivity to pro-angiogenic signals (e.g. VEGF-A) once angiogenesis is completed."

END OF COMMENTS

General Comments

This manuscript provides compelling insights into how endurance training influences epigenetic regulation in skeletal muscle, particularly through histone H3 ubiquitylation and its potential impact on Notch1 activation via MDM2. The study highlights an underexplored mechanism with important implications for exercise-induced muscle adaptation. However, several methodological and interpretative issues need to be addressed to fully support the authors' conclusions. These include the use of "immature"/still developing mice, bouncing back and forth between gastroc, plantaris, and TA, while making some important observations of differences in UbH3K27me3 (25kd) abundance in highly oxidative vs glycolytic tissues, limited sample sizes in key assays, and insufficient conditions (e.g., lack of EPS-treated groups, specifically in figure 9). Additionally, the current design does not test the hypothesis of "capped" adaptations, as it lacks comparisons across different training durations. Lastly, while angiogenic markers (Kdr, Notch1) are relevant targets, a broader methylomic+transcriptomic integration approach could better capture the global scope of epigenetic remodeling (although good points are made in the discussion about the disconnect between methylation and transcriptional changes). Addressing these issues would greatly improve the manuscript's coherence and impact.

Methods

(Page 4)

- The mice used (7–8 weeks old) are still undergoing postnatal development (as per Jackson Labs). How do the authors account for potential interactions between natural maturation and training-induced effects? Would they anticipate similar molecular outcomes in fully mature mice (12+ weeks)? Including a brief discussion of this limitation or justifying the use of young mice would be helpful.

(Page 6)

- Were sedentary animals also placed on stationary treadmills to control for handling and environmental stressors? Even minimal manipulation can trigger physiological responses. If not, this introduces a potential confounder that should be acknowledged or addressed in future designs.

(Page 7)

- Please clarify what confluency (%) was considered "confluent" before differentiation. This can substantially influence downstream differentiation and gene expression.

(Page 8)

- What solvents were used to dissolve GSK343 and Serdemetan? If DMSO was used, were vehicle controls included? Please ensure these are explicitly described in the methods.
- GSK343, while potent for EZH2, also exhibits activity toward EZH1 (60-fold selectivity). Given recent evidence tying EZH1 to exercise-induced H3K27me3 and H3K4me3 changes (e.g., PMID: 35666835), could off-target effects be influencing results? The

authors could either address this in the discussion or consider testing an EZH1-selective or dual EZH1/2 inhibitor to exclude confounding effects.

- “24–48 hours” is a broad time window. It seems per figure 9 that it is 24 or 48 hours per condition. Please specify to improve clarity and reproducibility.
- Please clarify the number of biological and technical replicates per group for the in vitro experiments. This is critical for evaluating reproducibility and statistical robustness.

(Pages 6–7)

- Were the EPS stimulation settings (frequency, pulse duration, rest interval) derived from published protocols? What type of in vivo exercise does this stimulation attempt to mimic (e.g., endurance vs. resistance)? Including rationale for EPS parameters would improve interpretation.

(Page 12)

- Was a priori power analysis conducted to justify the use of n=7 for animal experiments? Were data tested for normality and equal variance? Please describe the statistical workflow in more detail.

Results

A major claim here is the role that methylation plays in gene silencing, and this idea of “capped muscle adaptations” occurs when stimuli are maintained at similar levels for a prolonged time. A little more rationale is needed to support the methodology set for the 9-week running protocol implemented here. The reviewer partially expected an earlier timepoint of 4-6 weeks where adaptations should be seen and then compared at the later 9-week protocol, and with that have a better handle on adaptations indeed “stopped” or “slowed down” from 6 to 9 weeks.

Figure 1 (Page 14)

- Can the authors include representative images for the capillary-to-fiber ratio quantification? Were there any observed differences in cross-sectional area (CSA), or was the increase driven solely by capillary density?
- In Figure 1, why do the n-values differ between COXIV and PECAM quantifications (7 vs. 4 samples from the same experimental groups)? Were some samples excluded due to technical failure or outlier removal? Please clarify exclusion criteria.
- Given that both gastrocnemius and plantaris are mixed fiber-type muscles, would the authors expect a different outcome in more oxidative muscles such as the soleus or diaphragm? Brief speculation or rationale would help contextualize tissue-specific findings.
- Was the SMP14-detected band for MDM2 quantified at 75 or 100 kDa? Please clarify which isoform is shown.

- Some control samples for EZH2 and the hypo-phosphorylation antibody show low protein levels while others are comparable to trained. Is this expected biological variability, or could it reflect technical inconsistency?
- In Figure 2A, the graph is marked as ** ($p < 0.01$) but the text states $p \leq 0.05$. Please revise to ensure consistency between text, figure legend, and graph.

Page 19

- The DUB experiments support your claim, but it would strengthen the conclusion to include additional protein samples (e.g., from both trained and sedentary animals) to verify consistency across biological replicates.

Page 22

- The structure of the results is sometimes hard to follow. For example, why is plantaris used in Figure 6 instead of gastrocnemius as in Figure 1? Improved figure transitions and rationale for tissue selection would enhance clarity and narrative flow.
- It is appropriate to hypothesize that increased H3K27me3-Ub is tied to myogenic differentiation considering its role in expression control of important myogenic genes such as MyoD1. However, skeletal muscle homogenates contain satellite cells, macrophages, and other non-myogenic populations (as recognized by the authors in the context of lack of silencing of notch1 in the discussion). Could the observed changes be partially driven by these cells? This possibility should be acknowledged.

Figure 7 (Page 24)

- In Figure 7C, statistical significance may be driven by a single high-value outlier. Similarly, Figure 7D's correlation is based on only 5 data points, heavily influenced by extremes. These findings are potentially underpowered. Consider increasing biological replicates to strengthen confidence in these associations.
- Additionally, could the Kdr IP correlation graph be included as it is referenced in the text?
- The switch to tibialis anterior (TA) in Figure 7E for H3K9ac is inconsistent with prior data from soleus, plantaris and gastrocnemius. Either unify muscle groups across epigenetic analyses or clearly justify these choices.
- The $n=4$ per group in these panels appears insufficient given the large variance, especially in control groups (especially as seen in E-F). Please comment on whether these datasets were adequately powered to detect interaction effects.
- Since the H3K27me3-Ub form is linked to differentiation, did the authors assess markers of differentiation or quantify fusion index in C2C12 cells? Including representative images of treated cultures (e.g., EPS or inhibitors) could help confirm that observed changes are not due to differences in differentiation states.

Figure 9

- Please include the relevant statistical significance markers in the figure panel, as they are mentioned in the figure legend/text (Figure 9B).
- Including an EPS group in this experiment would strengthen the argument that MDM2 mediates epigenetic responses to contractile activity. This could tie together the MDM2 mechanism with exercise-mimetic stimuli response more convincingly.

Discussion

(Page 31)

- The sentence below seems to be incomplete. Do the authors mean “higher of H3Ub in stimulated myotubes”?

“We mimicked daily exercise over 4 days using EPS (90 minutes daily) and observed higher levels of H3Ub in myotubes (detected at 25kDA using an α -H3K27me3).”

(Page 33)

- Please insert citation following this sentence:

“In C57Bl6 mice, 4 weeks of endurance running exercise enhances the abundance of myonuclear H3K27me3 in tibialis anterior muscles collected 2 hours post-exercise.”

(Page 34)

Although the statement is true to the data, vascular adaptations in rodents are noticeable as early as 2 weeks and more pronounced at 4 weeks post-training. Do the author infer that these vascular adaptations need to be stable so that gene silencing of Krd, for example, occurs? Again, having an earlier 4-week training group would be helpful here to directly support this statement.

“After 9 weeks of training, muscles achieved microvascular adaptation, as shown by an increased C-to-F ratio and PECAM expression. The increased abundance of H3K27me3 on Kdr regulatory sequence and its reduced mRNA levels suggests that establishment of silencing marks by the PRC2 complex might help reduce the muscle sensitivity to pro-angiogenic signals (e.g. VEGF-A) once angiogenesis is completed.”

We would like to thank the editor and the referees for their comments. These comments have helped us improving the overall quality of the manuscript. The main modifications are briefly summarized below.

For the method section:

1. A better description in regards of how sedentary animal were handled and how muscle collection was performed under isoflurane before animal euthanasia.
2. Clarifications were made in the text (see methods section) regarding the statistical approach and more information are provided about biological replicates for the *in-vitro* experiments.
3. While not included in the revised version of the manuscript, we now provide a reply to the reviewers about the calculation of the n number to reach sufficient statistical power.

The results have been re-arranged to include some of the additional experiments requested by the reviewers to the best of our abilities. We would like to editorial team to take in consideration that the additional experiments performed where chosen based on the best values for money approach to ensure improvement of the manuscript due to our restricted budget. While we agree with the referee 2 that a broader “omic” approach might be valuable. The current paper was designed to bring proof-of-concept that it is worth pursuing in this direction in a second steps once more funding will be secured. This represents a significant investment for our laboratory annual funding and will therefore be the subject of a follow-up article and project. The main changes and supplemental experiments added to results section include:

1. Changing all figures to present standard deviation (mean±SD) instead of s.e.m. on all figures and present exact p values
2. Moving material from supplemental material into the two main figures (figures 1 and 7):
 - a. Pictures of cross-sectional sections of the EDL used to measure capillary-to-fiber ration in Figure 1C
 - b. The correlation between H3K27me3 and H3K4me3 for the KDR regulatory sequence is now presented in the main manuscript in Fig.7
3. For figure 7C we have added more biological replicates, and a rationale is now provided in our reply to the referees regarding number of biological replicates for ChIP experiments. In Figure 7C, we increased the number of biological replicates from 5 to 8 animals (Fig. 7C).
4. We performed additional analyses of the soleus and plantaris muscles (see new figures 4, 5 and 8), measuring level of EZH2, MDM2, H3K27me3 (proteins) and *Kdr* and *Notch1*. These new data confirm divergent response in muscle glycolytic-dominant (IIB fibers) versus oxidative dominant.
5. To confirm that endurance exercise increases the abundance of H3K27me3 at 25KDa due to a potential ubiquitylation DUB assay was performed on gastrocnemius muscle of 3 sedentary and 3 trained mice, confirming that the band detected with a greater abundance by α -H3K27^{me3} in trained gastrocnemius muscle is decreased by deubiquitinating enzymes.
6. More experiments were performed on the C2C12 myotubes:
 - a. To delineate whether the ubiquityl-H3 was triggering proteolysis C2C12 myotubes were treated with MG132 to determine whether the putative ubiquityl-H3 is associated with proteolytic activity (n=6 biological replicates). The abundance of histone H3 and H3K27me3 remained unchanged in the presence of the proteasome inhibitor.

- b. We tested the impact of GSK-343 and Serdemetan inhibition on EPS-driven changes in *Notch1* as well as *Kdr*. The new data confirm that both EZH2 and MDM2 are required to regulate *Notch1* upregulation in response to EPS in C2C12 myotubes. While our new data confirms that Mdm2 upregulates *Notch1* in myotubes. Levels of *Kdr* expression were leading to very high Ct values, reaching the limit of detection for qPCR. This is probably due to the fact the C2C12 myotubes expressed low levels of *Kdr* under our conditions.
- c. Fusion index was measured in C2C12 myotubes (4 days of differentiation) pre-treated with GSK-343 and Serdemetan before being exposed to repeated electrical pulse stimulation (90 min per day for 4 days). Our new results confirm that inhibition of EZH2 and MDM2 had minimal impact on the differentiation state of our C2C12 cultures.

For the discussion, we are now discussing:

- the possibility that other cell types found in the skeletal muscle could gained Ubiquityl-H3; indicating that it will be valuable to perform further studies to determine whether ubiquitylation of H3 is observed more broadly in cells shifting phenotype in response to endurance training and involved in muscle remodelling.
- We added brief points of discussion about the divergent response observed between the glycolytic and oxidative dominant muscle types based on the new results inserted in the revised version.
- We examine the role of MDM2 as a ring E3 ubiquitin ligase and its capacity to interact with transcription factors, with PRC1 and EZH2 and histones, also discussing whether the ubiquityl-form of H3 is linked or not to proteolysis.
- We added a few points about potential limitations, including:
 - Working on mice that the training protocol started when mice were 9 weeks old (end of growth).
 - the need to identify the exact residue that are ubiquitinate to be able perform genome-wide approach to confirm that ubiquityl-H3 help restore or prevent silencing by PRC2.

Point by point answers to specific points.

Editor's comments: We have added additional information regarding the anaesthesia protocol with more information about dosages, route and evaluation of deep anaesthesia. See below the text that has been added in the main text.

“Inhaled isoflurane/oxygen was used to achieve deep anesthesia during tissue collection. First the animal will be placed in the induction chamber with no isoflurane flow but with oxygen flow (1-2L/min). Then, isoflurane flow was increased gradually (0.5% up to 5%). When the animal reaches deep anesthesia, the isoflurane flow will be decreased to 3%. The animal will be transferred to the surgery table and immediately placed into a non-rebreathing tubing to maintain isoflurane delivery. Mice were placed on a warming pad to maintain core body temperature while the animal were under anesthesia. Our previous experience with sedentary and trained mice suggests that 3% isoflurane with 1.5 l/min oxygen flow rate is usually sufficient for mice of this age and weight to be maintained under deep anesthesia. Peri-anesthesia, animals were monitored for absence of pain reflex (foot withdrawal from

pinch) prior to commencing surgery. Animals were under close observation during the full anesthesia procedure, to ensure that deep anesthesia was maintained throughout the protocol.”

Now, all figures show mean \pm SD and present exact p values everywhere. Some data originally presented in the supplemental have been moved in the main figures, and we have reordered the figures, re-arranged the order of panels within some figure and created new figures. Please see below a point-by-point answer to the reviewers’ comments.

Referee 1

Point 1: Now, all data show mean \pm SD. The figure 1 is now included the representative images of the staining as well as the merged images. We included missing values for PECAM measurements in fig. 1 A (PECAM in gastrocnemius muscle). Initially, we did not test for outlier. After testing again all the western blot quantification, two outliers were removed from the plantaris PECAM measures based on the Grubb method with $\alpha=0.05$. p values of comparison statistically different with $p\leq 0.05$ are now shown directly into the figure.

Point 2: All western blots were optimized for each specific tissue or cell type before performing analysis of whole experiments. Before starting to analyze samples, experimenters have verified their capacity to quantify changes in protein on both loading control and protein of interest. Rep-fig.1 (next page) illustrates some of the experiments perform to ensure optimal quantification of western blots. For loading control, we did test $\alpha\beta$ -tubulin and β -actin. $\alpha\beta$ -tubulin showed better representation of overall total protein measured by red ponceau or total protein, as shown in Rep_Fig1 below. The variability observed in tubulin is due to slight variation in total proteins. Normalization with $\alpha\beta$ -tubulin appears as the most appropriate approach under our conditions of western blotting to avoid saturation of signal where blotting for protein of interest and loading control on the same membrane, conditions optimal for accurate estimated for relative protein content.

It is important to note that for all western blots, red ponceau were performed to ensure homogenous transfer was achieved. We used to acquire western blot imaging on a Kodak 4000 MM Pro; Carestream Health, Concord, ON, Canada. This image station was not optimal to record ponceau red image (not inverted bright light source); yet efficient to acquire ECL signals. Red ponceau images were performed on cell phone camera. An example is now provided below (Rep_Fig1). We recently purchased a more efficient imaging station (IbrihtFL1500, Thermofisher) allowing us to acquire images of red ponceau, total protein and ECL signal efficiently. A certain number of experiments especially figures 1 and 2 were performed before we had accessed to this new imaging station.

Rep Figure 1: Example of one western blot optimization and calibration of experiment for $\alpha\beta$ -tubulin and histone H3 used to expression relative protein expression in muscle protein extracts. **(A)** impacts of loading increasing concentration of protein extract from control gastrocnemius muscle on the quality of H3K27me3 detection at 25 KDa and associated loading controls $\alpha\beta$ -tubulin and H3 (3 minutes acquisition time on Kodak 4000MM). **(B)** Western blotting signals after final optimization of

H3K27me3 and $\alpha\beta$ -tubulin with 10 and 20 μg of a pool of sedentary (pS) and trained (pTr) gastrocnemius protein extracts to ensure detection of protein quantity is efficient (Kodak 4000MM). No saturated signals are observed for H3K27^{me3}, H3 and $\alpha\beta$ -tubulin proteins. Both 17KDa and 25KDa bands for H3k27me3 were above the threshold of detection when 20 μg gastrocnemius extract were loaded with loading control not saturated. **(C)** Comparison of Ponceau red staining of membrane and western blot for $\alpha\beta$ -tubulin and β -actin to test most efficient loading control on our conditions on randomly selected protein extract from hindlimb muscles analyzed after western blotting (20 μg of protein, Kodak 4000 MM Pro). **(D)** Comparison of Ponceau red, no-stain protein labeling and $\alpha\beta$ -tubulin and MDM2 (2A10 blot) western blotting on gastrocnemius muscle samples remaining from proteins analyzed in Figure 2 (Ibriht FL1500, 20 μg of protein, 3 sedentary and 3 trained mice).

Point 3:

The 25KDa band is present in the C2C12 myotubes and less abundant in the mouse muscle endothelial cells, this is an observation that all experimenters in my laboratory have observed. It is difficult to detect the 25KDa in endothelial cells. We do agree that other cell types found within the muscle might express this band. We hope that our work will instill curiosity about this form of HPTMs to investigate whether this ubiquityl-H3 could be a hallmark of adaptation resolution in cells supporting muscle tissue remodeling.

- Fig.4B is a qualitative observation. Therefore, we have edited the text in the results section to ensure it is clear to the reader. And, we have added a comment in the discussion to acknowledge other cell types might express this ubiquityl-H3.
- While major gap of research remains regarding the role of H3 mono-ubiquitination, it cannot be excluded as other cell types could express this form of H3. Multiple mono-ubiquitination of H3 have been reported on a few lysine residues; K4, K18, K23, K121, K125. Mono-ubiquityl forms of H3 are associated to the activity of other E3 ligase RNF8, CUL4A/B, UHRF1, NEDD4 or CUL4A/B that might be associated to either the maintenance of DNA methylation, DNA replication, DNA damage repair (DDR) and modulation of chromatin structure (as review in (Oss-Ronen *et al.*, 2022)). It cannot be excluded that cells having phenotypical flexibility and playing a role in replenishing and supporting muscle regeneration and adaptation might harbour this ubiquityl-form, multiple cell types could be candidate macrophages, T-regulatory and other cells within the satellite cell niche (Sawano *et al.*, 2014; Walton *et al.*, 2019; Tokinoya *et al.*, 2021; Wu *et al.*, 2022).

In our previous submission we wrote: “*The band detected at 25kDa was present in C2C12 myotubes and was significantly less abundance and less detectable in mouse skeletal muscle endothelial cells (mSMECs), suggesting a degree of cell-type specific expression of this form of histone modification (Fig. 4B).*” We have edited the sentence, it does now read as “*The band detected at 25kDa was present in C2C12 myotubes **and was difficult to detect** in mouse skeletal muscle endothelial cells (mSMECs), suggesting a degree of cell-type specific expression of this form of histone modification (Fig. 4B).*”

We added a comment on the eventuality that other cell types might express H3Ub in the muscle tissues as well as the fact that EZH2 and H3K27me3 might have different role in muscle stem cells (see new paragraph 11 of discussion pages 44-45).

Point 4:

Both referees have shown interest in investigating whether *Kdr* and *Notch1* show different pattern in expression in the soleus muscle. We have performed additional experiments to explore this point and assessed whether the Ubiquityl-rich soleus at baseline shows differences in pattern of *Kdr* and *Notch1* expression. We also added more information regarding the impact of training on the global level of H3K27me3 at 17 and 25KDa for this muscle, as well as on MDM2 and EZH2 expression. We did compare the soleus muscle to the plantaris muscles in the new figures 4 and 5. The results section has been modified accordingly.

Point 5: The figure panels have been re-arranged following the reviewer recommendation, improving reading significantly. Since we added additional results in the section other figures have been modified to include most data related to other muscle types. The flow of ideas of the figure

Point 6: We added more information regarding the E3 ligase function of MDM2 in the discussion (last paragraph). While we did mention that MDM2 is a E3 ligase in the introduction we do agree that further information is required regarding its E3 ligase activity. We now provide more information specify that MDM2 is a Ring domain E3 ubiquitin ligase, providing rationale for using the Serdemetan inhibitor, which inhibits its capacity to transfer ubiquitin but not interfere with MDM2 protein binding activities. At this point, we do not know whether the ubiquitin is a mono- or poly-ubiquitination. Mdm2 has been reported to support mono-ubiquitination, multiple mono-ubiquitination and poly-ubiquitination to its protein target. When you chose the antibody to detect ubiquitin after immunoprecipitation with α -H3K27me3 or α -H3, we chose the clone P4D1 this antibody detects both unconjugated and conjugated ubiquitin including mono-ubiquitylated and poly-ubiquitylated. It does not allow to distinguish K48 or K63 linkages. Yet, based on the molecular weight observed, a mono-ubiquitination is most probable.

To answer the question of whether the H3Ub leads to proteolysis, we have treated C2C12 cells with MG132, an inhibitor of the proteasome. After 4 days of differentiation C2C12 myotubes were treated for 7 hours with MG132 we observed no change in the protein abundance of H3K27me3 (17 and 25 KDa) and H3. Since we observed no accumulation of ubiquityl-H3 detected at 25KDa with the α -H3K27me3 antibody when MG132 is present, it is plausible that MDM2 supports a mono-ubiquitylation of H3 that is non-proteolytic under our cell culture conditions.

Point 7: The greater level of H3K27me3 in the soleus is observed at baseline and not in response to training. You have edited the sentence to improve reading. Our data brings two important information:

1. The soleus muscle has a greater abundance of H3K27me3 at 17KDa and 25KDa at baseline when compared to the plantaris and gastrocnemius muscles.
2. Endurance training increased the abundance of the 25KDa band of H3K27me3 in the plantaris (former figure 6B) and gastrocnemius (former figure 3D) muscles.
3. We performed additional analysis of the soleus muscle. These new results support the notion that the soleus has a divergent of response to training when compared to glycolytic-dominant muscles (rich in fiber type IIB, figures 4 and 5).

To support better reading, we added the text in red to bring further clarification regarding points 1 and 2 above.

Page 34 was edited see added text in red : *“This suggests that in response to endurance exercise training, activation and silencing HPTMs collaboratively shift transcription towards a more oxidative phenotype. Interestingly, at baseline, we report an overall greatest abundance of both H3K27me3 bands (17KDa and 25KDa) in the more oxidative soleus muscle compared to glycolytic plantaris and mixed-phenotype gastrocnemius muscles. Yet and in contrast to the observations made on plantaris and gastrocnemius muscles, the overall abundance of H3K27me3 detected at 25KDa remained unchanged in the soleus muscle after endurance. Together, these findings lead us to question whether these PRC2-related PTMs of H3 could be key for acquiring greater oxidative metabolism and a higher capillarization phenotype in muscles.”*

Page 36 the text now reads as below (text added in red): *“The abundance of this putative H3^{Ub} form was significantly greater in myotubes when compared to myoblasts. The highly oxidative soleus muscle has greater level of ubiquityl-H3 than other muscle at rest. Yet, weeks of training led to greater level of H3K27^{me3} detected at 25KDa in the plantaris and gastrocnemius muscles (trained versus sedentary mice), not in the oxidative soleus muscle. And repeated exposures to electrostimulation increased the levels of this putative form of H3^{Ub} in C2C12 myotubes, mimicking the increase observed after endurance training in the tissue of glycolytic and mixed muscles.”*

The previous figure 6 presented baseline data comparing soleus, gastrocnemius and plantaris muscle. As well as the impact of training of the plantaris but not in the soleus. Now, additional data has been included in the figures 4 and 5 to show the impact of endurance training on the H3K27me3 forms, EZH2 and MDM2 in the soleus muscle and on the plantaris muscles. The flow of ideas of the results had been adjusted accordingly.

In the soleus muscle, *Notch1* was significantly decreased after 9 weeks of training while *Kdr* remained unchanged. Interestingly, MDM2 was decreased after training in the soleus muscle (see new figures 4 and 8). H3K27me3 detected at 25kDa remained unchanged after weeks of training in the oxidative soleus (see new figure 5). This confirmed that the regulation of H3K27me3, MDM2 and *Notch1* is highly dependent on the muscle phenotype. Indeed, new added data indicates that the soleus muscle had a divergent response to training when compared the gastrocnemius and plantaris muscles. The capacity of endurance training to increase *Notch1*mRNA through H3 ubiquitination might only be true in muscles that have a predominance of type IIB or mixed phenotype capable of engaging in a metabolic shift in response to endurance training. Indeed, while the plantaris and gastrocnemius muscle presented an increased in the 25KDa band in response to training. This response is not observed in the soleus muscle.

Point 7: We did not measure the level of *Kdr* in the C2C12 cells as we believe that *Kdr* is mainly expressed in endothelial cells. Ct values for *Kdr* are right high, indicating very low level of expression in our C2C12 myotubes culture. At 9 weeks of training *Kdr* has an enrichment in H3K27me3 and the concomitant reduction in *Kdr* mRNA might support the idea that a EZH2-related silencing.

Point 8: We have previously reported that 7 weeks of endurance training fails to promote muscle angiogenesis in MDM2 hypomorphic mice, which expressed only 35% of the normal level of MDM2 while wild-type litterate mice do (Roudier *et al.*, 2012). So, we already know that a lack of MDM2 *in-vivo* reduces endurance-driven muscle angiogenesis. This might be in part due to a lack of capacity of MDM2 to interact with some of its targets, more particularly transcription factors such as Foxo1 in endothelial cells (Aiken *et al.*, 2016). We do not know whether MDM2 activity could counteract H3K27me3-dependent silencing of *Kdr*. In other cell types, MDM2 has been shown to interact with

EZH2 to either support or reduce the protein stability of this methyltransferase. As endurance increases MDM2 proteins, as reported here in mice and previously in human muscle (Roudier *et al.*, 2012, 2013), this E3 ubiquitin ligase might support a greater interaction with the chromatin leading to greater ubiquityl-H3 (Riscal *et al.*, 2016b, 2016a). Unfortunately, we do not have access to the hypomorphic mice currently. Future in-vitro studies might help answering this question by studying the regulation of EZH2 and MDM2 in skeletal muscle endothelial cells. This mechanism will be best tested when the angiogenesis process is at its peak 3 to 4 weeks of endurance exercise training. We edited the two last paragraph of the discussion to address this point.

General comments –

Referee 2:

The referee points out that one important discovery of the paper is the differences observed between highly glycolytic and highly oxidative tissues in term of global abundance in the ubiquityl-form of H3. We do agree with this comment. Our goal was not to perform an in-depth analysis of the differences between muscle types. Our study focuses on investigating the impact of training rather than investigating the muscle phenotype at baseline. The data showing the increased abundance of both 17KDa and 25KDa in the soleus compared to mixed muscle, aimed to bring evidence to further question whether the greater abundance of H3K27me3 detected at 25KDa (i.e. ubiquityl-H3 form) after training could be a feature of a greater oxidative metabolism and greater capillarization in mixed muscle, i.e. gastrocnemius and plantaris. A comparison with the soleus is always valuable as it is postural muscle and highly rich in type IIA fiber when compared to any other muscle rich in IIB and IID (TA, Pla, o Gastroc) in C57Bl6 mice. Due to the muscle mass of plantaris or soleus (8 to 12 mg on C57bl6J) only a limited number of experiments could be conducted on these muscles, performing ChIP is challenging impossible without pooling muscle from different mice. Yet muscles presenting a predominance of type IIB/IID fibers seems to have similar responses. We collected new data on plantaris and soleus muscle that are now presented in figures 4, 5 and 8.

We also agree that it will be valuable to perform omics approach to better capture the global scope of epigenetic remodeling. This will require to perform ChIP-sequencing (using a H3K27me3) complemented by RNA-sequences. This type of experiments is relatively costly (our lab is current running on a 28'000 CAD annual budget). We will consider this approach for future funded studies. Yet, we still believe that the current results are valuable. Indeed, the current study was designed to bring proof-of-concept that it is worth pursuing in this direction with omics approach in a second steps once more funding will be secured.

The reviewer commented on our sample size being relatively limited. A sample size of 7 was used for our training protocol, a smaller sample size, n=5, was used for the muscle type comparison. The n numbers were determined as indicated below.

Calculation of sample size for comparison of muscles: This calculation was based on our observation that major differences were observed between the plantaris and soleus muscles, in regard to MDM2 expression (Roudier *et al.*, 2012). The n values for our experiments we based on the assumption that muscle type and training will have similar impact on H3K27me3 than previously reported by us on MDM2 protein Roudier et al 2012 (FASEB J.), since we expected H3K27me3 to be

regulated in an MDM2-dependent manner. So, we calculated our n numbers using the formula below, keeping in mind the three R guiding principles

$$N = \frac{(r + 1) \left(\frac{Z_{\alpha/2} + Z_{1-\beta}}{2} \right)^2 \sigma^2}{rd^2}$$

where $r = n_1/n_2$ - the ratio of sample size, σ - pooled standard deviation, d - difference of means of 2 groups, $Z_{1-\beta}$ - 0.84 for power 0.80, $Z_{\alpha/2}$ -1.96 for alpha 0.05. Based on our previous publication regarding differences in MDM2 expression between soleus and plantaris (Roudier et al FASEB J 2012), $\sigma=0.25$, $d=0.40$, $n=7$ for both groups. N will be equal to $((1.96+0.84)^2 * 0.25^2)/(1*0.40^2)=3.1$. By choosing $n=5$, we should be able to detect a difference in protein intensity between two groups of roughly 0.35 (any increase of roughly 40%). The effect reported here for the ubiquityl-form of H3K27me³ are much greater (see former figure 6).

Training sample size: We did use a similar approach to calculate keeping in mind that training has a smaller impact than muscle type on MDM2 level. From the previously published data, we took in consideration $\sigma=0.28$, $d=0.30$, retrieving a n value of 6.8 when using a power of 0.8 and $\alpha=0.05$. Therefore, we used $n=7$ for our training conditioning. We initially add $n=9$, muscle from mice unable to acclimate to the treadmill were removed from the protocol. The remaining mice were randomized in two groups: one sedentary group and one trained group. All animals were manipulated similarly. While sedentary mice were brought in the treadmill room, handled and put on the treadmill a few minutes. And stay in the room while the trained mice ran. This aimed to ensure the numbers of manipulation and similar exposure to noise. Mice removed from the protocol were kept on similar condition, muscles were collected as for other groups and used to set up and optimize assays (e.g. DUB, ChIP).

Sample size of ChIP-qPCR: Initially, we aimed to perform most of our experiments on the same muscle, i.e. gastrocnemius. Yet, after performing protein and mRNA extractions, preparing samples for DUB and immunoprecipitation, we randomly selected 4 controls and 4 trained gastrocnemii for which we had sufficient muscle tissue left to perform the ChIP assay. We decided to run the experiments on the gastrocnemius with $n \geq 4$, which has been reported by others to be sufficient to retrieve difference in H3K27me³ enrichment when MDM2 expression varies (Wienken *et al.*, 2016). For figure 7 biological replicates have been added from $n=5$ to $n=8$. See new figure 7 and below in our reply to comment related to the result section.

Methods related comments:

Point 1: Formerly page 4. Mice arrived in house at the age of 7 weeks. After one week acclimating to the animal facilities, they started to be exposed to the treadmill the following week (5 days a week). So, the training program started when the mice were 9-week-old. We cannot exclude that results might have been different if mice were fully mature 3 to 6 months old (12-24 weeks). We have added a few sentences in the limitations to acknowledge that young mice might have a greater skeletal muscle plasticity. Most exercise training protocol in C57Bl6 mice including from our own work are commonly started when mice are 10 to 12 weeks old. See discussion new paragraph 11 page 44. In this paragraph we are now discussing the fact that mice are not fully mature when the endurance

training protocol starts and questions whether this might have an impact on our results, knowing that maturation and aging has an impact on the level of expression of EZH2 in the muscle and H3K27me3

“Expression of EZH2 and H3K27me3 is crucial to muscle development and growth (Caretto et al., 2004; Juan et al., 2011; Woodhouse et al., 2013). EZH2 is highly expressed in activated muscle stem cells leading to an increased abundance of H3K27me3 in young mice (8 weeks). In pre-mature mice, the EZH2 pathway might support the acquisition of a permissive state of chromatin where genes important for muscle growth and adaptation are marked with both H3K27me3 and activating HPTMs (i.e. H3K4me3) (Liu et al., 2013; Shimizu & Kawano, 2022). In adult mice, EZH2 supports muscle regeneration through the maintenance of the pool of stem cells (Juan et al., 2011). As mice grow and age, muscle EZH2 expression decreases gradually postnatal and H3K27me3 abundance increases globally in quiescent muscle stem cells of older mice (i.e. 24 months) (Juan et al., 2011; Liu et al., 2013). It is important to note that in our study mice were trained from week 9 to week 18, reaching mature age while being engaged in endurance training for a few weeks. Our observation that endurance training increases H3Ub might only prevail in young adult mice where muscle stem cells retain a permissive state of chromatin that might favor muscle adaptation and epigenetic memory (Liu et al., 2013; Sharples et al., 2016). Under similar endurance training protocol, Murugathasan and colleagues reported that macrophages present important changes in their phenotype due to important epigenetic remodeling (Murugathasan et al., 2023). Whether ubiquitylation of H3 plays a role in determining the phenotypes of muscle macrophages remains largely unknown in response to training. It cannot not be excluded that the increase of H3Ub observed *in-vivo* could be driven by changes in the abundance and phenotype of muscle stem cells and macrophages (Juan et al., 2011; Liu et al., 2013; Sawano et al., 2014; Tonkin et al., 2015; Walton et al., 2019; Oss-Ronen et al., 2022; Wu et al., 2022). Future investigations conducted on young, mature and older mice are warranted to delineate the exact role H3Ub in different cell types recruited during muscle adaptation to training across mouse lifespan.”

Point 2: The animal conditioning section was confusing in the method has been edited and now reads as it follows:

“Fourteen female C57Bl6 mice arrived in the animal facilities at the age of 7 weeks. All mice were fed the standard chow. After one week of quarantine, all mice were acclimatized to treadmill exercise for 5 days at 20m/min prior to the eight-week training program, before being randomized in a trained vs. sedentary group (n=7 per group, at the age of 9 weeks) (trained and sedentary) Sedentary mice were allowed to roam their cages. During the training program, trained mice ran for one hour per day, five days per week, as previously described (Murugathasan et al., 2023). The speed was gradually increased up to 30 m/min as the training program progressed: week one – 20 meters/min, week two - 22m/min, week three - 24.2m/min, week four - 26.6m/min, weeks five-nine - 30m/min. This represents speeds of 1.2 kmph to 1.8 kmph. The mice ran at approximately 70-75% of their VO₂ max, which can be characterized as moderate to vigorous intensity. Sedentary mice were exposed to the same environment and manipulations. Instead of running, sedentary mice were put in and out the treadmill and stay in their cage exposed to the sound of the treadmill as the trained group was performing their running session. After 9 weeks (1 week treadmill acclimatization + 8 weeks of training), sedentary and trained mice were anaesthetized (isoflurane/oxygen inhalation) 72 hours after the last bout of exercise. Inhaled isoflurane/oxygen was used to achieve deep anesthesia during tissue collection. First the animal will be placed in the induction chamber with no isoflurane flow but with oxygen flow (1-2L/min). Then, isoflurane flow was increased gradually (0.5% up to 5%). When the animal reaches deep anesthesia, the isoflurane flow will be decreased to 3%. The animal will be transferred to the surgery table and immediately placed into a non-rebreathing tubing to maintain isoflurane delivery. Mice were placed on a warming pad to maintain core body temperature while the animal were under anesthesia. Our

previous experience with sedentary and trained mice suggests that 3% isoflurane with 1.5 l/min oxygen flow rate is usually sufficient for mice of this age and weight to be maintained under deep anesthesia. Peri-anesthesia, animals were monitored for absence of pain reflex (foot withdrawal from pinch) prior to commencing surgery. Animals were under close observation during the full anesthesia procedure, to ensure that deep anesthesia was maintained throughout the protocol.”

Point 3 (formerly page 7). We have now provided more information regarding confluency in the methods.

Points 4 to 7 (formerly page 8). The solvent used for GSK343 and Serdemetan were DMSO. It is now specified in the methods clearly. Control dishes were treated with equal concentration of DMSO for the same volume of cell media. Time of incubation is either 24 hours or 48 hours the text has been edited. The method section has been modified as indicated below (added text in red)

“Stock solution of Serdemetan and GSK343 were dissolved in DMSO (Sigma Millipore, Burlington, ON, Canada, cat no D2650) prepared at 1000x. After 4 days of differentiation, C2C12 myotubes were incubated with an EZH2 inhibitor (GSK343; 1 and 5µM, Cell Signaling, cat. no. 66244), an MDM2 inhibitor (Serdemetan; 1µM, SelleckChem, Burlington, ON, Canada, cat no. S1172) or vehicle control (DMSO, dilution 1/1000) for 24 or 48 hours. Inhibitor-treated and vehicle control myotubes were subsequently subjected to EPS for 90 minutes a day for 4 days (as described above) before cell harvesting.”

We chose GSK343 on purpose as based on IC50 reported by supplier it is a more specific inhibitor of EZH2 than EZH1. We added one sentence in the discussion. To specify that this is a limitation and that it cannot be excluded that EZH1 could partially compensate EZH2 activity when EZH2 is inhibited. Added references regarding the role of EZH1 and EZH2 in the skeletal muscle, including PMID 3566835).

Point 8: The method section paragraph related to statistical analysis has been improved:

1. we have now clarified the number of biological and technical replicates performed. It now reads as follows:

“For in-vivo experiments, protein and mRNA analyses were performed on all animals (n=7 per group) for most proteins and genes of interest. Some protein analyses had lower n number (n=5) when samples with lower protein concentration become unavailable. For in-vitro analyses, one representative experiment is shown with n=6 biological replicates per conditions; except when qualitative analyses were performed (i.e. Fig.4A and Fig.5, n=1-3). Most western blot analyses were performed using one technical replicate. When blotting issues raised, a second technical replicate was performed for all samples. When two technical replicates were available the average of both values was used to express relative protein quantity. For qPCR measures, 3 technical replicates were performed. When the standard deviation for the Ct technical replicate was ≥ 0.3 , 3 additional technical replicates were performed. All technical replicates (3 or 6) were used to perform the $\Delta\Delta Ct$ analyses.”

2. We provide more information regarding strength of correlation, defining weak, moderate and strong correlation when analysing ChIP-qPCR data. The paragraph was edited and now reads as below:

“Statistical analyses were performed with Student’s t test and 1- and 2-way ANOVAs using Prism 9.2 (GraphPad, San Diego, CA, USA). For 1- and 2-way ANOVAs, Tukey’s multiple comparison and Bonferroni post hoc tests were used, respectively. Stand alone comparison used Fisher’s LSD test.

Welch's correction was performed when variances were significantly different. To test correlation between the distribution of HPTM on H3, we have performed a Pearson's coefficient analysis and a linear regression. Squared r values (r^2) with a range from 0.8 to 1 (i.e. [0.8-1]) were considered as strong, r^2 ranging [0.5-0.79] were considered as moderate, and r^2 ranging [0-0.49] were considered as weak. For all statistical analyses, a value of $p \leq 0.05$ was considered to be statistically significant. Outliers were identified using the Grubb's method with an α value equal to 0.05."

Point 9: We have added information regarding the EPS setting providing a short rationale for choosing 8 volts at a frequency of 5 Hz, it was derived from the original publication by (Connor et al., 2001).

Briefly, we aimed to generate contractile activity mimicking an acute bout of submaximal muscle contraction closer to aerobic exercise and could be repeated over a few days. Typically the motoneuron frequency in vivo of slow twitch fiber is 5 to 20 Hz and during acute aerobic exercise muscles contract at a frequency of ~ 1 Hz (Vepkhvadze et al., 2021). In-vivo, submaximal exercise/muscle contractions generally involve wave summations (as opposed to twitches (Tamura et al., 2020a). Lower frequency activations (1-3Hz) correspond to twitch contractions. No twitches were observed in frequencies above 10Hz; this may indicate the threshold for tetanic contractions (Marotta et al., 2004; Tamura et al., 2020a). Therefore, the desired frequency to investigate submaximal contractions may lie between 4Hz and 9Hz. A frequency of 5Hz because it falls within this range, it was the frequency used by (Connor et al., 2001) when determining optimal voltage, and Pan and colleagues (Pan et al., 2012) reported an increase in IL-6 production (an expected outcome for exercising muscle) with a frequency of 5Hz.

We chose 8V is based on previous work by Dr. Connor (Connor et al., 2001). The authors reported that C2C12 myotubes contracted synchronously at 5Hz with voltages as low as 35V. Myotubes reached maximal shortening at 55V with no further increase in contraction intensity between 55 and 110V. Therefore, they decided on a voltage of 65V ($1.2\text{V}/\text{cm}^2$). Based on their setup, this 55V threshold would translate to a voltage/area of $\sim 1\text{V}/\text{cm}^2$ and 35V is $\sim 0.6\text{V}/\text{cm}^2$. In our experiments, the C2C12 myotubes were plated in 6-well plates. Each well has a surface area of 9.6cm^2 . Therefore, to achieve the same threshold for maximal shortening, our voltage setting should be 9.6V. The minimal threshold for synchronous contractions is 5.8V. Our EPS conditions is meant to mimic submaximal exercise. Therefore, a voltage between the minimal threshold for synchronous contraction (5.8V) and maximal contraction (9.6V) was chosen (i.e. 8V). FYI, since the EPS circuit/wiring for each well in the 6-well plate is connected in parallel (not series), the voltage for each well will be the same as the total voltage (8V).

The paragraph now reads as it follows (added text in red): "Differentiated C2C12 myotubes were subjected to electrostimulation (EPS) to mimic skeletal muscle contraction **observed at submaximal aerobic exercise (Connor et al., 2001; Carter et al., 2001; Vepkhvadze et al., 2021)**. C2C12 myotubes were allowed to differentiate for 5 days prior to being exposed to EPS. The C2C12 cells were stimulated using a 6-well cell culture lid which was modified with two parallel platinum wire electrodes extending into the wells containing medium. A total volume of 5 mL per well was required to ensure consistent and equal delivery of electrical pulse stimulation throughout the media. Cells were stimulated using a Harvard Apparatus Stimulator CS System (Harvard, MTL, QC). 8 volts were applied at a frequency of 5 Hz with alternating polarity of electrical current (5ms) for 90 minutes once a day for four consecutive days. **These conditions support synchronous contractions below maximal contractility and avoid tetanic contractions**(Connor et al., 2001; Marotta et al., 2004; Pan et al., 2012; Tamura et al., 2020b). The cultured myotubes were harvested after 3 hours of the last bout of EPS."

Results

In the introduction, we wrote “Based on our previous observation that endurance training increases the expression of MDM2 protein in the skeletal muscle (Roudier *et al.*, 2013), we propose here to investigate whether training changes the abundance of MDM2-EZH2 related histone silencing marks in the skeletal muscle after adaptations have settled.” It is important to note that intensity of exercise is increased progressively over 5 weeks followed by 4 weeks at the constant intensity. We did not intent to claim that H3K27me3 (not DNA methylation) is a silencing mark that capped muscle adaptations. We want to investigate whether this silencing mark related to MDM2 and EZH2 are modified by endurance training. We have read and edited the manuscript in the key points section to ensure this idea of capped muscle adaptation is not present in the manuscript. Now it reads as it follows: “Whether establishing epigenetic silencing histone marks play a role **once skeletal muscle adaptations have occurred post endurance training** remains unclear.” We have also edited the conclusion to ensure consistency in the flow of ideas.

Figure 1: Images of capillary-to-fiber ratio have been included now the figure. New data shows all biological replicates as well as the supplemental data. New figures 4, 5 and 8 present more data related to plantaris and soleus. Yet ChIP analyses could not be performed in the muscle due to their limited amount of material available. All MDM2 blots quantified both the 75KDa and 100 KDa. Figures have been modified accordingly to indicate in brackets the bands analysed. We have redone the western blotting analysis for MDM2 using the 2A10 antibodies. This new figure presents a revised version of the western blot images.

Figure 2A. Now, exact p values are now shown for all statistically differences in all figures.

Page 19. The DUB experiment has been performed on gastrocnemius muscles from 3 sedentary mice and 3 trained mice, consolidated our previous data (see revised Fig.6)

Comments related to page 22. New figures have been added and the flow of idea re-organized in the “results” section. All muscle-related data now precede C2C12 in-vitro experiments.

- Muscle adaptations are presented (figure 1)
- Gastrocnemius muscle data related to epigenetic writers (Fig. 2) followed by histone marks (Fig.3)
- Data comparing plantaris and soleus muscle (epigenetic writers Fig.4 and histone marks Fig. 5)
- Identification of the ubiquityl form of H3 in muscle tissue (Fig. 6, with 3 biological replicates from sedentary and trained muscle n=3 per group)
- Gene expression in muscle and presence of histone marks on NOTCH1 and KDR genes (Fig. 7 with new biological replicates for Fig.7C)
- Confirmation that the soleus muscle is divergent from mixed muscle in term of *Notch1* and *Kdr* expression (Fig. 8)

- Confirmation of the presence of ubiquityl-H3 in C2C12 cells and increased abundance after EPS (Fig. 9) and new data with MG132 inhibition.
- Impact of EZH2 inhibition on C2C12 myotubes (Fig. 10) with new data combining EZH2 inhibition and EPS
- Impact of MDM2 inhibition on C2C12 myotubes (Fig. 11) with new data combining MDM2 inhibition and EPS.
- Confirmation that under our cell culture conditions. After 4 days of differentiation EZH2 inhibition or MDM2 inhibition does not impact the differentiation state of myotubes culture (Fig. 12).

Cell types and main results. We have now added a paragraph in the discussion (page 44-45) to discuss the possibility that other cells can drive some of the changes observed more particularly macrophages and muscle stem cells. The changes observed *in-vivo* could be driven by different proportion of these cells.

Comments related to figure 7. We also addressed most comments from referee 2 regarding the figure 7. See determination of sample size above in §“Sample size of ChIP-qPCR”

Fig. 7B. The analysis of the DNA quantity in INPUT chromatin clearly identified one sample (trained 2) having significant different content in DNA content than all other samples. This sample was clearly identified as outlier using the Grubb’s method ($\alpha=0.05$) and removed before statistical analysis (former Fig 7B). No other outlier was identified, and no other experimental bias justifies removing additional data points. N values are then n=4 for Sedentary and n=3 for Trained.

Fig. 7C. ChIP assay was originally performed on n=5 gastrocnemius muscles. To address the concern of referee 2, regarding a potential outlier driving the main observed effect, we did run additional ChIP on 3 additional gastrocnemius muscles. Initial DNA concentration of the input chromatin is a key determinant of ChIP-qPCR efficiency. We added a table in the supplemental file that includes the DNA concentration of all 8 INPUTs. We ran a Grubbs test that indicates no outlier are present in our dataset. So, we have no valid reason to remove any datapoint.

Table 1. DNA content of chromatin INPUT for figure 7C (added to supplemental file):

	Sample	DNA quantity of input (ng/μl)
	Gastrocnemius-1	10.08
	Gastrocnemius-2	7.75
	Gastrocnemius-3	9.79
	Gastrocnemius-4	12.38
	Gastrocnemius-5	17.04
	Gastrocnemius-6	6.54
	Gastrocnemius-7	10.8
	Gastrocnemius-8	12.5

Rep-Figure 2 compares our previously submitted results, to the new set collected, allowing us to assess the impact of the new muscle collected on the dataset and the correlations for KDR and NOTCH. To address the referee 2’s comment, we assessed the impact of excluding the extreme datapoint, as suggested. For all analysis NOTCH1 had higher abundance of H3K27me3 and H3K4me3 for all dataset (Rep-Fig2. A-C). When removing the extreme datapoint (Rep-Fig.2), NOTCH1 had significantly higher level of H3K27me3 and a trend to higher with H3K4me3 with p=0.073.

In the revised version of manuscript, we decided to show with all data points (as in Rep_Fig2C). By showing all data points, the reader could more efficiently interpret the dataset. We have modified the correlation figures adding the KDR correlation. We also added information in the methods to better define the correlation strength, specifying the threshold used to define a weak ($r^2 = [0.0-0.49]$), moderate ($r^2 = [0.5-0.79]$), strong correlation ($r^2 = [0.8-1.0]$).

Rep_Figure 2: Greater enrichment in H3K27me3 and H3K4me3 in Notch-1 when compared to KDR in gastrocnemius muscles. **A**) Results previously submitted (n=5, 2W-ANOVA); **B**) new biological replicates (n=3, 2W-ANOVA); **C**) data combining all data points (n=8, 2W-ANOVA); **D**) Impact of removing the putative outlier identified by referee 2.

We did run more correlation test. Linear regressions were performed on the original dataset (n=5), the new muscles analysed (n=3) and the full dataset (n=8) (**Table 2**). **Rep-Fig3** shows the correlation curves panel C and D have been inserted in the revised figure 7 (previous only Rep-Fig3A was shown). Despite no true outlier being identified, we followed the suggestion of referee 2 and tested the impact of removing the extreme data point corresponding to gastrocnemius muscle #5. The best fitted curve to analyze the dataset was a polynomial quadratic equation (Rep_Fig 4). Panel **A** shows the fitted curves for the whole data set (n=8), a strong correlation was observed for **Notch1** $r^2=0.8232$ and a moderate correlation for **KDR** ($r^2=0.5041$). Panel **B** shows the fitted curves when the extreme

datapoint is removed (referee 2 suggest, not a true outlier). A strong correlation is observed for **NOTCH1** $r^2=0.9269$, a weak correlation was observed for **KDR** $r^2=0.2888$.

	Dataset	Notch 1			KDR		
		r^2	Slope (a)	p slope $\neq 0$	r^2	slope	p slope $\neq 0$
Linear regression ($y=ax+c$)	Set 1, n=5	0.8237 Strong positive	0.7767	0.033	0.4293 Weakly positive	0.4519	0.2300 ns
	Set 2, n=3	0.9988 Strong positive	3.061	0.0221	0.9004 Strong flat slope	3.009	0.2044 ns
	Set 1&2 n=8	0.8173, strong positive	0.7541	0.0021	0.4490 Weakly positive	0.4470	0.0502 ns

Table 2: Correlation analyses and regression indicates a stronger correlation between the enrichments in H3K27me3 and H3K4me3 on the regulatory sequences of NOTCH1 when compared to KDR.

Rep_Figure 3: Correlation between H3K27me3 and HK4me3 on KDR and Notch regulatory sequences. **A-B**) Results previously submitted (n=5, 2W-ANOVA); **C-D**) data combining all data points (n=8, 2W-ANOVA).

Rep_Fig 4: Fitted curves to test relationship between H3K4me3 and H3K27me3 in the whole dataset ($n=8$, **A**) and when removing referee 2 extreme point ($n=7$, **B**).

Fig. 7 E to F. We did not have enough gastrocnemius muscle left to run additional ChIP on sedentary and training muscle. We therefore performed additional ChIP assays on the TA muscles to analyze the impact of training. TA and gastrocnemius muscles have a predominance of type IIb and IIbd fibers (IIb and similar distribution of muscle fiber types in adult C57Bl6/J mice, especially when considering fiber type IIb (60% to 55%) (Augusto *et al.*, 2004; Ballak *et al.*, 2014). Yet, due to the plantaris size (9 to 12 mg), it is impossible for us to run ChIP experiments on this muscle. We opted for utilizing the TA muscle, despite this muscle having significantly less type 1 fiber than the plantaris and gastrocnemius muscles. Since we observed an increased in the Ubiquityl-H3 for muscle having a predominance of type IIb fiber, the TA represented the best alternative to perform additional ChIP experiments to further investigate whether training could drive a greater H3K9 acetylation on Notch promoter, since ubiquityl-H3 was reported to easy H3 acetylation in other cell models.

We do agree that this might be confusing to the reader to see a shift in the analysis to a different muscle group. So, we have re-arranged the flow of ideas of the figures and edited the text in the result section to ease reading.

See page 25: “We aimed to investigate whether changes in H3K27me3 could explain changes in basal levels of gene expression after endurance training in glycolytic-dominant muscles rich in type IIb fiber (e.g. gastrocnemius, plantaris and tibialis anterior)(Augusto *et al.*, 2004).

continues page 26: “Next, we tested whether *Notch1* and *Kdr* had differences in H3K9^{Ac} enrichment. A gene effect was observed where *Notch1* had greater levels of H3K9^{Ac} than *Kdr* TSS in another glycolytic and type IIb-dominant muscle (i.e. tibialis anterior, 2W-ANOVA, $n=4$). Training did not impact the enrichment in H3K9Ac of both *Kdr* and *Notch1* promoter (2W-ANOVA, Fig. 7F). Interestingly, post-training *Notch1* enrichment in H3K9Ac tended to be greater than that of *Kdr* (+78%, T-test, $p=0.059$, $n=4$, Fig. 7G).”

We measured the fusion index in our C2C12 culture (4 days of differentiation) and treated the culture with MDM2 or EZH2 inhibitor to test whether it impacted the differentiation states. It was not the case under our specific conditions. As previously stated, we also added a paragraph regarding the potential impact of differentiation in the discussion in-vivo.

We did run additional experiments with EPS +/- inhibition of MDM2 and EZH2 confirming a key role of these epigenetic writers in establishing H3^{Ub}.

Discussion.

Suggested edits have been made to improve clarity and to ensure that the reader clearly understands your points.

You do agree that having a time point at 4 weeks to compare to our current conditioning might help supporting the point that H3K27me3 silencing could support silencing of pro-angiogenic molecules. The present study had a focus on the H3^{Ub}. Yet, we agree it would be a very interesting and valuable study to perform with omics approach. Something we hope to be able to perform in the future. A sentence has been added in the discussion to highlight the need of additional study.

Dear Dr Roudier,

Re: JP-RP-2025-288947R1 "**Endurance training increases a ubiquitylated form of Histone H3 in the skeletal muscle, supporting *Notch1* upregulation in an MDM2-dependent manner**" by Brian Lam, Sokaina Akhtar, Manpreet Gulri, Pierre Lemieux, Monica Tawadrous, Mayoorey Murugathasan, Ali A. Abdul-Sater, and Emilie Roudier

Thank you for submitting your manuscript to The Journal of Physiology. It has been assessed by a Reviewing Editor and by 2 expert referees and we are pleased to tell you that it is acceptable for publication following satisfactory revision.

REVISION CHECKLIST:

We look forward to receiving your revised submission.

Yours sincerely,

Paul Greenhaff
Senior Editor
The Journal of Physiology

REQUIRED ITEMS

- Please check spelling of author name in revised version article (Word) file. It should be:
Ali A. Abdul-Sater

- Author photo and profile. First or joint first authors are asked to provide a short biography (no more than 100 words for one author or 150 words in total for joint first authors) and a portrait photograph. These should be uploaded and clearly labelled together in a Word document with the revised version of the manuscript. See Information for Authors for further details.
[***We have the photo but the profile is missing?***]

- Please include an Abstract Figure file, as well as the Figure Legend text within the main article file. The Abstract Figure is a piece of artwork designed to give readers an immediate understanding of the research and should summarise the main conclusions. If possible, the image should be easily 'readable' from left to right or top to bottom. It should show the physiological relevance of the manuscript so readers can assess the importance and content of its findings. Abstract Figures should not merely recapitulate other figures in the manuscript. Please try to keep the diagram as simple as possible and without superfluous information that may distract from the main conclusion(s). Abstract Figures must be provided by authors no later than the revised manuscript stage and should be uploaded as a separate file during online submission labelled as File Type 'Abstract Figure'. Please also ensure that you include the figure legend in the main article file. All Abstract Figures should be created using BioRender. Authors should use The Journal's premium BioRender account to export high-resolution images. Details on how to use and access the premium account are included as part of this email.
[***We have the figure but the legend is missing?***]

EDITOR COMMENTS

Reviewing Editor:

Your work has been evaluated by the original two reviewers, and both were very positive. The authors did an excellent job responding to the reviewers. There are a few lingering minor concerns from reviewer two that need to be addressed. Once those are managed, we can make a final decision on the manuscript.

Please also see 'Required Items' above.

Senior Editor:

Thank you for the revised manuscript that has been considered by the same individuals that considered the original manuscript. All are of the opinion that the authors have done a good job at addressing the concerns raised when revising the manuscript. Reviewer 2 has a few minor comments that require further consideration.

Please note authors must include in the Methods section a statement that the investigators understand the animal ethical principles under which the journal operates. Please see: <https://physoc.onlinelibrary.wiley.com/hub/animal-experiments>. Please check the section of the Methods titled "Statistical analysis" covers all of the statistical methods employed in the paper. For example, the legend to Figure 7 states non parametric methods was employed. Please also provide a legend to

Figure 1 and a profile to go with the author photograph supplied. Thank you for considering the Journal of Physiology and we look forward to receiving the revised version of the manuscript.

REFEREE COMMENTS

Referee #1:

The authors have made considerable improvements in the revised manuscript. They have appropriately addressed all my concerns from the initial review.

Referee #2:

This reviewer has no further comments other than suggestion to add the light schedule in which mice were kept under and the time of day of tissue collection and training program.

The authors did an outstanding job with these revisions and the final product is a much improved manuscript with very interesting findings. I think the authors succeeded in their goal of "proof-of-concept" and the reviewer hoped that fundings are plentiful for the continuation of this work with multi-omics approaches as mentioned.

END OF COMMENTS

Dear referees and editors.

We are really thankful for the valuable feedback provided and the professionalism you all have offered to improve your manuscript and support our reflection. The whole process was truly helped in generating a strong study.

Please note we have made the following changes as recommended:

- We created a legend for the figure abstract. "Whether epigenetic silencing histone marks play a role once skeletal muscle adaptations have occurred post endurance training remains unclear. In the Polycomb repressive complex 2 (PRC2), EZH2 tri-methylates the lysine 27 on histone H3 (H3K27me3) favoring a repressive chromatin state. As a proof-of-concept, we assessed the impact of training on the muscle H3K27me3. C57Bl6 mice ran for 9 weeks. Our work revealed that training increased the abundance of an ubiquityl-form of H3 (H3Ub) in mixed and glycolytic-dominant muscles. Training led to an H3K27me3 enrichment on the promoter of genes important for muscle remodelling: Kdr and Notch1. Following the canonical model, Kdr expression decreased post-training. Yet, Notch1 mRNA was upregulated. Our results provide evidence that MDM2-dependent ubiquitylation of H3 is required to activate Notch1 expression after repeated exposures to contractile activity. The discovery of H3Ub might explain the dichotomic effect of H3K27me3 marking on Kdr and Notch1 genes."
- We add a sentence in the methods where we acknowledge that all investigators who have handled the mice understand the ethics principles under which the journal of physiology operates.
- We also specify that mice had ad libitum access to normal chow food and were under a 12-hour light/12-hour dark cycle. We also specify the time of the training sessions and muscle collections. We did our best to ensure muscle collections were following a similar schedule than time of exposure to running exercise (trained) or to the treadmill environments.
- We add the missing information regarding how statistical analysis was performed to compare the enrichment in H3K9^{Ac}
- We have provided a photo of the first author (Brian Lam) and a short biography of the first and last authors.

Dear Dr Roudier,

Re: JP-RP-2025-288947R2 "**Endurance training increases a ubiquitylated form of Histone H3 in the skeletal muscle, supporting *Notch1* upregulation in an MDM2-dependent manner**" by Brian Lam, Sokaina Akhtar, Manpreet Gulri, Pierre Lemieux, Monica Tawadrous, Mayoorey Murugathasan, Ali A. Abdul-Sater, and Emilie Roudier

We are pleased to tell you that your paper has been accepted for publication in The Journal of Physiology.

Please see Editor comments below for some items that should be checked/amended at proof stage.

Yours sincerely,

Paul Greenhaff
Senior Editor
The Journal of Physiology

If you would like to receive our 'Research Roundup', a monthly newsletter highlighting the cutting-edge research published in The Physiological Society's family of journals (The Journal of Physiology, Experimental Physiology, Physiological Reports, The Journal of Nutritional Physiology and The Journal of Precision Medicine: Health and Disease), please click this link, fill in your name and email address and select 'Research Roundup':
<https://www.physoc.org/journals-and-media/membernews>

- You can help your research get the attention it deserves! Check out Wiley's free Promotion Guide for best-practice recommendations for promoting your work at: www.wileyauthors.com/eeo/guide. You can learn more about Wiley Editing Services which offers professional video, design, and writing services to create shareable video abstracts, infographics, conference posters, lay summaries, and research news stories for your research at: www.wileyauthors.com/eeo/promotion.

EDITOR COMMENTS

Reviewing Editor:

Thank you for your thorough and detailed responses to the reviewers. No further comments.

Senior Editor:

Thank you for the revised manuscript and for undertaking the changes requested.

We appreciate your willingness to engage with our image checking processes.

Please check your proofs very carefully in due course, to make sure that everything is OK (including most recent versions of figures).

Legend to Fig 7 still states "Paired non-parametric Wilcoxon tested the difference in the abundance of H3K9Ac on Kdr and Notch1 TSS region", which is not mentioned in the statistical analysis section of the Methods. Please make sure this is corrected.

Congratulations and thank you for considering The Journal of Physiology.